# Non-Uniform Noise-to-Signal Ratio
# in the REINFORCE Policy-Gradient Estimator

**Haoyu Han** [1]  **Heng Yang** [1]

## Abstract

Policy-gradient methods are widely used in reinforcement learning, yet training often becomes unstable or slows down as learning progresses. We study this phenomenon through the *noise-to-signal ratio* (NSR) of a policy-gradient estimator, defined as the estimator variance (noise) normalized by the squared norm of the true gradient (signal). Our main result is that, for (i) finite-horizon linear systems with Gaussian policies and linear state-feedback, and (ii) finite-horizon polynomial systems with Gaussian policies and polynomial feedback, the NSR of the REINFORCE estimator can be characterized exactly—either in closed form or via numerical moment-evaluation algorithms—without approximation. For general nonlinear dynamics and expressive policies (including neural policies), we further derive a general upper bound on the variance. These characterizations enable a direct examination of how NSR varies across policy parameters and how it evolves along optimization trajectories (*e.g.,* SGD and Adam). Across a range of examples, we find that the NSR landscape is highly non-uniform and typically increases as the policy approaches an optimum; in some regimes it blows up, which can trigger training instability and policy collapse.

## 1. Introduction

Reinforcement learning (RL) studies how agents learn to make sequential decisions from interaction, and has shown successes across a wide range of domains, including games (Mnih et al., 2015; Silver et al., 2016; 2017), public health (Komorowski et al., 2018; Bastani et al., 2021), robotics (OpenAI, 2020; Song et al., 2023), and, more recently, language-model alignment (Ouyang et al., 2022; OpenAI et al., 2024). At the heart of many advanced RL

algorithms stands the *policy-gradient theorem* (Sutton et al., 1998; 1999; Williams, 1992), which enables gradient-based learning of policy parameters directly from sampled trajectories, without requiring an explicit model of the environment.

Despite their generality, policy-gradient methods are often brittle: training can become unstable or get stuck (Henderson et al., 2018; Dohare et al., 2023). Prior work has identified multiple reasons for this *instability*. One is the *high variance* of stochastic policy-gradient estimators, which can cause updates to poorly track the true ascent direction; this motivates variance-reduction techniques such as baselines, generalized advantage estimation, and actor–critic methods (Sutton et al., 1999; Greensmith et al., 2004; Schulman et al., 2015b; Haarnoja et al., 2018). Another is overly aggressive policy updates, which motivates methods that explicitly constrain update size, *e.g.,* via Kullback–Leibler trust regions in TRPO (Schulman et al., 2015a) or clipping in PPO (Schulman et al., 2017). Other related failure modes include *policy collapse* induced by Adam momentum (Dohare et al., 2023), degradation due to *improper baseline selection* (Chung et al., 2021; Mei et al., 2022), and loss of plasticity caused by nonstationary environments (Sokar et al., 2023; Kumar et al., 2023; Muppidi et al., 2024).

In this work, we are interested in understanding the mechanisms by which *gradient-estimator statistics* translate into optimization behavior. In particular, we uncover the *non-uniform* structure of policy-gradient variance (w.r.t. policy parameters) and how it evolves along optimization trajectories. We focus on the vanilla REINFORCE estimator (Sutton et al., 1999). Despite its simplicity, it already captures sharp behavior near optima and exposes nontrivial variance effects along optimization trajectories; extensions to richer estimators—including those with baselines or entropy regularization—are discussed in the conclusion.

**Problem setup and REINFORCE estimator.** We consider a discounted, finite-horizon Markov decision process with state $s_t \in \mathbb{R}^n$, action $a_t \in \mathbb{R}^m$, reward $r$, horizon $T$, discount factor $\gamma$, and initial state distribution $\rho_0$. A trajectory is $\tau := (s_0, a_0, s_1, a_1, \ldots, s_{T-1}, a_{T-1}, s_T)$, generated by a stochastic policy $\pi_\theta(a \mid s)$. The objective is to maximize the expected return over initial states and trajectories: $J(\theta) := \mathbb{E}_{s_0 \sim \rho_0, \, \tau \sim \pi_\theta}[R(\tau)]$, where $R(\tau) :=$

[1]School of Engineering and Applied Sciences, Harvard University. {hyhan,hankyang}@seas.harvard.edu.

*Proceedings of the 43rd International Conference on Machine Learning*, Seoul, South Korea. PMLR 306, 2026. Copyright 2026 by the author(s).

$\sum_{t=0}^{T-1} \gamma^t r(s_{t+1}, a_t)$ denotes the discounted return. We use rewards of the form $r(s_{t+1}, a_t)$, instead of $r(s_t, a_t)$, to avoid rewards over a constant initial-state distribution and a terminal action that is sampled but never applied. Following common practice in continuous-control policy optimization, *e.g.,* TRPO / PPO and Linear-Quadratic-Regulator / Linear-Quadratic-Gaussian (Schulman et al., 2015a; 2017; Fazel et al., 2018), we focus on Gaussian policies with diagonal covariance: $\pi_\theta(a \mid s) = \mathcal{N}(\mu_\theta(s), \Sigma), \Sigma = \operatorname{diag}(\sigma^2), \ell = \log \sigma \in \mathbb{R}^m$. We restrict attention to the finite-horizon setting because (i) many widely used benchmarks are episodic (Brockman et al., 2016), and (ii) stability considerations for structured systems (*e.g.,* linear–quadratic control under linear Gaussian policies) can be handled cleanly. We consider the REINFORCE policy gradient estimator

$$\nabla_\theta J(\theta) = \mathbb{E}\left[\widehat{G}_\theta := \left(\sum_{t=0}^{T-1} \nabla_\theta \log \pi_\theta(a_t \mid s_t)\right) R(\tau)\right]. \quad (1)$$

**Noise-to-signal ratio (NSR).** To relate gradient-estimator noise to optimization behavior, in addition to the variance of the gradient estimator, we study the *noise-to-signal ratio* (NSR), defined as the estimator variance (noise) normalized by the squared norm of the true gradient (signal):

$$\mathrm{NSR}(\theta) := \frac{\mathrm{Var}_{\mathrm{Fro}}(\widehat{G}_\theta)}{\|\nabla J(\theta)\|_{\mathrm{F}}^2}, \quad \mathrm{Var}_{\mathrm{Fro}}(\widehat{G}) := \mathbb{E}[\|\widehat{G} - \mathbb{E}[\widehat{G}]\|_{\mathrm{F}}^2].$$

To avoid degeneracy at stationary points, we report NSR only for $\|\nabla J(\theta)\|_{\mathrm{F}} > 0$. NSR is a natural measure of how informative a stochastic gradient is relative to its mean, and is closely related to the *strong growth condition* underlying favorable convergence rates and batch-size rules in stochastic optimization (Bottou et al., 2018; Cevher & Vũ, 2019).

**Contributions.** Classical analyses of stochastic optimization often assume *uniformly bounded* gradient variance or NSR. Our central contribution is to show that, in structured RL problems, the NSR and the variance of the REINFORCE estimator can be highly non-uniform and can increase sharply—potentially grow unbounded—as the policy approaches optimality. Our theoretical results include:

- (Theorem 3) **Closed-form expressions of the NSR for one-step linear systems**, revealing explicit dependence of the NSR on the dynamics, feedback gain, initial-state covariance, and policy covariance.
- (Theorems 6 & 8) **Exact computation of the NSR for multi-step linear systems.** We provide an exact (non-asymptotic) procedure to compute the NSR in finite-horizon linear systems with linear-Gaussian policies via lifted dynamics and Gaussian moment evaluation, together with bounds that expose key scaling factors.
- (Proposition 10) **Exact computation of the NSR for polynomial systems.** This is an extension of the moment-evaluation approach to polynomial dynamics with polynomial feedback under Gaussian exploration.

- (Theorem 11) **Upper bounds for nonlinear systems.** For general nonlinear dynamics and expressive (including neural) Gaussian policies, we provide an upper bound on the variance of the REINFORCE estimator.

Through these characterizations, we show that the variance and the NSR of the REINFORCE estimator scale inversely with policy covariance as the policy becomes more deterministic, and scale proportionally with the initial-state covariance as the initial distribution becomes broader. Moreover, we prove they can grow exponentially with the horizon when the closed-loop dynamics are unstable for linear systems.

These theoretical results enable us to numerically investigate how the NSR evolves along optimization trajectories. We show—for both linear and polynomial systems—that the NSR typically *increases* as the policy approaches optimality. Moreover, gradient descent (GD) can drive the policy toward nearly deterministic solutions, whereas stochastic gradient descent (SGD) often fails to converge as the NSR grows. This behavior is consistent with Corollary 7, which implies that the optimal policy is deterministic (*i.e.,* the policy covariance satisfies $\Sigma \to \mathbf{0}$). Consequently, as the policy nears optimality, the NSR can grow unbounded, providing a concrete mechanism for slowdowns and instability near optima. This highlights an inherent exploration–exploitation tension for REINFORCE-style updates.

We stress the value of *exact* characterizations of REINFORCE variance and the NSR. Although both can be estimated from Monte Carlo rollouts, accurate estimation often requires a prohibitively large number of trajectories; with only a modest rollout budget, sampling noise can yield variance and NSR estimates that misrepresent policy-gradient optimization dynamics, as we illustrate in Appendix B.

**Paper organization.** In Section 2, we characterize the NSR for linear–quadratic systems under Gaussian policies; we then extend the analysis to polynomial systems (Section 3) and nonlinear dynamics (Section 4). Each section presents theory first, then numerical experiments. All the proofs and related work not covered above are deferred to the appendix.

## 2. Linear Systems, Linear Feedback

We start with the linear–quadratic regulator problem with a Gaussian linear policy (LQG). Dynamics and reward are:

$$s_{t+1} = As_t + Ba_t, \quad s_t \in \mathbb{R}^n, \ a_t \in \mathbb{R}^m, \quad (2a)$$

$$r(s, a) = -(s^\top Q_s s + a^\top Q_a a), \quad Q_s, Q_a \succeq 0. \quad (2b)$$

Actions are sampled from a stochastic policy $\pi(a \mid s) \sim \mathcal{N}(Ks, \Sigma)$ with $\Sigma \succ 0$, and the initial state satisfies $s_0 \sim \mathcal{N}(0, \Sigma_0)$ with $\Sigma_0 \succeq 0$. The closed-loop dynamics of (2) can be written as $s_{t+1} = Fs_t + B\epsilon_t$, where $\epsilon_t \sim \mathcal{N}(0, \Sigma)$ and $F := A + BK$. We study the policy gradient and its REINFORCE estimator w.r.t. the gain matrix $K$ and

the policy covariance $\Sigma = \text{diag}(\sigma)^2$, parameterized by the log-standard deviation (log-std) $\ell = \log \sigma \in \mathbb{R}^m$.

**Outline.** In §2.1, we derive closed-form expressions for one-step LQG and analyze scaling with the initial-state and policy covariances. In §2.2, we extend the analysis to multi-step LQG and provide an exact numerical procedure for computing the variance and NSR, along with an upper bound that reveals key scaling factors. In §2.3, we visualize the NSR along optimization trajectories on a double-integrator example and discuss the observed behavior.

### 2.1. NSR of the One-step REINFORCE Estimator

Before stating one-step LQG characterization, we introduce two technical tools used throughout the proof: (i) closed-form score-function gradients for linear Gaussian policies, and (ii) a compact notation for Gaussian moments of quadratic forms (via Wick/Isserlis theorem).

**Lemma 1.** *For a linear Gaussian policy $a_t \sim \mathcal{N}(Ks_t, \Sigma)$ with $\Sigma = \text{diag}(e^{2\ell})$, the score-function gradients are*

$$\nabla_K \log \pi(a_t \mid s_t) \;=\; \Sigma^{-1}(a_t - Ks_t)s_t^\top \;=\; \Sigma^{-1}\varepsilon_t s_t^\top,$$

$$\nabla_\ell \log \pi(a_t \mid s_t) \;=\; \Sigma^{-1}(\varepsilon_t \odot \varepsilon_t) - \mathbf{1},$$

*where $\odot$ is the Hadamard product, $\mathbf{1} \in \mathbb{R}^m$ denotes the all-ones vector, and $\varepsilon_t = a_t - Ks_t \sim \mathcal{N}(0, \Sigma)$.*

**Lemma 2** (Gaussian quadratic-product shorthand)**.** *Let $x \sim \mathcal{N}(0, \Omega)$ with $\Omega \succeq 0$. For any symmetric real matrices $A_1, \ldots, A_k$ of compatible dimension, define the shorthand*

$$\text{IS}_\Omega(A_1, \ldots, A_k) \;:=\; \mathbb{E}\Big[\prod_{i=1}^k (x^\top A_i x)\Big]. \tag{3}$$

*For $k \in \{1, 2, 3\}$, the shorthand expands as*

$$\text{IS}_\Omega(A) = \text{tr}(\Omega A), \tag{4a}$$

$$\text{IS}_\Omega(A, B) = \text{tr}(\Omega A)\,\text{tr}(\Omega B) + 2\,\text{tr}(\Omega A \,\Omega B), \tag{4b}$$

$$\text{IS}_\Omega(A, B, C) = \text{tr}(\Omega A)\,\text{tr}(\Omega B)\,\text{tr}(\Omega C) +$$
$$2\,\text{tr}(\Omega A \Omega B)\,\text{tr}(\Omega C) + 2\,\text{tr}(\Omega A \Omega C)\,\text{tr}(\Omega B) +$$
$$2\,\text{tr}(\Omega A)\,\text{tr}(\Omega B \Omega C) + 8\,\text{tr}(\Omega A \Omega B \Omega C). \tag{4c}$$

*If in addition $\Omega \succ 0$ and $A_i \succeq 0$ for all $i$ with $A_i \neq 0$, then*

$$\text{IS}_\Omega(A_1, \ldots, A_k) > 0. \tag{5}$$

We now present the main result for one-step LQG.

**Theorem 3** (Variance and gradients for one-step LQG)**.** *Let $T = 1$. The expected return is $J(K, \Sigma) = \mathbb{E}[r_0]$ where $r_0 = -(s_1^\top Q_s s_1 + a_0^\top Q_a a_0)$ with $s_1$ and $a_0$ follow (2). The true gradients of $J(K, \Sigma)$ are*

$$\nabla_K J = \mathbb{E}[\widehat{G}_K] = -2\,M_{se}^\top \Sigma_0, \tag{6}$$

$$\nabla_\ell J = \mathbb{E}[\widehat{G}_\ell] = -2\,\text{diag}(\Sigma M_{ee}), \tag{7}$$

*where $M_{ss} := F^\top Q_s F + K^\top Q_a K$, $M_{se} := F^\top Q_s B + K^\top Q_a$ and $M_{ee} := B^\top Q_s B + Q_a$.*

*The Frobenius second moment of $\widehat{G}_K$ decomposes as*

$$\mathbb{E}\Big[\|\widehat{G}_K\|_\text{F}^2\Big] = E_{1,K} + E_{2,K} + E_{3,K} + E_{4,K},$$

*where, with $I_n$ the $n$-dimensional identity matrix and*

$$U := M_{se}M_{se}^\top, \qquad V := M_{se}\Sigma M_{se}^\top,$$

*we have*

$$E_{1,K} = \text{IS}_\Sigma(\Sigma^{-2})\,\text{IS}_{\Sigma_0}(I_n, M_{ss}, M_{ss})$$

$$E_{2,K} = \text{IS}_{\Sigma_0}(I_n)\,\text{IS}_\Sigma(\Sigma^{-2}, M_{ee}, M_{ee})$$

$$E_{3,K} = 2\,\text{IS}_{\Sigma_0}(I_n, M_{ss})\,\text{IS}_\Sigma(\Sigma^{-2}, M_{ee})$$

$$E_{4,K} = 4\Big(2\,\text{IS}_{\Sigma_0}(I_n, U) + \text{IS}_\Sigma(\Sigma^{-2})\,\text{IS}_{\Sigma_0}(I_n, V)\Big).$$

*The Frobenius second moment of $\widehat{G}_\ell$ decomposes as*

$$\mathbb{E}[\|\widehat{G}_\ell\|_2^2] \;=\; E_{1,\ell} + E_{2,\ell} + E_{3,\ell} + E_{4,\ell},$$

*where, defining $S := \Sigma^{1/2} M_{ee} \Sigma^{1/2}$, we have*

$$E_{1,\ell} = 2m\,\text{IS}_{\Sigma_0}(M_{ss}, M_{ss}),$$

$$E_{2,\ell} = (2m+16)\,\text{tr}(S)^2 + (4m+32)\,\text{tr}(S^2) + 24\sum_{i=1}^m S_{ii}^2,$$

$$E_{3,\ell} = 2\,\text{IS}_{\Sigma_0}(M_{ss}) \cdot (2m+8)\,\text{tr}(S),$$

$$E_{4,\ell} = 4\,(2m+8)\,\text{IS}_{\Sigma_0}\Big(M_{se}\,\Sigma\,M_{se}^\top\Big).$$

Theorem 3 characterizes the mean and second moment of the REINFORCE estimator; together, they determine its variance and yield a closed-form NSR.

**Analysis of NSR.** To intuitively study the scaling factors of the NSR, we consider the isotropic setting $\Sigma = \sigma^2 I$ and $\Sigma_0 = \sigma_0^2 I$ for scalars $\sigma, \sigma_0 > 0$. Using Lemma 2 and the fact that $M_{ss}, M_{ee}, U, V \succeq 0$, it is easy to see that $E_{i,K}$ and $E_{i,\ell}$ are nonzero when $A, B \neq 0$. Assuming moreover that $\mathbb{E}[\widehat{G}_K]$ and $\mathbb{E}[\widehat{G}_\ell]$ are nonzero, all terms in the second moments can be expressed using only $(\sigma, \sigma_0)$ (absorbing constants into $\Theta(\cdot)$):

$$E_{1,K} = \Theta\big(\sigma_0^6 \sigma^{-2}\big), \quad E_{2,K} = \Theta(\sigma^2 \sigma_0^2), \quad E_{3,K} = \Theta(\sigma_0^4),$$
$$E_{4,K} = \Theta(\sigma_0^4), \quad E_{1,\ell} = \Theta(\sigma_0^4), \quad E_{2,\ell} = \Theta(\sigma^4),$$
$$E_{3,\ell} = \Theta(\sigma^2 \sigma_0^2), \quad E_{4,\ell} = \Theta(\sigma^2 \sigma_0^2),$$
$$\|\mathbb{E}[\widehat{G}_K]\|_\text{F}^2 = \Theta(\sigma_0^4), \qquad \|\mathbb{E}[\widehat{G}_\ell]\|_\text{F}^2 = \Theta(\sigma^4).$$

Consequently, letting $\widehat{G} = [\widehat{G}_K, \widehat{G}_\ell]$, we have

$$\mathbb{E}[\|\widehat{G}\|_\text{F}^2] = \Theta\left(\frac{\sigma_0^6}{\sigma^2} + \sigma^2\sigma_0^2 + \sigma_0^4 + \sigma^4\right), \quad \|\mathbb{E}[\widehat{G}]\|_\text{F}^2 = \Theta(\sigma_0^4 + \sigma^4)$$

and the noise-to-signal ratio satisfies

$$\mathrm{NSR} := \frac{\mathrm{Var}_{\mathrm{Fro}}(\widehat{G})}{\|\mathbb{E}[\widehat{G}]\|_{\mathrm{F}}^2} = \Theta\left(\frac{\frac{\sigma_0^6}{\sigma^2} + \sigma^2\sigma_0^2 + \sigma_0^4 + \sigma^4}{\sigma_0^4 + \sigma^4}\right) - 1.$$

Moreover, by the AM–GM inequality, we have $0 \leq \sigma^2\sigma_0^2 \leq (\sigma^4 + \sigma_0^4)/2$, hence

$$\frac{\sigma^2\sigma_0^2 + \sigma_0^4 + \sigma^4}{\sigma_0^4 + \sigma^4} = \Theta(1),$$

and therefore

$$\mathrm{NSR} = \Theta\left(\frac{(\sigma_0^6/\sigma^2)}{\sigma_0^4 + \sigma^4} + 1\right) - 1 = \Theta\left(\frac{(\sigma_0/\sigma)^6}{1 + (\sigma_0/\sigma)^4} + 1\right) - 1.$$

Let $\alpha := \sigma_0^2/\sigma^2$. Then $\alpha^3/(1 + \alpha^2)$ is large only when $\alpha$ is large, *i.e.,* when $\sigma_0 \gg \sigma$. In this regime,

$$\mathrm{NSR} = \Theta(\alpha) = \Theta\left(\frac{\sigma_0^2}{\sigma^2}\right). \tag{8}$$

On the other hand, variance is dominated by the $E_{1,K}$ term:

$$\mathrm{Var}_{\mathrm{Fro}}(\widehat{G}) = \Theta\left(\frac{\sigma_0^6}{\sigma^2}\right).$$

In words, when the policy covariance $\Sigma$ is small and the initial-state covariance $\Sigma_0$ is large, the NSR typically grows linearly with $\mathrm{tr}(\Sigma^{-1})\,\mathrm{tr}(\Sigma_0)$, and the variance grows with $\mathrm{tr}(\Sigma^{-1})\,\mathrm{tr}(\Sigma_0)^3$. Thus, both the variance and the NSR can blow up in this regime, suggesting that stochastic-gradient updates may even fail to remain within a bounded neighborhood. To formalize the connection between variance blow-up and training instability, we state Theorem 4.

**Theorem 4** (Escape from a fixed ball). *Let $\{\sigma_k\}_{k\geq 0}$ follow the stochastic-gradient update*

$$\sigma_{k+1} = \sigma_k - \eta(g(\sigma_k) + \varepsilon_{k+1}), \qquad \eta > 0,$$

*where $\eta$ is the learning rate, $g$ is the deterministic gradient, and $\varepsilon_{k+1}$ is a centered gradient-noise term. Suppose there exists a constant $c > 0$ such that, for all $k$ and all $\sigma_k \neq 0$,*

$$\mathbb{E}[\varepsilon_{k+1}^2 \mid \sigma_k] \geq \frac{c}{\sigma_k^2}.$$

*Assume further that $g$ is continuous at 0 and $g(0) = 0$. Then, for every radius $R > 0$, there exists $\delta > 0$ such that*

$$0 < |\sigma_k| \leq \delta \implies \mathbb{P}(|\sigma_{k+1}| > R \mid \sigma_k) > 0.$$

Theorem 4 states that, if the variance grows at least as fast as $1/\sigma^2$ when $\sigma$ is small, then there is a positive probability that the next iterate will escape any fixed ball around 0.

**Example.** Consider the double integrator system

$$A = \begin{bmatrix} 1 & h \\ 0 & 1 \end{bmatrix}, \quad B = \begin{bmatrix} 0 \\ h \end{bmatrix}, \quad Q_s = I_2, \quad Q_a = 0.01, \tag{9}$$

with $h = 0.1$ and policy $K = [-1.0, -3.0]$. Figure 1 visualizes the NSR w.r.t. the initial-state and policy covariances. The upper-left region (small $\Sigma = \sigma^2 I$, large $\Sigma_0 = \sigma_0^2 I$) shows a rapid NSR blow-up, matching the prediction in (8).

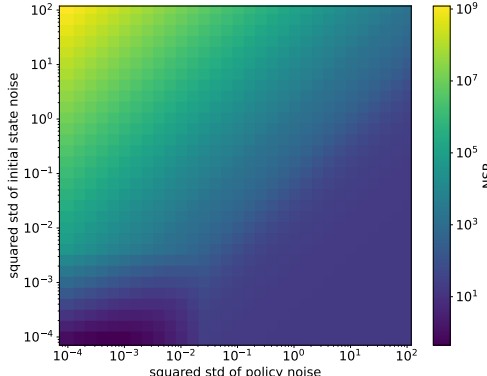

*Figure 1.* NSR of the REINFORCE estimator in one-step LQG with isotropic $\Sigma = \sigma^2 I$ and $\Sigma_0 = \sigma_0^2 I$ for double integrator (9).

## 2.2. NSR of the Multi-step REINFORCE Estimator

We reduce the $T$-step system to a larger one-step system by lifting the dynamics and the policy to block form.

**Theorem 5** (Lifted $T$-step system). *Let the stacked states, actions, and noises be*

$$\bar{s} := \begin{bmatrix} s_0 \\ \vdots \\ s_{T-1} \end{bmatrix} \in \mathbb{R}^{nT}, \quad \bar{s}^+ := \begin{bmatrix} s_1 \\ \vdots \\ s_T \end{bmatrix} \in \mathbb{R}^{nT},$$

$$\bar{a} := \begin{bmatrix} a_0 \\ \vdots \\ a_{T-1} \end{bmatrix} \in \mathbb{R}^{mT}, \quad \bar{\varepsilon} := \begin{bmatrix} \varepsilon_0 \\ \vdots \\ \varepsilon_{T-1} \end{bmatrix} \in \mathbb{R}^{mT}.$$

*There exist matrices*

$$\mathcal{F}_S, \mathcal{F}_S^+ \in \mathbb{R}^{nT \times n}, \quad \mathcal{F}_E, \mathcal{F}_E^+ \in \mathbb{R}^{nT \times mT}$$
$$\mathcal{K}_S \in \mathbb{R}^{mT \times n}, \quad \mathcal{K}_E \in \mathbb{R}^{mT \times mT},$$

*such that*

$$\bar{s} = \mathcal{F}_S s_0 + \mathcal{F}_E \bar{\varepsilon}, \quad \bar{s}^+ = \mathcal{F}_S^+ s_0 + \mathcal{F}_E^+ \bar{\varepsilon}, \quad \bar{a} = \mathcal{K}_S s_0 + \mathcal{K}_E \bar{\varepsilon}.$$

*Let $D_\gamma := \mathrm{diag}(\gamma^0, \gamma^1, \dots, \gamma^{T-1})$ and define the discounted block weights*

$$\mathsf{Q}_{s,\gamma} := D_\gamma \otimes Q_s, \quad \mathsf{Q}_{a,\gamma} := D_\gamma \otimes Q_a, \quad \bar{\Sigma} := I_T \otimes \Sigma.$$

*Then the discounted return admits a single-step quadratic decomposition*

$$R(\tau) = -(x + 2y + z), \tag{10}$$

$$x := s_0^\top \overline{M}_{ss} s_0, \quad y := s_0^\top \overline{M}_{se} \bar{\varepsilon}, \quad z := \bar{\varepsilon}^\top \overline{M}_{ee} \bar{\varepsilon}, \tag{11}$$

*where the lifted blocks are*

$$\overline{M}_{ss} := \mathcal{F}_S^{+\top} \mathsf{Q}_{s,\gamma} \mathcal{F}_S^+ + \mathcal{K}_S^\top \mathsf{Q}_{a,\gamma} \mathcal{K}_S, \tag{12}$$

$$\overline{M}_{se} := \mathcal{F}_S^{+\top} \mathsf{Q}_{s,\gamma} \mathcal{F}_E^+ + \mathcal{K}_S^\top \mathsf{Q}_{a,\gamma} \mathcal{K}_E, \tag{13}$$

$$\overline{M}_{ee} := \mathcal{F}_E^{+\top} \mathsf{Q}_{s,\gamma} \mathcal{F}_E^+ + \mathcal{K}_E^\top \mathsf{Q}_{a,\gamma} \mathcal{K}_E. \tag{14}$$

With the lifted one-step system, we can study the variance and gradients of the REINFORCE estimator.

**Theorem 6** (Variance and gradients of multi-step LQG). *Consider the lifted multi-step LQG in Theorem 5, the following statements hold.*

**(i) Second moments.** *With the lifted notation $R(\tau) = -(x + 2y + z)$ and the shorthand $\mathrm{IS}_\Omega(\cdot)$ of Lemma 2, we have the decompositions of the second moments:*

$$\mathbb{E}\left[\|\widehat{G}_K\|_F^2\right] = \sum_{t,p=0}^{T-1} \sum_{i=1}^{6} E_{i,tp}^K, \quad \mathbb{E}\left[\|\widehat{G}_\ell\|_2^2\right] = \sum_{t,p=0}^{T-1} \sum_{i=1}^{4} E_{i,tp}^\ell$$

*where the terms $E_{1,tp}^K, \ldots, E_{6,tp}^K$ are given by the six-term decompositions (38)–(49) in the proof, and $E_{1,tp}^\ell, \ldots, E_{4,tp}^\ell$ are given by (51), (52), (54), (65).*

**(ii) Mean gradients.** *Let $P_t := \mathrm{Cov}(s_t)$ satisfy the covariance recursion*

$$P_{t+1} = F P_t F^\top + B \Sigma B^\top, \quad P_0 = \Sigma_0$$

*and let $\{\Lambda_t\}_{t=1}^T$ satisfy the backward recursion*

$$\Lambda_T = \frac{1}{\gamma} Q_s, \quad \Lambda_t = \frac{1}{\gamma} Q_s + K^\top Q_a K + \gamma F^\top \Lambda_{t+1} F.$$

*Then the gradients of the objective $J(K, \Sigma)$ are*

$$\nabla_K J = -2 \sum_{t=0}^{T-1} \gamma^t \Big( Q_a K P_t + \gamma B^\top \Lambda_{t+1} F P_t \Big), \quad (15)$$

$$\nabla_\ell J = -2\mathrm{diag}(\Sigma \sum_{t=0}^{T-1} \gamma^t \Big( Q_a + \gamma B^\top \Lambda_{t+1} B \Big)). \quad (16)$$

**Corollary 7** (Stationary points of $J(K, \Sigma)$). *Under the assumptions of Theorem 6, if $Q_a \succ 0$, then $\nabla_\ell J = 0$ if and only if $\Sigma = \mathbf{0}$.*

Theorem 6 provides an exact numerical procedure for computing the REINFORCE estimator's variance (and NSR) in any finite-horizon LQG system. The resulting expressions are considerably more involved than in the one-step case, making it difficult to directly extract clean scaling laws. We therefore analyze a simple upper bound to clarify how the NSR scales with the horizon, the policy and initial-state covariances, and closed-loop stability.

**Theorem 8** (Multi-step LQG variance bound). *Under the lifted multi-step LQG in Theorem 5, we have*

$$\mathrm{Var}_{\mathrm{Fro}}(\widehat{G}_K) \leq 2 \underbrace{\mathbb{E}[\|\bar{\Sigma}^{-1}\bar{\varepsilon}\|_2^2 \|\mathcal{F}_S s_0\|_2^2 R(\tau)^2]}_{(\star)}$$

$$+ 2 \underbrace{\mathbb{E}[\|\bar{\Sigma}^{-1}\bar{\varepsilon}\|_2^2 \|\mathcal{F}_E \bar{\varepsilon}\|_2^2 R(\tau)^2]}_{(\dagger)}, \quad (17)$$

$$\mathrm{Var}_{\mathrm{Fro}}(\widehat{G}_\ell) \leq T \mathbb{E}[(2\|\bar{\Sigma}^{-1/2}\bar{\varepsilon}\|_2^4 + 2mT) R(\tau)^2], \quad (18)$$

*and the expectation on the right-hand side admits an explicit Gaussian-moment expansion given in (69), (70) and (71).*

**Analysis of NSR.** Same as the analysis in the one-step case, $\overline{M}_{ee}, \overline{M}_{ss} \succeq 0, \overline{M}_{se}$ are nonzero matrices, so that every term in $(\star)$, $(\dagger)$ and $\mathbb{E}\left[\|\widehat{G}_\ell\|_2^2\right]$ are nonzero. With $\Sigma = \sigma^2 I$ and $\Sigma_0 = \sigma_0^2 I$, we can write these terms as

$$(\star) = \Theta(\frac{\sigma_0^6}{\sigma^2} + \sigma_0^4 + \sigma_0^2\sigma^2), \quad (\dagger) = \Theta(\sigma^4 + \sigma_0^4 + \sigma_0^2\sigma^2),$$

$$\mathbb{E}\left[\|\widehat{G}_\ell\|_2^2\right] = \Theta(\sigma^4 + \sigma_0^4 + \sigma_0^2\sigma^2).$$

From Theorem 6 we know $\|\mathbb{E}[\widehat{G}_K]\|_F^2, \|\mathbb{E}[\widehat{G}_\ell]\|_2^2 = \Theta((\sigma^2 + \sigma_0^2)^2)$ if they are nonzero, thus

$$\frac{\mathrm{Var}_{\mathrm{Fro}}(\widehat{G}_K) + \mathrm{Var}_{\mathrm{Fro}}(\widehat{G}_\ell)}{\|\mathbb{E}[\widehat{G}_K]\|_F^2 + \|\mathbb{E}[\widehat{G}_\ell]\|_F^2} = O\left(\frac{\frac{\sigma_0^6}{\sigma^2} + \sigma_0^4 + \sigma_0^2\sigma^2 + \sigma^4}{(\sigma_0^2 + \sigma^2)^2}\right).$$

Again, define $\alpha := \sigma_0^2/\sigma^2$, we have

$$\frac{\frac{\sigma_0^6}{\sigma^2} + \sigma_0^4 + \sigma_0^2\sigma^2 + \sigma^4}{(\sigma_0^2 + \sigma^2)^2} = \frac{\sigma_0^4 + \sigma^4}{\sigma^2(\sigma_0^2 + \sigma^2)} = \frac{\alpha^2 + 1}{\alpha + 1}.$$

Therefore, if $\sigma_0 \gg \sigma$, we have $\alpha \to \infty$ and the ratio scales as $O(\mathrm{tr}(\Sigma_0)\,\mathrm{tr}(\Sigma^{-1}))$, same as in the one-step case but stated here as an upper bound.

Theorem 8 further indicates the variance bound depends on the spectral norms of the lifted state maps $\mathcal{F}_S$ and $\mathcal{F}_E$, which exponentially grow with the horizon $T$ if the closed-loop system is unstable.

**Theorem 9** (Spectral norm of lifted state map). *The lifted state map $\mathcal{F}_S$ satisfies*

$$\max_{0 \leq t \leq T-1} \|F^t\|_2^2 \leq \|\mathcal{F}_S\|_2^2 \leq \sum_{t=0}^{T-1} \|F^t\|_2^2. \quad (19)$$

*Thus, $\|\mathcal{F}_S\|_2^2$ is $O(1)$ if $\rho(F) < 1$, and $O(\rho(F)^{2T})$ if $\rho(F) > 1$ (with $\rho(\cdot)$ denoting the spectral radius).*

Figure 3 illustrates how the *true* variance $\mathrm{Var}_{\mathrm{Fro}}(\widehat{G})$ (computed exactly from Theorem 6) scales with the horizon $T$ across four systems: it remains bounded when $\rho(F) < 1$, grows roughly polynomially when $\rho(F) = 1$, and grows exponentially when $\rho(F) > 1$. This verifies the prediction of Theorems 8 and 9. Thus, in long-horizon tasks, closed-loop instability can dramatically worsen the NSR.

### 2.3. Experiments

Figure 2 visualizes the NSR landscape for a double-integrator system and overlays the optimization trajectories of three methods: GD, SGD, and Adam (Kingma & Ba, 2014). The policy has three parameters: a 2D gain matrix $K$ and a 1D log-std $\ell$. We observe that the NSR increases as the iterates approach the optimum. GD converges smoothly, whereas SGD and Adam exhibit oscillations once the NSR becomes large. This behavior is consistent with Corollary 7,

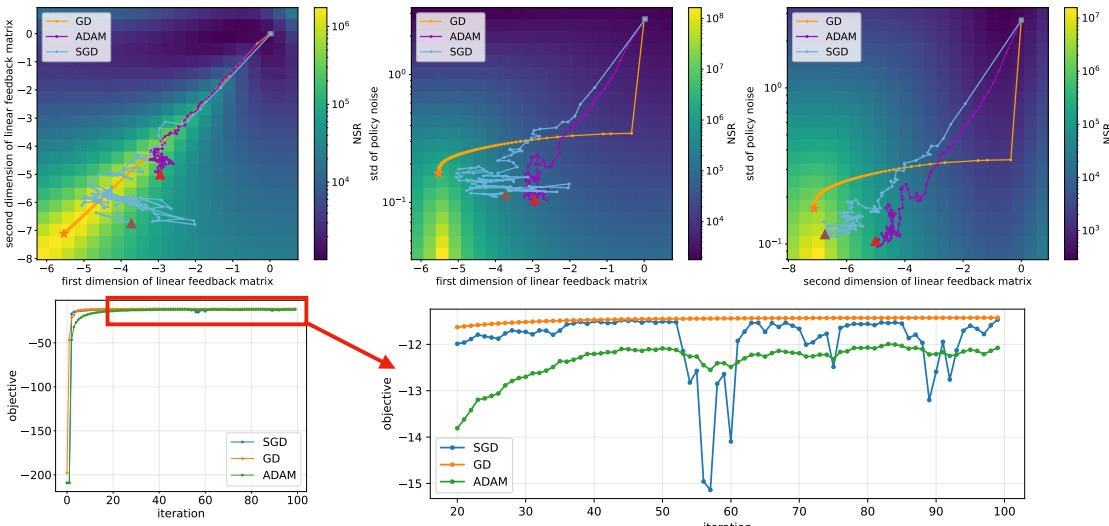

Figure 2. NSR and objective along optimization trajectories on a double-integrator system with $T = 30$. Top: optimizer trajectories overlaid on the NSR landscape. Bottom: learning curves (objective vs. iteration). The three optimizers (GD, SGD, Adam) start from the same initial policy (square), move toward the optimal policy (star), and terminate at triangles. The NSR increases markedly as the policy approaches optimality, consistent with our theory. As the NSR grows, SGD and Adam exhibit oscillations in the learning curves.

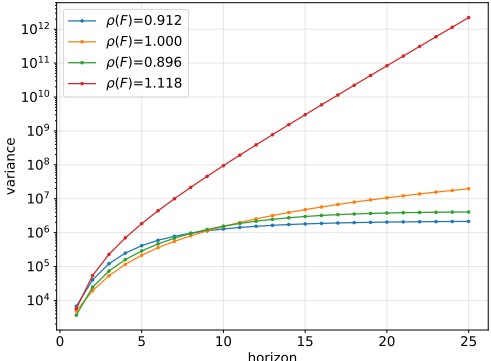

Figure 3. Variance growth as a function of the horizon $T$ for four 2D systems (1D control) with varying $\rho(F)$. We fix the initial-state covariance at $\Sigma_0 = I_2$ and the policy covariance at $\Sigma = 0.1I_1$.

which implies that the optimal policy is deterministic (*i.e.,* $\Sigma \to 0$), so the NSR diverges near optimality; empirically, SGD and Adam are stable early on but oscillate near the optimum in both parameter space and objective value.

## 3. Polynomial Systems, Polynomial Feedback

We then consider polynomial dynamics with polynomial reward, and a Gaussian policy with polynomial mean

$$s_{t+1} = P(s_t, a_t), \quad \pi(a \mid s) = \mathcal{N}(\Phi(s)^\top \theta, \Sigma), \quad (20)$$

where $\Phi(s) \in \mathbb{R}^{d \times m}$ is a matrix of polynomial features. Equivalently, $a_t = \Phi(s_t)^\top \theta + \varepsilon_t$ with $\varepsilon_t \sim \mathcal{N}(0, \Sigma)$. The initial state satisfies $s_0 \sim \mathcal{N}(0, \Sigma_0)$.

**Proposition 10** (Polynomial form of REINFORCE and exact Gaussian-moment evaluation). *The following statements*

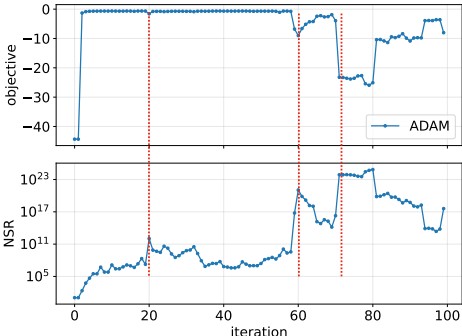

Figure 4. NSR (bottom) and objective (top) during training on the quadratic system. As the policy oscillates near the optimum, a single inaccurate gradient step triggers policy collapse.

*hold for the polynomial system* (20)*:*

*(i) Polynomial estimator. For each $t \le T$, the random variables $s_t$ and $a_t$ are vector-valued polynomials in $\xi := (s_0, \varepsilon_0, \ldots, \varepsilon_{T-1})$. Consequently, $\widehat{G}_\theta$ and $\widehat{G}_\ell$ are polynomials in $\xi$.*

*(ii) Exact moments. $\mathbb{E}[\widehat{G}_\theta], \mathbb{E}[\widehat{G}_\ell]$, $\mathbb{E}[\|\widehat{G}_\theta\|_F^2]$, $\mathbb{E}[\|\widehat{G}_\ell\|_F^2]$ are exactly computable from $\mathrm{Cov}(\xi)$ via Isserlis theorem.*

Proposition 10 shows the NSR can in principle be computed symbolically for polynomial systems.

However, the worst-case complexity grows rapidly with the dimension, degree, and horizon. Let $P$ have degree $d_P$, $\Phi$ have degree $d_\Phi$, and $r$ have degree $d_r$. Then $\deg s_t \approx d_P^t d_\Phi^t$ and $\deg a_t \approx d_P^t d_\Phi^{t+1}$. Thus, the REINFORCE estimator involves Gaussian moments up to order $d_P^{2T} d_\Phi^{2T+1} d_r$. In the worst case, the number of monomial terms can be as large

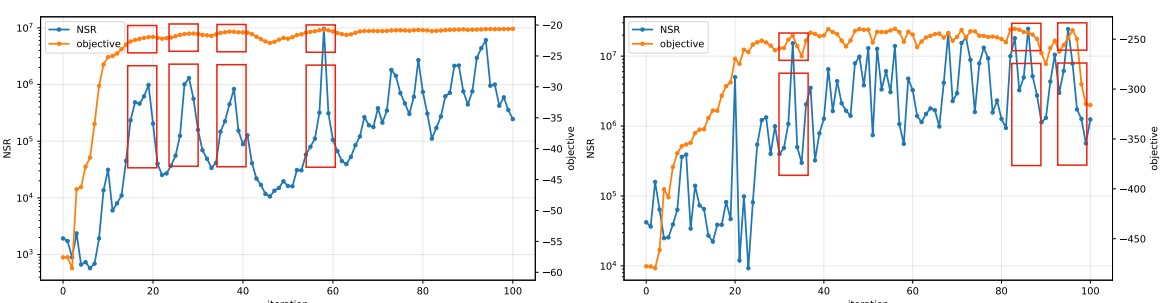

*Figure 5.* NSR and objective along optimization trajectories on polynomial systems. (a): quadratic system; (b): cubic system. Top: optimizer trajectories and NSR landscape; Bottom: learning curves. The NSR increases when approaching the optimal policy, as in linear systems. GD consistently approaches the optimum while SGD and Adam may get stuck in high-NSR regions.

*Figure 6.* Trajectory of NSR and objective for LQR with MLP policy (left) and Pendulum with MLP policy (right). Both the NSR and the objective are calculated through Monte Carlo evaluation. The NSR increases when approaching the optimal policy, and the red square shows the synchronized fluctuations in objective and NSR.

as $\binom{n+mT+d_P^{2T} d_\Phi^{2T+1} d_r}{d_P^{2T} d_\Phi^{2T+1} d_r}$, leading to exponential dependence in $T$ and a combinatorial blow-up in $n$ and $m$.

**Experiments**. We implement code to compute the objective, gradient, and variance for two polynomial systems:

*Quadratic system:* $s_{t+1} = s_t + h(-s_t^2 + a_t)$ with reward $r(s, a) = -s^2 - a^2$; where $h = 0.05$ is the step size. The policy is parameterized as $a_t = \theta_1 s_t + \theta_2 s_t^2 + \varepsilon_t$.

*Cubic system:* $s_{t+1} = s_t + h(-s_t^3 + a_t)$ with reward $r(s, a) = -1.25s^6 - a^2$; where $h = 0.05$. The policy is parameterized as $a_t = \theta_1 s_t + \theta_2 s_t^3 + \varepsilon_t$. Note the optimal policy can be proved to be an odd function because of the symmetry of dynamics and initial state distribution.

The cubic example is motivated by an infinite-horizon continuous-time analogue where the optimal feedback is polynomial. For $\dot{s} = -s^3 + a$ and the selected cost $(-r(s, a))$, the Hamilton-Jacobi-Bellman equation admits a polynomial value function $V(s) = 0.25s^4$ and the stationary optimal policy $a^\star(s) = -0.5\, V'(s) = -0.5s^3$.

Due to limited computational resources, we set the horizon $T = 6$ for the quadratic system and $T = 3$ for the cubic system. Initial state covariance is set as $\Sigma_0 = 0.1$.

We plot the NSR heatmap for both systems in Figure 5. As in the LQG case, we examine the trajectories of three optimizers (GD, SGD, Adam) in a 3D policy-parameter space (two polynomial coefficients and one log-std). The trajectories show the same trend: the NSR increases as the policy approaches optimality. This provides evidence that the non-uniform variance / NSR phenomenon exists across different systems.

With smaller batch sizes and larger learning rates, we observe policy collapse: even near the optimum, a single noisy gradient step can drive the log-standard deviation downward, causing the NSR to blow up and the policy to collapse (Figure 4). We further verify that this collapse is caused by statistical error by comparing against two baselines: a larger batch size and the exact gradient. The larger-batch setting still collapses, but only after a longer time, whereas the exact-gradient setting does not collapse at all (Figure 7).

## 4. Nonlinear Systems, Generic Feedback

We now consider generic nonlinear dynamics and policies:

$$s_{t+1} = f(s_t, a_t), \quad a_t = \mu_\theta(s_t) + \varepsilon_t, \quad \varepsilon_t \sim \mathcal{N}(0, \Sigma). \quad (21)$$

In this nonlinear setting, $\widehat{G} = g(\xi)$ is a nonlinear function of Gaussian noise, so NSR depends on expectations $\mathbb{E}[g(\xi)]$ and $\mathbb{E}[g(\xi)^2]$, which are generally intractable Gaussian integrals. This is the same issue that arises in nonlinear estimation where even computing moments of a nonlinear transformation of a Gaussian is difficult (Julier & Uhlmann,

2004). Therefore, we focus on providing upper bounds.

**Theorem 11** (Upper bounds for generic nonlinear systems). *The REINFORCE estimators for* (21) *are*

$$\widehat{G}_\theta(\tau) := R(\tau) \sum_{t=0}^{T-1} J_\theta(s_t)^\top \Sigma^{-1} \varepsilon_t, \quad (22)$$

$$\widehat{G}_\ell(\tau) := R(\tau) \sum_{t=0}^{T-1} \left( \Sigma^{-1}(\varepsilon_t \odot \varepsilon_t) - \mathbf{1} \right), \quad (23)$$

*Assume the following conditional moments are finite:*

$$\mathbb{E}\big[R(\tau)^4 \,\big|\, s_0\big] < \infty, \quad \mathbb{E}\big[\|J_\theta(s_t)\|_F^4 \,\big|\, s_0\big] < \infty \quad \forall t, s_0.$$

*Then the following bounds hold:*

*(i) Mean-parameter gradient.*

$$\mathbb{E}\left[\|\widehat{G}_\theta(\tau)\|_2^2 \,\Big|\, s_0\right] \leq T\, \|\Sigma^{-1}\|_2^2 \, \sqrt{\mathbb{E}[R(\tau)^4 \,|\, s_0]}$$

$$\sqrt{(\mathrm{tr}\,\Sigma)^2 + 2\,\mathrm{tr}(\Sigma^2)} \sum_{t=0}^{T-1} \sqrt{\mathbb{E}[\|J_\theta(s_t)\|_F^4 \,|\, s_0]}. \quad (24)$$

*(ii) Log-std gradient.*

$$\mathbb{E}\left[\|\widehat{G}_\ell(\tau)\|_2^2 \,\Big|\, s_0\right] \leq C\sqrt{\mathbb{E}[R(\tau)^4 \,|\, s_0]}. \quad (25)$$

*where $C$ is a constant depends on the system.*

Theorem 11 upper-bounds the variance of the REINFORCE estimator in generic nonlinear systems in terms of the return's fourth moment and the Jacobian of the policy mean. The bounds are conditioned on the initial state $s_0$, which can be removed by taking expectation w.r.t. $s_0$. This reveals a few important factors:

**Policy covariance $\Sigma$.** The mean-parameter gradient variance scales with $\|\Sigma^{-1}\|_2^2 \sqrt{(\mathrm{tr}\,\Sigma)^2 + 2\,\mathrm{tr}(\Sigma^2)}$, which scales as $O(\sigma^{-2})$ when $\Sigma = \sigma^2 I$. This agrees with the intuition that smaller exploration noise leads to higher variance.

**Jacobian $J_\theta(s_t)$.** More sensitive policies (larger Jacobians) lead to higher gradient-estimation variance. In particular, optimal controllers in physical systems are often "bang-bang" (*e.g.,* pendulum and acrobot), corresponding to sharp state-to-action changes and potentially large Jacobians (Åström & Furuta, 2000; Han & Yang, 2024).

**Experiments.** We plot the NSR and objective along Adam optimization trajectories for LQR with a multi-layer perceptron (MLP) policy and for the inverted pendulum problem with an MLP policy in Figure 6. We make two observations: (i) the NSR increases as the optimizer approaches the optimum; and (ii) near the optimum, the objective and NSR exhibit coupled fluctuations (highlighted in red). This suggests that once the optimizer enters a neighborhood of the optimum, the NSR increases and thus leads to unreliable gradient estimates that cause objective drops. We also

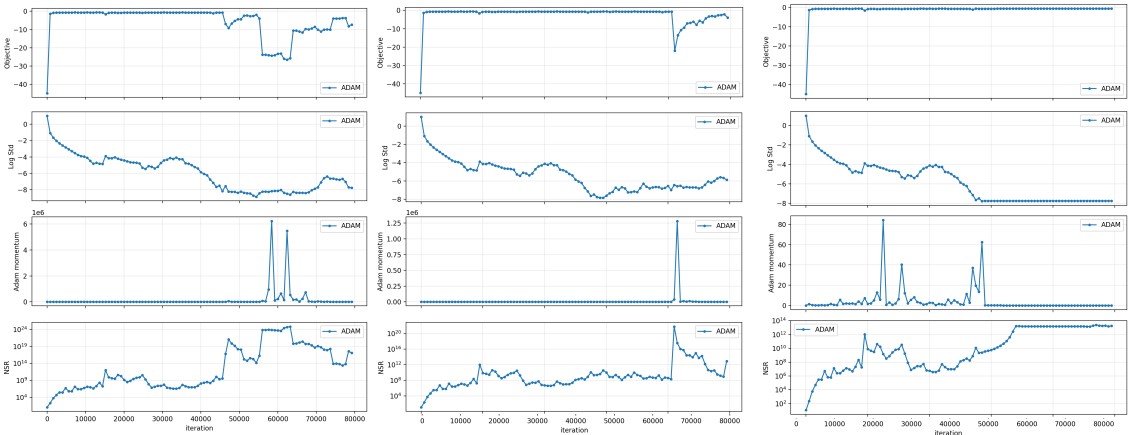

*Figure 7.* Detailed ablation study of policy collapse on the quadratic system. Left: original policy-collapse curves. Middle: with a larger batch size, the policy still collapses but takes longer. Right: with the exact gradient, the policy does not collapse. From top to bottom, the rows show the objective, the log-standard deviation of the policy covariance, Adam's momentum, and the NSR. These results confirm that the collapse is caused by statistical error.

observe similar coupled fluctuations in the MuJoCo Hopper and HalfCheetah environments (Figure 10 in Appendix B), suggesting that high-NSR regimes are associated with instability in policy optimization across different systems.

## 5. Discussion and Conclusions

We studied the non-uniformity of the noise-to-signal ratio (NSR) of the REINFORCE estimator. For finite-horizon linear systems with linear-Gaussian policies and polynomial systems with Gaussian policies and polynomial feedback, we characterized NSR exactly, either in closed form or via moment-evaluation algorithms. For nonlinear dynamics with expressive Gaussian policies, we derived a general variance upper bound. These results show that NSR varies sharply across policy parameters and typically increases as policies approach optimality, sometimes diverging. This provides a concrete explanation for the oscillations and occasional policy collapse observed in our experiments.

**Baselines and actor–critic methods.** Baselines can substantially reduce NSR, but they do not automatically remove the small-covariance pathology. For a baseline estimator, the return target is replaced by the residual $\delta_t := G_t - b_t(s_t)$ where $b_t(s_t)$ is the baseline. In finite-horizon LQG, an exact quadratic baseline can cancel the state-only component of the return when $\Sigma = \sigma^2 I$ and $\sigma \to 0$, thus the resulting estimator becomes $O(1)$ rather than $O(\sigma^{-1})$. However, this cancellation is fragile. If the baseline has approximation error that remains $O(1)$ as $\sigma \to 0$, then the residual remains $O(1)$ and the same small-covariance blow-up can persist. This suggests that approximate learned baselines and critics may reduce NSR in practice, but avoiding the blow-up generally requires sufficiently accurate value-function approximation. A similar interpretation applies to actor–critic estimators.

**Entropy regularization.** Entropy regularization provides a complementary stabilization mechanism by discouraging premature collapse of the policy covariance. For the entropy-regularized objective

$$J_\tau(K, \Sigma) := J(K, \Sigma) + \tau \mathbb{E}\left[\sum_{t=0}^{T-1} \gamma^t H(\pi(\cdot \mid s_t))\right], \tau > 0,$$

the Gaussian entropy term is explicit in the log-standard deviation. In the diagonal covariance case, the stationary covariance satisfies

$$\Sigma_{ii}^\star = \frac{\tau \sum_{t=0}^{T-1} \gamma^t}{2M_{ii}(K^\star)}, \quad M(K) := \sum_{t=0}^{T-1} \gamma^t (Q_a + \gamma B^\top \Lambda_{t+1} B).$$

Thus entropy regularization yields a strictly positive target covariance. Plugging this covariance into our exact variance/NSR expressions can be used to select an entropy coefficient or covariance floor corresponding to a desired NSR level. However, if stochastic updates drive the policy covariance far below its entropy-regularized target, the policy can still enter a high-NSR region and become unstable.

Our findings suggest several promising directions for future work. First, developing variance-reduction techniques tailored to high-NSR regimes could improve optimization stability. Second, extending our analysis beyond Gaussian policies to other policy distributions, such as uniform or $\beta$-distributions, would help assess the generality of our conclusions. Third, studying the interaction between NSR and exploration mechanisms, including adaptive covariance and entropy control, may lead to more robust reinforcement-learning algorithms. Finally, while our experiments show that large NSR can induce instability, and our SGD analysis demonstrates that noisy updates have a positive probability of escaping an arbitrarily large ball, a formal characterization of the relationship between the NSR landscape and instability in policy optimization remains an open question.

## Impact Statement

This paper presents work whose goal is to advance the field of Machine Learning by characterizing the noise-to-signal ratio of policy-gradient estimators and its implications for optimization stability. The primary expected impact is improved understanding and design of more reliable reinforcement-learning algorithms. While such improvements could enable more capable automated decision-making in real-world systems, they also carry general risks associated with broader deployment of RL. We do not foresee societal or ethical consequences beyond those already well established for reinforcement learning research.

## Conflict of Interest Disclosure

The authors declare no financial conflicts of interest.

## Acknowledgements

We thank Jincheng Mei and Na Li for helpful related work discussions; Shucheng Kang for proof-reading the paper, and members of the Harvard Computational Robotics Group for various discussions throughout the project. This project was supported in part by the Office of Naval Research grant N00014-25-1-2322.

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

## A. Related Work

**Policy gradient methods.** In addition to the algorithms mentioned in Section 1, representative policy-gradient variants include Natural Policy Gradient (NPG) (Kakade, 2001), Deep Deterministic Policy Gradient (DDPG) (Lillicrap, 2015), and Soft Actor–Critic (SAC) (Haarnoja et al., 2018). NPG preconditions the gradient using the Fisher information matrix, but it still estimates gradients using REINFORCE-style likelihood-ratio estimators; therefore, it falls within the scope of our analysis. In contrast, DDPG and SAC are actor–critic methods and use critic-based gradient surrogates.

For policy-gradient methods on LQR/LQG, (Fazel et al., 2018) study infinite-horizon LQR with fixed Gaussian exploration covariance from an optimization perspective, showing that the objective is gradient dominated and that policy gradient converges to the global optimum. (Furieri et al., 2020) extends these guarantees to a class of finite-horizon LQR problems. (Malik et al., 2020; Mohammadi et al., 2021) further consider stochastic estimation settings and establish convergence and sample complexity results. (Yang et al., 2019) analyzes the convergence of actor–critic algorithms on LQR. For partially observed LQG, (Tang et al., 2023) characterizes the optimization landscape, including connectivity properties of stable controllers and the structure of stationary points. Beyond LQR/LQG, (Mei et al., 2020; Agarwal et al., 2021) study global convergence in tabular settings and derive error bounds in terms of distribution shift/transfer.

More closely related to our focus on estimator variability, (Zhao et al., 2011) shows that parameter-based exploration can yield lower-variance policy-gradient estimates than the vanilla REINFORCE estimator; both admit $O(\sigma^{-2})$ upper and lower bounds, where $\sigma$ denotes the standard deviation of a Gaussian policy. Their analysis assumes bounded states and rewards and does not characterize an exact NSR expression. (Preiss et al., 2019) derives variance upper bounds for finite-horizon LQG and studies contributing factors empirically, but focuses on stable systems and does not make the scaling with the initial-state distribution or policy covariance explicit. We complement these works by providing exact finite-horizon variance expressions and by extending the analysis to polynomial systems and to generic nonlinear systems.

**Convergence dynamics.** (Ilyas et al., 2018) empirically observes that the cosine similarity between the estimated and true gradients can deteriorate as optimization progresses; we interpret this phenomenon through our NSR analysis. (Dohare et al., 2023) further reports a "policy collapse" phenomenon and relates it to large Adam momentum. A complementary line of work studies how baseline choices affect optimization beyond their impact on variance: (Chung et al., 2021) shows that even when the variance is held fixed, different baseline choices can lead to markedly different convergence behavior, and (Mei et al., 2022) establishes convergence results in bandit settings while showing that the variance near the optimal policy can be unbounded for any choice of baseline.

## B. Additional Experiments

**Comparison between REINFORCE and GPOMDP (Baxter & Bartlett, 2001).** In the main text, we focused on the vanilla REINFORCE estimator. We also conducted experiments with the GPOMDP estimator, which replaces the full trajectory return with reward-to-go terms using the Markov property. GPOMDP can have lower variance than REINFORCE (Papini et al., 2022). The same moment-evaluation framework can be directly applied to these cases by changing the return $R(\tau)$ to running return $G_t$ which is still a quadratic function of the underlying Gaussian variables. We find that the NSR landscape is qualitatively similar to the REINFORCE case, with the same trend of increasing NSR near optimality. (Figure 8).

**Monte Carlo estimation of NSR.** We empirically investigate the accuracy of Monte Carlo estimation of the variance and the NSR. Our results show that Monte Carlo estimates fail to accurately capture the trends of both the variance and the NSR along optimization trajectories, even when using a large number of rollouts (Figure 9). This highlights the value of exact characterizations of the REINFORCE variance and the NSR, which can further enable rigorous analyses of policy-gradient optimization dynamics.

**Experiment details.** For all experiments below, we set the initial exploration covariance to $\Sigma_0 = I$. In the multi-step LQG experiments with double-integrator dynamics, we use Adam with learning rate $10^{-3}$, SGD with learning rate $10^{-4}$, and batch size $2^{10}$. In the polynomial-system experiments, we use the same learning rates for Adam and SGD, with batch size $2^{12}$. For the policy-collapse experiments on the quadratic system, we use Adam with learning rate $10^{-2}$ and batch size $2^{6}$. For the experiments with nonlinear dynamics and MLP policies, we use two-layer MLPs with 64 hidden units and ReLU activations, Adam with learning rate $10^{-2}$, and batch size $2^{12}$. The NSR and variance for the nonlinear dynamics experiments are estimated via Monte Carlo with 1 minute limit per iteration.

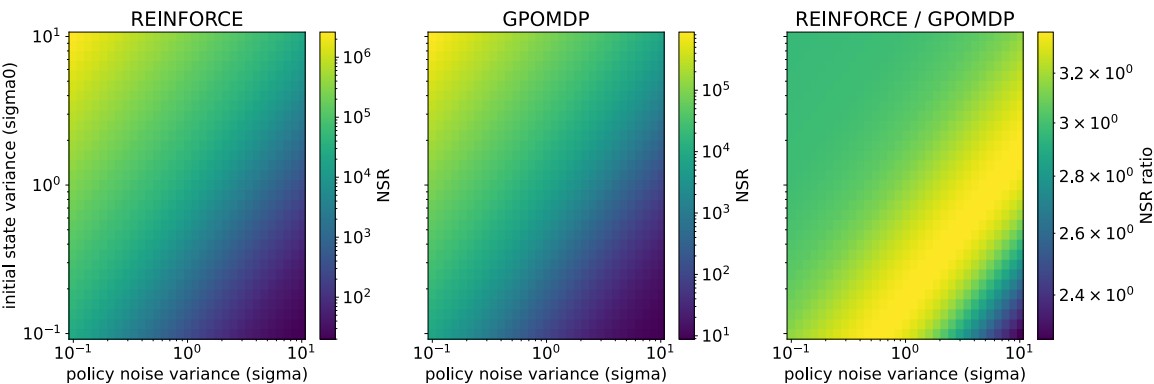

*Figure 8.* NSR landscape for REINFORCE, GPOMDP and their ratio. The NSR landscape for GPOMDP is qualitatively similar to that of REINFORCE, with the same trend of increasing NSR near optimality.

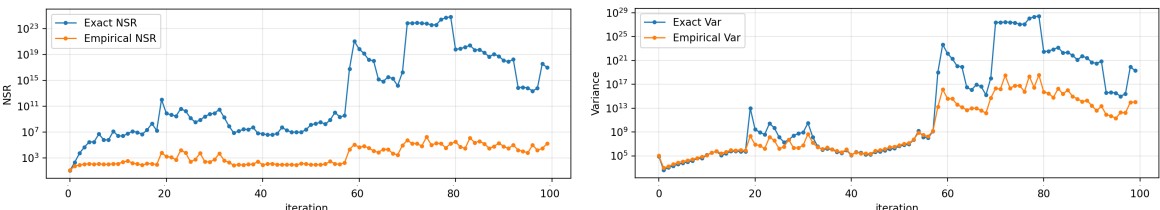

*Figure 9.* Monte Carlo estimation of the NSR. Orange line: empirical Monte Carlo estimate with 10 minutes limit each iteration. Blue line: exact NSR. Even with a large number of rollouts, the empirical Monte Carlo estimate fails to accurately capture the trends of the variance and the NSR along optimization trajectories.

**Additional MuJoCo experiments.** In addition to the pendulum/LQR with MLP policy and polynomial system experiments in the main text, we also conducted experiments on the MuJoCo Hopper and HalfCheetah environments with MLP policies. We train these policies using PPO. We observe coupled fluctuations between NSR and objective value, suggesting that high-NSR regimes are associated with instability in policy optimization (Figure 10).

## C. Proof of Lemma 1

*Proof.* The Gaussian density is

$$\pi(a \mid s) = (2\pi)^{-m/2} \det(\Sigma)^{-1/2} \exp\left(-\frac{1}{2}(a - Ks)^\top \Sigma^{-1}(a - Ks)\right).$$

Let $\varepsilon := a - Ks$. Taking logarithms yields

$$\log \pi(a \mid s) = -\frac{1}{2}\varepsilon^\top \Sigma^{-1}\varepsilon - \frac{1}{2}\log \det(\Sigma) + c,$$

where $c$ is a constant independent of $K, \ell$.

**Gradient w.r.t. $K$.** Using differentials, $d\varepsilon = -dKs$, hence

$$d(\varepsilon^\top \Sigma^{-1}\varepsilon) = 2\,\varepsilon^\top \Sigma^{-1}d\varepsilon = -2\,\varepsilon^\top \Sigma^{-1}(dK)s.$$

Rewrite the scalar as a trace:

$$-2\,\varepsilon^\top \Sigma^{-1}(dK)s = -2\operatorname{tr}\left(s\,\varepsilon^\top \Sigma^{-1}\,dK\right).$$

Therefore,

$$d\log \pi(a \mid s) = -\frac{1}{2}\,d(\varepsilon^\top \Sigma^{-1}\varepsilon) = \operatorname{tr}\left(s\,\varepsilon^\top \Sigma^{-1}\,dK\right),$$

which implies (by the identity $df = \operatorname{tr}((\nabla_K f)^\top dK)$)

$$\nabla_K \log \pi(a \mid s) = \Sigma^{-1}\varepsilon\,s^\top = \Sigma^{-1}(a - Ks)\,s^\top.$$

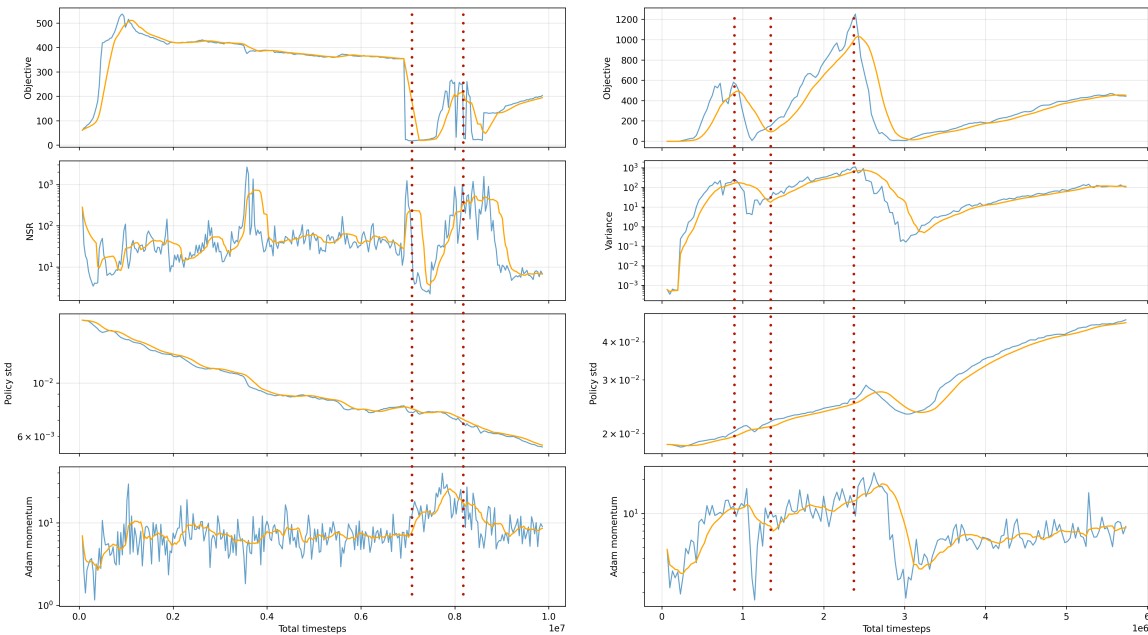

**Figure 10.** MuJoCo experiments that show instability caused by large NSR. Left: Hopper-v4 environment; Right: HalfCheetah-v4 environment. From top to bottom we show the objective, the NSR, the log standard deviation of the policy covariance, and Adam's momentum. Red dashed lines show the synchronized fluctuations in objective and NSR.

**Gradient w.r.t.** $\ell = \log \sigma$. Since $\Sigma = \mathrm{diag}(\sigma)^2$ with $\sigma_i = e^{\ell_i}$, we have $\Sigma_{ii} = \sigma_i^2 = e^{2\ell_i}$ and $(\Sigma^{-1})_{ii} = \sigma_i^{-2}$. The log-density can be written componentwise as

$$\log \pi(a \mid s) = -\frac{1}{2}\sum_{i=1}^{m}\left(\frac{\varepsilon_i^2}{\sigma_i^2} + \log \sigma_i^2\right) + c = -\frac{1}{2}\sum_{i=1}^{m}\left(\frac{\varepsilon_i^2}{\sigma_i^2} + 2\ell_i\right) + c.$$

Differentiate each coordinate:

$$\frac{\partial}{\partial \ell_i}\log \pi(a \mid s) = -\frac{1}{2}\left(\frac{\partial}{\partial \ell_i}\frac{\varepsilon_i^2}{\sigma_i^2} + 2\right).$$

Because $\sigma_i^2 = e^{2\ell_i}$, we have $\sigma_i^{-2} = e^{-2\ell_i}$ and $\frac{\partial}{\partial \ell_i}\sigma_i^{-2} = -2\sigma_i^{-2}$, hence

$$\frac{\partial}{\partial \ell_i}\frac{\varepsilon_i^2}{\sigma_i^2} = \varepsilon_i^2\frac{\partial}{\partial \ell_i}\sigma_i^{-2} = -2\frac{\varepsilon_i^2}{\sigma_i^2}.$$

Substituting gives

$$\frac{\partial}{\partial \ell_i}\log \pi(a \mid s) = -\frac{1}{2}\left(-2\frac{\varepsilon_i^2}{\sigma_i^2} + 2\right) = \frac{\varepsilon_i^2}{\sigma_i^2} - 1.$$

Stacking the coordinates yields

$$\nabla_\ell \log \pi(a \mid s) = \Sigma^{-1}(\varepsilon \odot \varepsilon) - \mathbf{1}.$$

Finally, since $a \mid s \sim \mathcal{N}(Ks, \Sigma)$, the centered variable $\varepsilon = a - Ks$ satisfies $\varepsilon \sim \mathcal{N}(0, \Sigma)$. $\qquad\square$

## D. Proof of Lemma 2

*Proof.* The expansions in (4) follow from Isserlis (Wick's) theorem for centered Gaussians applied to quadratic forms. This is also proved in (Magnus et al., 1978).

To prove (5), for $A \succeq 0$, we have $x^\top A x = 0$ if and only if $x \in \ker(A)$. Since each $A_i \succeq 0$ and $A_i \neq 0$, its kernel $\ker(A_i)$ is a proper linear subspace of $\mathbb{R}^n$. Because $\Omega \succ 0$, the Gaussian $x \sim \mathcal{N}(0, \Omega)$ has a density that is strictly positive on all of

$\mathbb{R}^n$. In particular, any proper linear subspace has Lebesgue measure zero, hence

$$\mathbb{P}(x \in \ker(A_i)) = 0 \qquad \text{for each } i.$$

Since a finite union of measure-zero sets has measure zero, we have

$$\mathbb{P}\left(x \in \bigcup_{i=1}^{k} \ker(A_i)\right) = 0, \quad \text{so} \quad \mathbb{P}\left(x \notin \bigcup_{i=1}^{k} \ker(A_i)\right) = 1.$$

Therefore, with probability 1 we have $x \notin \ker(A_i)$ for every $i$, which implies $x^\top A_i x > 0$ for every $i$, and hence

$$\prod_{i=1}^{k}(x^\top A_i x) > 0 \qquad \text{almost surely.}$$

Since the integrand is nonnegative and strictly positive almost surely, the expectation must be strictly positive:

$$\mathbb{E}\left[\prod_{i=1}^{k}(x^\top A_i x)\right] > 0.$$

$\square$

# E. Proof of Theorem 3

We begin with a lemma that computes expectations of certain polynomial functions of Gaussian random variables. While such identities are broadly covered by Isserlis' theorem (Wick's formula), those results mainly provide expressions for generic even-order moments. In our analysis, we require more structured, problem-specific forms. Magnus (Magnus et al., 1978) derives formulas for expectations of products of quadratic forms of the type $\mathbb{E}\left[\prod_{i=1}^{N}(u^\top A_i u)\right]$. Our setting involves a non-homogeneous polynomial, hence we state the following lemma.

**Lemma 12.** *For $\xi \sim \mathcal{N}(0, I_m)$, define*

$$w(\xi) := \sum_{i=1}^{m}(\xi_i^2 - 1)^2.$$

*Then for any $u \in \mathbb{R}^m$ and any $M \in \mathbb{S}^m$,*

$$\mathbb{E}[w(\xi)] = 2m, \tag{26}$$

$$\mathbb{E}[w(\xi)\,(\xi^\top u)^2] = (2m + 8)\,\|u\|^2, \tag{27}$$

$$\mathbb{E}[w(\xi)\,(\xi^\top M\xi)] = (2m + 8)\,\operatorname{tr} M, \tag{28}$$

$$\mathbb{E}[w(\xi)\,(\xi^\top M\xi)^2] = (4m + 32)\,\operatorname{tr}(M^2) + (2m + 16)\,(\operatorname{tr} M)^2 + 24\sum_{i=1}^{m} M_{ii}^2. \tag{29}$$

*Proof.* To prove (26), use $\mathbb{E}[\xi^8] = 105$, $\mathbb{E}[\xi^6] = 15$, $\mathbb{E}[\xi^4] = 3$, $\mathbb{E}[\xi^2] = 1$:

$$\mathbb{E}[w(\xi)] = \sum_{i=1}^{m} \mathbb{E}[(\xi_i^2 - 1)^2] = \sum_{i=1}^{m}(3 - 2 + 1) = 2m.$$

For (27), expand and note that cross terms with $u_i u_j$ $(i \neq j)$ vanish by odd-moment cancellation, so only $u_i^2$ terms remain:

$$\mathbb{E}[w(\xi)(\xi^\top u)^2] = \sum_{i=1}^{m}\sum_{j=1}^{m} \mathbb{E}[(\xi_i^2 - 1)^2 \xi_j^2]\, u_j^2.$$

Compute

$$\mathbb{E}[(\xi_1^2 - 1)^2 \xi_1^2] = \mathbb{E}[\xi_1^6] - 2\mathbb{E}[\xi_1^4] + \mathbb{E}[\xi_1^2] = 15 - 6 + 1 = 10, \qquad \mathbb{E}[(\xi_i^2 - 1)^2 \xi_1^2] = 2 \ (i \neq 1),$$

hence $\mathbb{E}[w(\xi)(\xi^\top u)^2] = \|u\|^2(10 + 2(m-1)) = (2m+8)\|u\|^2$.

For (28), similarly only diagonal entries contribute:

$$\mathbb{E}[w(\xi)(\xi^\top M\xi)] = \sum_{i=1}^m \sum_{j=1}^m \mathbb{E}[(\xi_i^2 - 1)^2 \xi_j^2] \, M_{jj} = (2(m-1) + 10)\operatorname{tr} M = (2m+8)\operatorname{tr} M.$$

For (29), expand the quadratic form as

$$(z^\top M z)^2 = \sum_{i,j,k,l=1}^m M_{ij} M_{kl} \, z_i z_j z_k z_l.$$

After cancelling the odd terms, only two index patterns contribute: (i) $i = j$ and $k = l$ (diagonal–diagonal terms), and (ii) $\{i,j\} = \{k,l\}$ with $i \neq j$ (off-diagonal pairings), yielding

$$(z^\top M z)^2 = \sum_{i,k} M_{ii} M_{kk} \, z_i^2 z_k^2 \;+\; 2 \sum_{i<j} M_{ij}^2 z_i^2 z_j^2 \;+\; \text{(terms with vanishing expectation)}.$$

We then evaluate $\mathbb{E}[w(z) z_i^2 z_k^2]$ by splitting into the cases $i = k$ and $i \neq k$ and using the one-dimensional moments $\mathbb{E}[z^2] = 1, \mathbb{E}[z^4] = 3, \mathbb{E}[z^6] = 15, \mathbb{E}[z^8] = 105$, which gives

$$\mathbb{E}[w(z)\, z_i^4] = \mathbb{E}[(z_i^2 - 1)^2 z_i^4] + (m-1)\mathbb{E}[(z_1^2 - 1)^2]\mathbb{E}[z_i^4] = 78 + 6(m-1),$$

$$\mathbb{E}[w(z)\, z_i^2 z_k^2] = \mathbb{E}[(z_i^2-1)^2 z_i^2]\mathbb{E}[z_k^2] + \mathbb{E}[(z_k^2-1)^2 z_k^2]\mathbb{E}[z_i^2] + (m-2)\mathbb{E}[(z_1^2-1)^2]\mathbb{E}[z_i^2]\mathbb{E}[z_k^2] = 20 + 2(m-2), \quad (i \neq k),$$

and similarly

$$\mathbb{E}[w(z)\, z_i^2 z_j^2] = 2m + 16 \qquad (i \neq j).$$

Substituting these values into the diagonal–diagonal and off-diagonal contributions and collecting terms yields

$$\mathbb{E}[w(z)\, (z^\top M z)^2] = (4m + 32)\operatorname{tr}(M^2) + (2m + 16)(\operatorname{tr} M)^2 + 24 \sum_{i=1}^m M_{ii}^2,$$

concluding the proof. $\qquad\qquad\qquad\qquad\qquad\qquad\qquad\qquad\qquad\qquad\qquad\qquad\qquad\qquad\qquad\square$

Next we prove Theorem 3.

*Proof.* **Proof of $K$ part.** Using

$$s_1 = F s_0 + B\varepsilon_0, \quad a_0 = K s_0 + \varepsilon_0, \quad r_0 = -(s_1^\top Q_s s_1 + a_0^\top Q_a a_0),$$

we can expand $r_0$ as

$$r_0 = -(x + 2y + z),$$

where

$$x := s_0^\top M_{ss} s_0, \quad y := s_0^\top M_{se} \varepsilon_0, \quad z := \varepsilon_0^\top M_{ee} \varepsilon_0, \tag{30}$$

and

$$M_{ss} := F^\top Q_s F + K^\top Q_a K, \quad M_{se} := F^\top Q_s B + K^\top Q_a, \quad M_{ee} := B^\top Q_s B + Q_a.$$

Note $M_{ss}, M_{ee}$ are symmetric and positive semidefinite. The REINFORCE estimator is

$$\widehat{G}_K = \nabla_K \log \pi(a_0 \mid s_0)\, r_0 = \Sigma^{-1} \varepsilon_0 s_0^\top \, (-(x + 2y + z)).$$

Since $x$ depends only on $s_0$ and $z$ depends only on $\varepsilon_0$, and $s_0 \perp\!\!\!\perp \varepsilon_0$ with $\mathbb{E}[\varepsilon_0] = 0$, only the $y$ term remains in expectation:

$$\mathbb{E}[\widehat{G}_K] = \mathbb{E}\big[\Sigma^{-1}\varepsilon_0 s_0^\top (-2y)\big] = -2\,\Sigma^{-1}\mathbb{E}[\varepsilon_0\varepsilon_0^\top]\, M_{se}^\top \mathbb{E}[s_0 s_0^\top] = -2\, M_{se}^\top \Sigma_0 = -2\,(Q_a K + B^\top Q_s F)\Sigma_0.$$

For the second moment, using $\|AB\|_{\mathrm{F}}^2 = \mathrm{tr}(B^\top A^\top AB)$,

$$\|\widehat{G}_K\|_{\mathrm{F}}^2 = (\varepsilon_0^\top \Sigma^{-2} \varepsilon_0)\,(s_0^\top s_0)\,(x + 2y + z)^2.$$

By independence and odd-moment cancellation,

$$\mathbb{E}[\|\widehat{G}_K\|_{\mathrm{F}}^2] = \mathbb{E}\Big[(\varepsilon_0^\top \Sigma^{-2} \varepsilon_0)\,(s_0^\top s_0)\,(x^2 + z^2 + 2xz + 4y^2)\Big] = E_{1,K} + E_{2,K} + E_{3,K} + E_{4,K},$$

with

$$
\begin{aligned}
E_{1,K} &:= \mathbb{E}[\varepsilon_0^\top \Sigma^{-2} \varepsilon_0]\,\mathbb{E}[(s_0^\top s_0)\,x^2], \\
E_{2,K} &:= \mathbb{E}[s_0^\top s_0]\,\mathbb{E}[(\varepsilon_0^\top \Sigma^{-2} \varepsilon_0)\,z^2], \\
E_{3,K} &:= 2\,\mathbb{E}[(s_0^\top s_0)\,x]\,\mathbb{E}[(\varepsilon_0^\top \Sigma^{-2} \varepsilon_0)\,z], \\
E_{4,K} &:= 4\,\mathbb{E}\big[(s_0^\top s_0)\,\mathbb{E}\big[(\varepsilon_0^\top \Sigma^{-2} \varepsilon_0)\,y^2 \,\big|\, s_0\big]\big].
\end{aligned}
$$

We rewrite each term using Lemma 2. First,

$$\mathbb{E}[\varepsilon_0^\top \Sigma^{-2} \varepsilon_0] = \mathrm{IS}_\Sigma(\Sigma^{-2}), \qquad \mathbb{E}[s_0^\top s_0] = \mathrm{IS}_{\Sigma_0}(I_n).$$

Also,

$$\mathbb{E}[(s_0^\top s_0)x^2] = \mathrm{IS}_{\Sigma_0}(I_n, M_{ss}, M_{ss}), \qquad \mathbb{E}[(s_0^\top s_0)x] = \mathrm{IS}_{\Sigma_0}(I_n, M_{ss}),$$

and since $z = \varepsilon_0^\top M_{ee} \varepsilon_0$,

$$\mathbb{E}[(\varepsilon_0^\top \Sigma^{-2} \varepsilon_0)\,z] = \mathrm{IS}_\Sigma(\Sigma^{-2}, M_{ee}), \qquad \mathbb{E}[(\varepsilon_0^\top \Sigma^{-2} \varepsilon_0)\,z^2] = \mathrm{IS}_\Sigma(\Sigma^{-2}, M_{ee}, M_{ee}).$$

It remains to compute $E_{4,K}$. Since $y = s_0^\top M_{se} \varepsilon_0$,

$$y^2 = \varepsilon_0^\top X(s_0) \varepsilon_0, \qquad X(s_0) := M_{se}^\top s_0 s_0^\top M_{se}.$$

Conditioning on $s_0$ and applying Lemma 2 with $k = 2$,

$$\mathbb{E}_{\varepsilon_0}\big[(\varepsilon_0^\top \Sigma^{-2} \varepsilon_0)\,y^2 \,\big|\, s_0\big] = \mathrm{IS}_\Sigma(\Sigma^{-2}, X(s_0)) = \mathrm{tr}(\Sigma\Sigma^{-2})\,\mathrm{tr}(\Sigma X(s_0)) + 2\,\mathrm{tr}(\Sigma\Sigma^{-2}\Sigma X(s_0)).$$

Using $\mathrm{tr}(\Sigma\Sigma^{-2}) = \mathrm{IS}_\Sigma(\Sigma^{-2})$ and $\Sigma\Sigma^{-2}\Sigma = I$, and defining

$$U := M_{se} M_{se}^\top, \qquad V := M_{se} \Sigma M_{se}^\top,$$

we have

$$\mathrm{tr}\,(X(s_0)) = s_0^\top U s_0, \qquad \mathrm{tr}\,(\Sigma X(s_0)) = s_0^\top V s_0.$$

Therefore,

$$\mathbb{E}_{\varepsilon_0}\big[(\varepsilon_0^\top \Sigma^{-2} \varepsilon_0)\,y^2 \,\big|\, s_0\big] = \mathrm{IS}_\Sigma(\Sigma^{-2})\,(s_0^\top V s_0) + 2\,(s_0^\top U s_0).$$

Multiplying by $(s_0^\top s_0) = (s_0^\top I_n s_0)$ and taking expectation over $s_0$ gives

$$E_{4,K} = 4\Big(\mathrm{IS}_\Sigma(\Sigma^{-2})\,\mathrm{IS}_{\Sigma_0}(I_n, V) + 2\,\mathrm{IS}_{\Sigma_0}(I_n, U)\Big).$$

**Proof of log-std part.** Write $\varepsilon_0 = \Sigma^{1/2}\xi$ with $\xi \sim \mathcal{N}(0, I_m)$ and define

$$q := \Sigma^{-1}(\varepsilon_0 \odot \varepsilon_0) - \mathbf{1} = \xi \odot \xi - \mathbf{1}, \qquad S := \Sigma^{1/2} M_{ee} \Sigma^{1/2}.$$

Since $s_0$ and $\varepsilon_0$ are independent,

$$\mathbb{E}[\widehat{G}_\ell] = \mathbb{E}[q\,r_0] = -\,\mathbb{E}[q\,z], \qquad z = \varepsilon_0^\top M_{ee} \varepsilon_0 = \xi^\top S \xi.$$

For the $i$-th component,

$$[\mathbb{E}(q\,z)]_i = \sum_{j,k} S_{jk}\,\mathbb{E}[(\xi_i^2 - 1)\,\xi_j \xi_k].$$

By independence,

$$\mathbb{E}[(\xi_i^2 - 1)\,\xi_j \xi_k] = \begin{cases} \mathbb{E}[\xi_i^4 - \xi_i^2] = 3 - 1 = 2, & j = k = i, \\ 0, & \text{otherwise,} \end{cases}$$

so $[\mathbb{E}(q\,z)]_i = 2\,S_{ii}$ and hence

$$\mathbb{E}[\widehat{G}_\ell] = -2\,\mathrm{diag}(S) = -2\,\mathrm{diag}(\Sigma^{1/2} M_{ee} \Sigma^{1/2}) = -2\,\mathrm{diag}(\Sigma M_{ee}),$$

where the last equality uses that $\Sigma$ is diagonal.

For the second moment, $\widehat{G}_\ell = q\,r_0$ and $r_0^2 = (x + 2y + z)^2$. Using independence and odd-moment cancellation,

$$\mathbb{E}[\|\widehat{G}_\ell\|_2^2] = \mathbb{E}[\|q\|_2^2]\,\mathbb{E}[x^2] + \mathbb{E}[\|q\|_2^2 z^2] + 2\,\mathbb{E}[x]\,\mathbb{E}[\|q\|_2^2 z] + 4\,\mathbb{E}[\|q\|_2^2 y^2].$$

The $x$-moments follow from Lemma 2:

$$x = s_0^\top M_{ss} s_0, \qquad \mathbb{E}[x] = \mathrm{IS}_{\Sigma_0}(M_{ss}), \qquad \mathbb{E}[x^2] = \mathrm{IS}_{\Sigma_0}(M_{ss}, M_{ss}).$$

For the $\varepsilon_0$-moments, $z = \xi^\top S \xi$ and $\|q\|_2^2 = \sum_{i=1}^m (\xi_i^2 - 1)^2 = w(\xi)$, so Lemma 12 gives

$$\mathbb{E}[\|q\|_2^2] = 2m, \qquad \mathbb{E}[\|q\|_2^2 z] = (2m + 8)\,\mathrm{tr}(S),$$

$$\mathbb{E}[\|q\|_2^2 z^2] = (2m + 16)\,\mathrm{tr}(S)^2 + (4m + 32)\,\mathrm{tr}(S^2) + 24 \sum_{i=1}^m S_{ii}^2.$$

Finally, for $y = s_0^\top M_{se} \varepsilon_0 = s_0^\top M_{se} \Sigma^{1/2} \xi$, define

$$u(s_0) := \Sigma^{1/2} M_{se}^\top s_0 \in \mathbb{R}^m, \qquad y = u(s_0)^\top \xi.$$

Then $y^2 = (u(s_0)^\top \xi)^2$ and Lemma 12 implies

$$\begin{aligned} \mathbb{E}[\|q\|_2^2 y^2] &= \mathbb{E}_{s_0}\big[\mathbb{E}_\xi\big[w(\xi)\,(\xi^\top u(s_0))^2 \mid s_0\big]\big] = \mathbb{E}_{s_0}\big[(2m + 8)\,\|u(s_0)\|^2\big] \\ &= (2m + 8)\,\mathbb{E}_{s_0}\big[s_0^\top (M_{se} \Sigma M_{se}^\top) s_0\big] = (2m + 8)\,\mathrm{tr}(\Sigma_0\,M_{se} \Sigma M_{se}^\top) \\ &= (2m + 8)\,\mathrm{IS}_{\Sigma_0}(M_{se} \Sigma M_{se}^\top), \end{aligned}$$

concluding the proof. $\qquad\square$

## F. Proof of Theorem 4

*Proof.* Fix $R > 0$. Define

$$m(\sigma) := \sigma - \eta g(\sigma).$$

Since $g$ is continuous at 0 and $g(0) = 0$, we have

$$m(\sigma) \to 0 \qquad \text{as } \sigma \to 0.$$

Hence, for all sufficiently small $\delta > 0$,

$$B_\delta := \sup_{|\sigma| \le \delta} |m(\sigma)| < \infty,$$

and moreover

$$B_\delta \to 0 \qquad \text{as } \delta \downarrow 0.$$

Choose $\delta > 0$ sufficiently small so that

$$\frac{c}{\delta^2} > \left(\frac{R + B_\delta}{\eta}\right)^2.$$

This is possible because $c/\delta^2 \to \infty$ as $\delta \downarrow 0$, while $B_\delta \to 0$.

Now fix $k$ and suppose that

$$0 < |\sigma_k| \leq \delta.$$

By the assumed lower bound on the conditional second moment,

$$\mathbb{E}[\varepsilon_{k+1}^2 \mid \sigma_k] \geq \frac{c}{\sigma_k^2} \geq \frac{c}{\delta^2} > \left( \frac{R + B_\delta}{\eta} \right)^2.$$

Therefore, it cannot be true that

$$|\varepsilon_{k+1}| \leq \frac{R + B_\delta}{\eta} \qquad \text{with conditional probability 1 given } \sigma_k,$$

because otherwise we would have

$$\mathbb{E}[\varepsilon_{k+1}^2 \mid \sigma_k] \leq \left( \frac{R + B_\delta}{\eta} \right)^2,$$

contradicting the preceding strict inequality. Hence

$$\mathbb{P}\left( |\varepsilon_{k+1}| > \frac{R + B_\delta}{\eta} \;\middle|\; \sigma_k \right) > 0.$$

On the event

$$\left\{ |\varepsilon_{k+1}| > \frac{R + B_\delta}{\eta} \right\},$$

we have, using the update rule,

$$|\sigma_{k+1}| = |m(\sigma_k) - \eta \varepsilon_{k+1}| \geq \eta|\varepsilon_{k+1}| - |m(\sigma_k)|.$$

Since $|\sigma_k| \leq \delta$, we have $|m(\sigma_k)| \leq B_\delta$. Therefore,

$$|\sigma_{k+1}| > (R + B_\delta) - B_\delta = R.$$

Thus,

$$\mathbb{P}(|\sigma_{k+1}| > R \mid \sigma_k) > 0.$$

This proves the claim. $\qquad\square$

## G. Proof of Theorem 5

*Proof.* From

$$\bar{s} = \mathcal{F}_S s_0 + \mathcal{F}_E \bar{\varepsilon}, \qquad \bar{s}^+ = \mathcal{F}_S^+ s_0 + \mathcal{F}_E^+ \bar{\varepsilon}, \qquad \bar{a} = \mathcal{K}_S s_0 + \mathcal{K}_E \bar{\varepsilon},$$

and using $s_{t+1} = F s_t + B \varepsilon_t$, we write out the explicit forms:

**State maps.**

$$\mathcal{F}_S = \begin{bmatrix} I \\ F \\ F^2 \\ \vdots \\ F^{T-1} \end{bmatrix} \in \mathbb{R}^{nT \times n}, \qquad \mathcal{F}_S^+ = \begin{bmatrix} F \\ F^2 \\ \vdots \\ F^T \end{bmatrix} \in \mathbb{R}^{nT \times n}, \tag{31}$$

$$\mathcal{F}_E = \begin{bmatrix} 0 & 0 & \cdots & 0 & 0 \\ B & 0 & \cdots & 0 & 0 \\ FB & B & \ddots & \vdots & 0 \\ \vdots & \ddots & \ddots & 0 & 0 \\ F^{T-2}B & \cdots & FB & B & 0 \end{bmatrix} \in \mathbb{R}^{nT \times mT}, \qquad \mathcal{F}_E^+ = \begin{bmatrix} B & 0 & \cdots & 0 \\ FB & B & \ddots & \vdots \\ \vdots & \ddots & \ddots & 0 \\ F^{T-1}B & \cdots & FB & B \end{bmatrix} \in \mathbb{R}^{nT \times mT}.$$

Equivalently, for $t, i \in \{1, \ldots, T\}$,

$$[\mathcal{F}_E]_{t,i} = \begin{cases} F^{t-1-i}B, & i \le t-1, \\ 0, & i \ge t, \end{cases} \qquad [\mathcal{F}_E^+]_{t,i} = \begin{cases} F^{t-i}B, & i \le t, \\ 0, & i > t. \end{cases}$$

**Action maps.**

$$\mathcal{K}_S = \begin{bmatrix} K \\ KF \\ \vdots \\ KF^{T-1} \end{bmatrix} \in \mathbb{R}^{mT \times n}, \qquad \mathcal{K}_E = \begin{bmatrix} I_m & 0 & \cdots & 0 & 0 \\ KB & I_m & \ddots & \vdots & 0 \\ KFB & KB & \ddots & 0 & 0 \\ \vdots & \ddots & \ddots & I_m & 0 \\ KF^{T-2}B & \cdots & KFB & KB & I_m \end{bmatrix} \in \mathbb{R}^{mT \times mT}.$$

Equivalently, for $t, i \in \{1, \ldots, T\}$,

$$[\mathcal{K}_E]_{t,i} = \begin{cases} KF^{t-1-i}B, & i \le t-1, \\ I_m, & i = t, \\ 0, & i > t. \end{cases}$$

$\square$

# H. Proof of Theorem 6

*Proof.* Recall the policy gradient estimators

$$\widehat{G}_K = \sum_{t=0}^{T-1} \Sigma^{-1} \varepsilon_t s_t^\top R(\tau), \qquad \widehat{G}_\ell = \sum_{t=0}^{T-1} \left( \Sigma^{-1}(\varepsilon_t \odot \varepsilon_t) - \mathbf{1} \right) R(\tau),$$

and the stacked noise $\bar{\varepsilon} := (\varepsilon_0^\top, \ldots, \varepsilon_{T-1}^\top)^\top \sim \mathcal{N}(0, \bar{\Sigma})$ with $\bar{\Sigma} := I_T \otimes \Sigma$.

**Second moment expansions.** Using $\|AB\|_{\mathrm{F}}^2 = \mathrm{tr}(B^\top A^\top AB)$,

$$\|\widehat{G}_K\|_{\mathrm{F}}^2 = \left\| \sum_{t=0}^{T-1} \Sigma^{-1} \varepsilon_t s_t^\top \right\|_{\mathrm{F}}^2 R(\tau)^2 = \sum_{t,p=0}^{T-1} (\varepsilon_t^\top \Sigma^{-2} \varepsilon_p)(s_t^\top s_p) R(\tau)^2.$$

For the log-std part, define

$$g_t := \Sigma^{-1}(\varepsilon_t \odot \varepsilon_t) - \mathbf{1} \in \mathbb{R}^m.$$

Then

$$\|\widehat{G}_\ell\|_2^2 = \left\| \sum_{t=0}^{T-1} g_t \right\|_2^2 R(\tau)^2 = \sum_{t,p=0}^{T-1} (g_t^\top g_p) R(\tau)^2.$$

Using the lifted decomposition $R(\tau) = -(x + 2y + z)$ with

$$x = s_0^\top \bar{M}_{ss} s_0, \qquad y = s_0^\top \bar{M}_{se} \bar{\varepsilon}, \qquad z = \bar{\varepsilon}^\top \bar{M}_{ee} \bar{\varepsilon},$$

we have

$$R(\tau)^2 = (x + 2y + z)^2 = x^2 + z^2 + 2xz + 4y^2 + 4xy + 4yz. \tag{32}$$

Therefore

$$\mathbb{E}[\|\widehat{G}_K\|_{\mathrm{F}}^2] = \sum_{t,p=0}^{T-1} \mathbb{E}\left[ (\varepsilon_t^\top \Sigma^{-2} \varepsilon_p)(s_t^\top s_p)(x^2 + z^2 + 2xz + 4y^2 + 4xy + 4yz) \right]. \tag{33}$$

**Even-only reduction for the $\ell$-part.** Note that $g_t^\top g_p$ is an *even* polynomial in $\bar{\varepsilon}$, while $xy$ and $yz$ have *odd* total degree in $\bar{\varepsilon}$; hence

$$\mathbb{E}_{\bar{\varepsilon}}[(g_t^\top g_p)(4xy + 4yz) \mid s_0] = 0,$$

and therefore

$$\mathbb{E}[\|\widehat{G}_\ell\|_2^2] = \sum_{t,p=0}^{T-1} \mathbb{E}\Big[(g_t^\top g_p)(x^2 + z^2 + 2xz + 4y^2)\Big]. \tag{34}$$

**Multi-step gradient.** Consider the original dynamics (2) (not the lifted one). Since $\mathbb{E}[s_0] = 0$ and $\mathbb{E}[\varepsilon_t] = 0$, it follows by induction that $\mathbb{E}[s_t] = 0$ for all $t$. Let $P_t := \mathrm{Cov}(s_t)$. Using independence of $s_t$ and $\varepsilon_t$,

$$P_{t+1} = \mathrm{Cov}(Fs_t + B\varepsilon_t) = FP_tF^\top + B\Sigma B^\top, \qquad P_0 = \Sigma_0.$$

Therefore

$$\mathbb{E}[s_{t+1}^\top Q_s\, s_{t+1}] = \mathrm{tr}(Q_s\, P_{t+1}).$$

Also $a_t = Ks_t + \varepsilon_t$ is zero-mean with

$$\mathrm{Cov}(a_t) = KP_tK^\top + \Sigma \quad \Rightarrow \quad \mathbb{E}[a_t^\top Q_a\, a_t] = \mathrm{tr}\,(Q_a(KP_tK^\top + \Sigma)) = \mathrm{tr}(Q_aKP_tK^\top) + \mathrm{tr}(Q_a\Sigma).$$

Taking expectations in $r(s_{t+1}, a_t) = -(s_{t+1}^\top Q_s\, s_{t+1} + a_t^\top Q_a\, a_t)$ and summing over $t = 0, \ldots, T-1$ with discount $\gamma^t$ yields

$$J_T(K, \Sigma) = -\sum_{t=0}^{T-1} \gamma^t \left(\mathrm{tr}(Q_s\, P_{t+1}) + \mathrm{tr}(Q_a\, KP_tK^\top) + \mathrm{tr}(Q_a\Sigma)\right). \tag{35}$$

To take the gradient of (35), we use a Lagrangian to enforce the covariance dynamics $P_{t+1} = FP_tF^\top + B\Sigma B^\top$. Introduce multipliers $\{\Lambda_{t+1}\}_{t=0}^{T-1}$ and set

$$\mathcal{L} = -\sum_{t=0}^{T-1} \gamma^t \left(\mathrm{tr}(Q_s\, P_{t+1}) + \mathrm{tr}(Q_a\, KP_tK^\top) + \mathrm{tr}(Q_a\Sigma)\right) + \sum_{t=0}^{T-1} \gamma^{t+1}\, \mathrm{tr}\left(\Lambda_{t+1}\,(P_{t+1} - FP_tF^\top - B\Sigma B^\top)\right).$$

(i) Stationarity w.r.t. $P_t$. For $t = 1, \ldots, T-1$, the matrix $P_t$ appears in the terms at indices $t-1$ and $t$:

$$\partial_{P_t}\mathcal{L}: \quad -\gamma^{t-1}Q_s - \gamma^t K^\top Q_aK + \gamma^t\Lambda_t - \gamma^{t+1}F^\top\Lambda_{t+1}F = 0.$$

Dividing by $\gamma^t$ yields the recursion $\Lambda_t = \dfrac{1}{\gamma}Q_s + K^\top Q_aK + \gamma F^\top\Lambda_{t+1}F$. For $t = T$, only the $(T-1)$-indexed terms contain $P_T$, giving $-\gamma^{T-1}Q_s + \gamma^T\Lambda_T = 0$, hence $\Lambda_T = \dfrac{1}{\gamma}Q_s$.

(ii) Envelope / stationarity w.r.t. $K$ and $\Sigma$. Only two parts of $\mathcal{L}$ depend explicitly on $K$: the quadratic term in $a_t$ and the dynamics through $F = A + BK$. Using $dF = B\, dK$, symmetricity of $P_t, \Lambda_t$, and standard identities for matrix differentials,

$$d\,\mathrm{tr}(Q_a\, KP_tK^\top) = 2\,\mathrm{tr}\,((Q_aKP_t)^\top dK), \quad d\,\mathrm{tr}(\Lambda_{t+1}FP_tF^\top) = 2\,\mathrm{tr}\,((B^\top\Lambda_{t+1}FP_t)^\top dK).$$

Therefore,

$$\partial_K\mathcal{L} = -2\sum_{t=0}^{T-1} \gamma^t\,(Q_aKP_t) - 2\sum_{t=0}^{T-1} \gamma^{t+1}\,(B^\top\Lambda_{t+1}FP_t) = -2\sum_{t=0}^{T-1} \gamma^t\left(Q_aKP_t + \gamma B^\top\Lambda_{t+1}F\, P_t\right)$$

Only two places depend on $\Sigma$: the immediate term $-\gamma^t\,\mathrm{tr}(Q_a\Sigma)$ and the constraint term $-\gamma^{t+1}\,\mathrm{tr}(\Lambda_{t+1}\, B\Sigma B^\top)$. Then

$$\frac{\partial\mathcal{L}}{\partial\Sigma} = -\sum_{t=0}^{T-1} \gamma^t Q_a - \sum_{t=0}^{T-1} \gamma^{t+1}\, B^\top\Lambda_{t+1}B = -\sum_{t=0}^{T-1} \gamma^t\left(Q_a + \gamma B^\top\Lambda_{t+1}B\right),$$

Therefore, the gradient with respect to the log-std parameter $\ell$ is

$$\nabla_\ell J_T = -2\,\mathrm{diag}\left(\Sigma\sum_{t=0}^{T-1} \gamma^t\left(Q_a + \gamma B^\top\Lambda_{t+1}B\right)\right) \in \mathbb{R}^m. \tag{36}$$

**Variance of $K$-part: selectors and lifted state expansion.** Introduce block selectors $\Pi_t$ such that $\varepsilon_t = \Pi_t \bar{\varepsilon}$. Then

$$\varepsilon_t^\top \Sigma^{-2} \varepsilon_p = \bar{\varepsilon}^\top W_{tp}^K \bar{\varepsilon}, \qquad W_{tp}^K := \Pi_t^\top \Sigma^{-2} \Pi_p.$$

Also, with $s_t = S_t \bar{s}$ and $\bar{s} = \mathcal{F}_S s_0 + \mathcal{F}_E \bar{\varepsilon}$, define

$$F_t := S_t \mathcal{F}_S, \qquad E_t := S_t \mathcal{F}_E,$$

so that $s_t = F_t s_0 + E_t \bar{\varepsilon}$ and hence

$$s_t^\top s_p = s_0^\top (F_t^\top F_p) s_0 + s_0^\top (F_t^\top E_p + F_p^\top E_t) \bar{\varepsilon} + \bar{\varepsilon}^\top (E_t^\top E_p) \bar{\varepsilon}. \tag{37}$$

Define

$$G_{tp} := F_t^\top F_p, \qquad J_{tp} := F_t^\top E_p + F_p^\top E_t, \qquad U_{tp} := E_t^\top E_p.$$

**Parity bookkeeping after conditioning on $s_0$ ($K$-part).** Condition on $s_0$. Since $s_0 \perp\!\!\!\perp \bar{\varepsilon}$ and $\bar{\varepsilon}$ is centered Gaussian, only even total degree polynomials in $\bar{\varepsilon}$ survive inside $\mathbb{E}_{\bar{\varepsilon}}[\cdot \mid s_0]$. Note

$$\bar{\varepsilon}^\top W_{tp}^K \bar{\varepsilon} \text{ is degree 2}, \quad s_0^\top G_{tp} s_0 \text{ is degree 0}, \quad s_0^\top J_{tp} \bar{\varepsilon} \text{ is degree 1}, \quad \bar{\varepsilon}^\top U_{tp} \bar{\varepsilon} \text{ is degree 2},$$

and among (32),

$$x^2 \text{ degree 0}, \quad z^2 \text{ degree 4}, \quad xz \text{ degree 2}, \quad y^2 \text{ degree 2}, \quad xy \text{ degree 1}, \quad yz \text{ degree 3}.$$

Therefore, for the *even* part $x^2 + z^2 + 2xz + 4y^2$, the linear term $s_0^\top J_{tp} \bar{\varepsilon}$ in (37) yields an odd integrand and vanishes, so

$$s_t^\top s_p \;\to\; s_0^\top G_{tp} s_0 + \bar{\varepsilon}^\top U_{tp} \bar{\varepsilon} \qquad \text{when multiplying} \qquad (x^2 + z^2 + 2xz + 4y^2).$$

For the *odd* part $4xy + 4yz$, only the linear term survives (odd×odd becomes even), hence

$$s_t^\top s_p \;\to\; s_0^\top J_{tp} \bar{\varepsilon} \qquad \text{when multiplying} \qquad (4xy + 4yz).$$

**Six-term decomposition ($K$-part).** Substituting $\varepsilon_t^\top \Sigma^{-2} \varepsilon_p = \bar{\varepsilon}^\top W_{tp}^K \bar{\varepsilon}$ and using the parity reductions, (33) becomes

$$\mathbb{E}[\|\widehat{G}_K\|_{\mathrm{F}}^2] = \sum_{t,p=0}^{T-1} (E_{1,tp}^K + E_{2,tp}^K + E_{3,tp}^K + E_{4,tp}^K + E_{5,tp}^K + E_{6,tp}^K),$$

where the *even-part* contributions are

$$E_{1,tp}^K := \mathbb{E}\left[ (\bar{\varepsilon}^\top W_{tp}^K \bar{\varepsilon}) \Big( s_0^\top G_{tp} s_0 + \bar{\varepsilon}^\top U_{tp} \bar{\varepsilon} \Big) x^2 \right],$$

$$E_{2,tp}^K := \mathbb{E}\left[ (\bar{\varepsilon}^\top W_{tp}^K \bar{\varepsilon}) \Big( s_0^\top G_{tp} s_0 + \bar{\varepsilon}^\top U_{tp} \bar{\varepsilon} \Big) z^2 \right],$$

$$E_{3,tp}^K := 2\,\mathbb{E}\left[ (\bar{\varepsilon}^\top W_{tp}^K \bar{\varepsilon}) \Big( s_0^\top G_{tp} s_0 + \bar{\varepsilon}^\top U_{tp} \bar{\varepsilon} \Big) xz \right],$$

$$E_{4,tp}^K := 4\,\mathbb{E}\left[ (\bar{\varepsilon}^\top W_{tp}^K \bar{\varepsilon}) \Big( s_0^\top G_{tp} s_0 + \bar{\varepsilon}^\top U_{tp} \bar{\varepsilon} \Big) y^2 \right],$$

and the *odd-part* contributions are

$$E_{5,tp}^K := 4\,\mathbb{E}\left[ (\bar{\varepsilon}^\top W_{tp}^K \bar{\varepsilon}) (s_0^\top J_{tp} \bar{\varepsilon}) \, x\, y \right], \qquad E_{6,tp}^K := 4\,\mathbb{E}\left[ (\bar{\varepsilon}^\top W_{tp}^K \bar{\varepsilon}) (s_0^\top J_{tp} \bar{\varepsilon}) \, y\, z \right]. \tag{38}$$

**Term $E_{1,tp}^K$ ($x^2$-part).** Since $x$ depends only on $s_0$, we can factor expectations:

$$\begin{aligned}
E_{1,tp}^K &= \mathbb{E}\left[ (\bar{\varepsilon}^\top W_{tp}^K \bar{\varepsilon})(s_0^\top G_{tp} s_0) \, x^2 \right] + \mathbb{E}\left[ (\bar{\varepsilon}^\top W_{tp}^K \bar{\varepsilon})(\bar{\varepsilon}^\top U_{tp} \bar{\varepsilon}) \, x^2 \right] \\
&= \mathrm{IS}_{\Sigma_0}(\mathbf{sym}(G_{tp}), \mathbf{sym}(\overline{M}_{ss}), \mathbf{sym}(\overline{M}_{ss})) \, \mathrm{IS}_{\overline{\Sigma}}(\mathbf{sym}(W_{tp}^K)) \\
&\quad + \mathrm{IS}_{\Sigma_0}(\mathbf{sym}(\overline{M}_{ss}), \mathbf{sym}(\overline{M}_{ss})) \, \mathrm{IS}_{\overline{\Sigma}}(\mathbf{sym}(W_{tp}^K), \mathbf{sym}(U_{tp})).
\end{aligned} \tag{39}$$

**Term $E_{2,tp}^K$ ($z^2$-part).**

$$E_{2,tp}^K = \mathrm{IS}_{\Sigma_0}(\mathbf{sym}(G_{tp}))\, \mathrm{IS}_{\overline{\Sigma}}(\mathbf{sym}(W_{tp}^K), \mathbf{sym}(\overline{M}_{ee}), \mathbf{sym}(\overline{M}_{ee}))$$
$$+ \mathrm{IS}_{\overline{\Sigma}}(\mathbf{sym}(W_{tp}^K), \mathbf{sym}(U_{tp}), \mathbf{sym}(\overline{M}_{ee}), \mathbf{sym}(\overline{M}_{ee})). \tag{40}$$

**Term $E_{3,tp}^K$ ($2xz$-part).**

$$E_{3,tp}^K = 2\,\mathrm{IS}_{\Sigma_0}(\mathbf{sym}(G_{tp}), \mathbf{sym}(\overline{M}_{ss}))\, \mathrm{IS}_{\overline{\Sigma}}(\mathbf{sym}(W_{tp}^K), \mathbf{sym}(\overline{M}_{ee}))$$
$$+ 2\,\mathrm{IS}_{\Sigma_0}(\mathbf{sym}(\overline{M}_{ss}))\, \mathrm{IS}_{\overline{\Sigma}}(\mathbf{sym}(W_{tp}^K), \mathbf{sym}(U_{tp}), \mathbf{sym}(\overline{M}_{ee})). \tag{41}$$

**Term $E_{4,tp}^K$ ($4y^2$-part).** Since $y = s_0^\top \overline{M}_{se}\bar{\varepsilon}$, we have

$$y^2 = \bar{\varepsilon}^\top H(s_0)\bar{\varepsilon}, \qquad H(s_0) := \overline{M}_{se}^\top s_0 s_0^\top \overline{M}_{se}.$$

Then

$$E_{4,tp}^K = 4\,\mathbb{E}\left[ (\bar{\varepsilon}^\top W_{tp}^K \bar{\varepsilon})\Big(s_0^\top G_{tp} s_0 + \bar{\varepsilon}^\top U_{tp}\bar{\varepsilon}\Big)(\bar{\varepsilon}^\top H(s_0)\bar{\varepsilon}) \right]$$
$$= 4\,\mathbb{E}_{s_0}\left[ (s_0^\top G_{tp} s_0)\underbrace{\mathbb{E}_{\bar{\varepsilon}}\left[(\bar{\varepsilon}^\top W_{tp}^K \bar{\varepsilon})(\bar{\varepsilon}^\top H(s_0)\bar{\varepsilon}) \,\big|\, s_0\right]}_{=:T_1(s_0)} + \underbrace{\mathbb{E}_{\bar{\varepsilon}}\left[(\bar{\varepsilon}^\top W_{tp}^K \bar{\varepsilon})(\bar{\varepsilon}^\top U_{tp}\bar{\varepsilon})(\bar{\varepsilon}^\top H(s_0)\bar{\varepsilon}) \,\big|\, s_0\right]}_{=:T_2(s_0)} \right]. \tag{42}$$

By Lemma 2 (the $k=2$ and $k=3$ cases) with $\Omega = \overline{\Sigma}$,

$$T_1(s_0) = \mathrm{IS}_{\overline{\Sigma}}(\mathbf{sym}(W_{tp}^K), H(s_0))$$
$$= \mathrm{tr}(\overline{\Sigma}\,\mathbf{sym}(W_{tp}^K))\,\mathrm{tr}(\overline{\Sigma}\, H(s_0)) + 2\,\mathrm{tr}(\overline{\Sigma}\,\mathbf{sym}(W_{tp}^K)\,\overline{\Sigma}\, H(s_0)), \tag{43}$$
$$T_2(s_0) = \mathrm{IS}_{\overline{\Sigma}}(\mathbf{sym}(W_{tp}^K), \mathbf{sym}(U_{tp}), H(s_0))$$
$$= \mathrm{tr}(\overline{\Sigma}\,\mathbf{sym}(W_{tp}^K))\,\mathrm{tr}(\overline{\Sigma}\,\mathbf{sym}(U_{tp}))\,\mathrm{tr}(\overline{\Sigma}\, H(s_0)) + 2\,\mathrm{tr}(\overline{\Sigma}\,\mathbf{sym}(W_{tp}^K)\,\overline{\Sigma}\,\mathbf{sym}(U_{tp}))\,\mathrm{tr}(\overline{\Sigma}\, H(s_0))$$
$$+ 2\,\mathrm{tr}(\overline{\Sigma}\,\mathbf{sym}(W_{tp}^K)\,\overline{\Sigma}\, H(s_0))\,\mathrm{tr}(\overline{\Sigma}\,\mathbf{sym}(U_{tp})) + 2\,\mathrm{tr}(\overline{\Sigma}\,\mathbf{sym}(U_{tp})\,\overline{\Sigma}\, H(s_0))\,\mathrm{tr}(\overline{\Sigma}\,\mathbf{sym}(W_{tp}^K))$$
$$+ 8\,\mathrm{tr}(\overline{\Sigma}\,\mathbf{sym}(W_{tp}^K)\,\overline{\Sigma}\,\mathbf{sym}(U_{tp})\,\overline{\Sigma}\, H(s_0)). \tag{44}$$

Introduce $\alpha_{tp} := \mathrm{tr}(\overline{\Sigma}\,\mathbf{sym}(W_{tp}^K))$, $\beta_{tp} := \mathrm{tr}(\overline{\Sigma}\,\mathbf{sym}(U_{tp}))$, $\delta_{tp} := \mathrm{tr}(\overline{\Sigma}\,\mathbf{sym}(W_{tp}^K)\,\overline{\Sigma}\,\mathbf{sym}(U_{tp}))$, and the matrices

$$V_0 := \overline{M}_{se}\,\overline{\Sigma}\,\overline{M}_{se}^\top, \qquad V_{tp} := \overline{M}_{se}\,\overline{\Sigma}\,\mathbf{sym}(W_{tp}^K)\,\overline{\Sigma}\,\overline{M}_{se}^\top,$$

$$\widetilde{V}_{tp} := \overline{M}_{se}\,\overline{\Sigma}\,\mathbf{sym}(U_{tp})\,\overline{\Sigma}\,\overline{M}_{se}^\top, \qquad \widehat{V}_{tp} := \overline{M}_{se}\,\overline{\Sigma}\,\mathbf{sym}(W_{tp}^K)\,\overline{\Sigma}\,\mathbf{sym}(U_{tp})\,\overline{\Sigma}\,\overline{M}_{se}^\top.$$

Using $H(s_0) = \overline{M}_{se}^\top s_0 s_0^\top \overline{M}_{se}$ and cyclicity of trace, we obtain

$$\mathrm{tr}(\overline{\Sigma}\, H(s_0)) = s_0^\top V_0 s_0, \qquad \mathrm{tr}(\overline{\Sigma}\,\mathbf{sym}(W_{tp}^K)\,\overline{\Sigma}\, H(s_0)) = s_0^\top V_{tp} s_0,$$

$$\mathrm{tr}(\overline{\Sigma}\,\mathbf{sym}(U_{tp})\,\overline{\Sigma}\, H(s_0)) = s_0^\top \widetilde{V}_{tp} s_0, \qquad \mathrm{tr}(\overline{\Sigma}\,\mathbf{sym}(W_{tp}^K)\,\overline{\Sigma}\,\mathbf{sym}(U_{tp})\,\overline{\Sigma}\, H(s_0)) = s_0^\top \widehat{V}_{tp} s_0.$$

Substituting into (43)–(44) yields

$$T_1(s_0) = \alpha_{tp}(s_0^\top V_0 s_0) + 2(s_0^\top V_{tp} s_0),$$

$$T_2(s_0) = (\alpha_{tp}\beta_{tp} + 2\delta_{tp})(s_0^\top V_0 s_0) + 2\beta_{tp}(s_0^\top V_{tp} s_0) + 2\alpha_{tp}(s_0^\top \widetilde{V}_{tp} s_0) + 8(s_0^\top \widehat{V}_{tp} s_0).$$

Taking $\mathbb{E}_{s_0}$ term-by-term and using Lemma 2 with $\Omega = \Sigma_0$ gives

$$E_{4,tp}^K = 4\Big(\alpha_{tp}\,\mathrm{IS}_{\Sigma_0}(\mathbf{sym}(G_{tp}), V_0) + 2\,\mathrm{IS}_{\Sigma_0}(\mathbf{sym}(G_{tp}), V_{tp}) + (\alpha_{tp}\beta_{tp} + 2\delta_{tp})\,\mathrm{IS}_{\Sigma_0}(V_0)$$
$$+ 2\beta_{tp}\,\mathrm{IS}_{\Sigma_0}(V_{tp}) + 2\alpha_{tp}\,\mathrm{IS}_{\Sigma_0}(\widetilde{V}_{tp}) + 8\,\mathrm{IS}_{\Sigma_0}(\mathbf{sym}(\widehat{V}_{tp}))\Big). \tag{45}$$

**Term $E_{5,tp}^K$ ($4xy$-part).** Recall $y = s_0^\top \overline{M}_{se} \bar\varepsilon$. Define

$$L_{tp}(s_0) := J_{tp}^\top s_0 s_0^\top \overline{M}_{se} \in \mathbb{R}^{mT \times mT}.$$

Then

$$(s_0^\top J_{tp} \bar\varepsilon)(s_0^\top \overline{M}_{se} \bar\varepsilon) = \bar\varepsilon^\top L_{tp}(s_0) \bar\varepsilon,$$

and since $x = s_0^\top \overline{M}_{ss} s_0$ depends only on $s_0$, conditioning on $s_0$ gives

$$\begin{aligned}
E_{5,tp}^K &= 4\, \mathbb{E}_{s_0} \left[ x\, \mathbb{E}_{\bar\varepsilon} \left[ (\bar\varepsilon^\top W_{tp}^K \bar\varepsilon)(\bar\varepsilon^\top L_{tp}(s_0) \bar\varepsilon) \,\big|\, s_0 \right] \right] \\
&= 4\, \mathbb{E}_{s_0} \left[ x\, \mathrm{IS}_{\overline{\Sigma}}(\mathbf{sym}(W_{tp}^K), \mathbf{sym}(L_{tp}(s_0))) \right].
\end{aligned} \tag{46}$$

Introduce $\alpha_{tp} := \mathrm{tr}(\overline{\Sigma}\, \mathbf{sym}(W_{tp}^K))$ and define

$$V_{tp}^{(0)} := \mathbf{sym}\left( \overline{M}_{se} \overline{\Sigma}\, J_{tp}^\top \right), \qquad V_{tp}^{(1)} := \mathbf{sym}\left( \overline{M}_{se} \overline{\Sigma}\, \mathbf{sym}(W_{tp}^K) \overline{\Sigma}\, J_{tp}^\top \right).$$

Then the same trace-to-quadratic-form yields

$$E_{5,tp}^K = 4\left( \alpha_{tp}\, \mathrm{IS}_{\Sigma_0}(\mathbf{sym}(\overline{M}_{ss}), V_{tp}^{(0)}) + 2\, \mathrm{IS}_{\Sigma_0}(\mathbf{sym}(\overline{M}_{ss}), V_{tp}^{(1)}) \right). \tag{47}$$

**Term $E_{6,tp}^K$ ($4yz$-part).** Using the same $L_{tp}(s_0)$ and $z = \bar\varepsilon^\top \overline{M}_{ee} \bar\varepsilon$, conditioning on $s_0$ gives

$$E_{6,tp}^K = 4\, \mathbb{E}_{s_0} \left[ \mathrm{IS}_{\overline{\Sigma}}\left( \mathbf{sym}(W_{tp}^K), \mathbf{sym}(\overline{M}_{ee}), \mathbf{sym}(L_{tp}(s_0)) \right) \right]. \tag{48}$$

Define

$$\alpha_{tp} := \mathrm{tr}(\overline{\Sigma}\, \mathbf{sym}(W_{tp}^K)), \qquad \mu := \mathrm{tr}(\overline{\Sigma}\, \mathbf{sym}(\overline{M}_{ee})), \qquad \kappa_{tp} := \mathrm{tr}(\overline{\Sigma}\, \mathbf{sym}(W_{tp}^K) \overline{\Sigma}\, \mathbf{sym}(\overline{M}_{ee})).$$

With

$$A^{(0)} := \overline{\Sigma}, \quad A^{(1)} := \overline{\Sigma}\, \mathbf{sym}(W_{tp}^K) \overline{\Sigma}, \quad A^{(2)} := \overline{\Sigma}\, \mathbf{sym}(\overline{M}_{ee}) \overline{\Sigma}, \quad A^{(3)} := \mathbf{sym}(\overline{\Sigma}\, \mathbf{sym}(W_{tp}^K) \overline{\Sigma}\, \mathbf{sym}(\overline{M}_{ee}) \overline{\Sigma}),$$

and

$$V_{tp}^{(j)} := \mathbf{sym}\left( \overline{M}_{se} A^{(j)} J_{tp}^\top \right) \quad (j = 0, 1, 2, 3),$$

the trace-to-quadratic-form conversion gives

$$E_{6,tp}^K = 4\left( (\alpha_{tp}\mu + 2\kappa_{tp})\, \mathrm{IS}_{\Sigma_0}(V_{tp}^{(0)}) + 2\mu\, \mathrm{IS}_{\Sigma_0}(V_{tp}^{(1)}) + 2\alpha_{tp}\, \mathrm{IS}_{\Sigma_0}(V_{tp}^{(2)}) + 8\, \mathrm{IS}_{\Sigma_0}(V_{tp}^{(3)}) \right). \tag{49}$$

**$\ell$-part: lifted representation of $g_t^\top g_p$.** Introduce $\Pi_t$ such that $\varepsilon_t = \Pi_t \bar\varepsilon$. Write $D := \Sigma^{-1} = \mathrm{diag}(d_1, \ldots, d_m)$. Then

$$g_t^\top g_p = (D(\varepsilon_t \odot \varepsilon_t) - \mathbf{1})^\top (D(\varepsilon_p \odot \varepsilon_p) - \mathbf{1}) = (\varepsilon_t \odot \varepsilon_t)^\top D^2 (\varepsilon_p \odot \varepsilon_p) - \varepsilon_t^\top D\varepsilon_t - \varepsilon_p^\top D\varepsilon_p + m.$$

To express the quartic term via quadratic forms, define for each $i \in \{1, \ldots, m\}$

$$Q_{t,i} := \Pi_t^\top e_i e_i^\top \Pi_t \in \mathbb{R}^{mT \times mT} \quad \Rightarrow \quad \varepsilon_{t,i}^2 = \bar\varepsilon^\top Q_{t,i} \bar\varepsilon,$$

and define

$$W_t^\ell := \Pi_t^\top D \Pi_t \in \mathbb{R}^{mT \times mT} \quad \Rightarrow \quad \varepsilon_t^\top D\varepsilon_t = \bar\varepsilon^\top W_t^\ell \bar\varepsilon.$$

Then

$$g_t^\top g_p = \sum_{i=1}^m d_i^2 (\bar\varepsilon^\top Q_{t,i} \bar\varepsilon)(\bar\varepsilon^\top Q_{p,i} \bar\varepsilon) - (\bar\varepsilon^\top W_t^\ell \bar\varepsilon) - (\bar\varepsilon^\top W_p^\ell \bar\varepsilon) + m. \tag{50}$$

**Four-term decomposition ($\ell$-part).** From (34),

$$\mathbb{E}[\|\widehat{G}_\ell\|_2^2] = \sum_{t,p=0}^{T-1} (E_{1,tp}^\ell + E_{2,tp}^\ell + E_{3,tp}^\ell + E_{4,tp}^\ell),$$

where

$$E_{1,tp}^\ell := \mathbb{E}[(g_t^\top g_p)x^2], \quad E_{2,tp}^\ell := \mathbb{E}[(g_t^\top g_p)z^2], \quad E_{3,tp}^\ell := 2\,\mathbb{E}[(g_t^\top g_p)xz], \quad E_{4,tp}^\ell := 4\,\mathbb{E}[(g_t^\top g_p)y^2].$$

**Term $E_{1,tp}^\ell$ ($x^2$-part).** Since $x$ depends only on $s_0$ and $g_t^\top g_p$ depends only on $\bar{\varepsilon}$,

$$E_{1,tp}^\ell = \mathbb{E}[x^2]\,\mathbb{E}[g_t^\top g_p] = \mathrm{IS}_{\Sigma_0}(\mathbf{sym}(\overline{M}_{ss}), \mathbf{sym}(\overline{M}_{ss}))\,\mathbb{E}[g_t^\top g_p].$$

Under $\overline{\Sigma} = I_T \otimes \Sigma$, $\{g_t\}$ are independent across time and $\mathbb{E}[g_t] = 0$, hence

$$\mathbb{E}[g_t^\top g_p] = \mathbb{E}[\|g_t\|_2^2]\mathbf{1}_{\{t=p\}} = 2m\,\mathbf{1}_{\{t=p\}},$$

so

$$E_{1,tp}^\ell = 2m\,\mathbf{1}_{\{t=p\}}\,\mathrm{IS}_{\Sigma_0}(\mathbf{sym}(\overline{M}_{ss}), \mathbf{sym}(\overline{M}_{ss})). \tag{51}$$

**Term $E_{2,tp}^\ell$ ($z^2$-part).** Using (50) and $z = \bar{\varepsilon}^\top \overline{M}_{ee}\bar{\varepsilon}$,

$$\begin{aligned}
E_{2,tp}^\ell = &\sum_{i=1}^m d_i^2\, \mathrm{IS}_{\overline{\Sigma}}(\mathbf{sym}(Q_{t,i}), \mathbf{sym}(Q_{p,i}), \mathbf{sym}(\overline{M}_{ee}), \mathbf{sym}(\overline{M}_{ee})) \\
&- \mathrm{IS}_{\overline{\Sigma}}(\mathbf{sym}(W_t^\ell), \mathbf{sym}(\overline{M}_{ee}), \mathbf{sym}(\overline{M}_{ee})) - \mathrm{IS}_{\overline{\Sigma}}(\mathbf{sym}(W_p^\ell), \mathbf{sym}(\overline{M}_{ee}), \mathbf{sym}(\overline{M}_{ee})) \\
&+ m\,\mathrm{IS}_{\overline{\Sigma}}(\mathbf{sym}(\overline{M}_{ee}), \mathbf{sym}(\overline{M}_{ee})).
\end{aligned} \tag{52}$$

**Term $E_{3,tp}^\ell$ ($2xz$-part).** Since $x$ depends only on $s_0$,

$$E_{3,tp}^\ell = 2\,\mathbb{E}[x]\,\mathbb{E}[(g_t^\top g_p)z] = 2\,\mathrm{IS}_{\Sigma_0}(\mathbf{sym}(\overline{M}_{ss}))\,\mathbb{E}[(g_t^\top g_p)z].$$

By (50),

$$\begin{aligned}
\mathbb{E}[(g_t^\top g_p)z] = &\sum_{i=1}^m d_i^2\, \mathrm{IS}_{\overline{\Sigma}}(\mathbf{sym}(Q_{t,i}), \mathbf{sym}(Q_{p,i}), \mathbf{sym}(\overline{M}_{ee})) \\
&- \mathrm{IS}_{\overline{\Sigma}}(\mathbf{sym}(W_t^\ell), \mathbf{sym}(\overline{M}_{ee})) - \mathrm{IS}_{\overline{\Sigma}}(\mathbf{sym}(W_p^\ell), \mathbf{sym}(\overline{M}_{ee})) + m\,\mathrm{IS}_{\overline{\Sigma}}(\mathbf{sym}(\overline{M}_{ee})),
\end{aligned} \tag{53}$$

hence

$$\begin{aligned}
E_{3,tp}^\ell = 2\,\mathrm{IS}_{\Sigma_0}(\mathbf{sym}(\overline{M}_{ss}))\Big[&\sum_{i=1}^m d_i^2\, \mathrm{IS}_{\overline{\Sigma}}(\mathbf{sym}(Q_{t,i}), \mathbf{sym}(Q_{p,i}), \mathbf{sym}(\overline{M}_{ee})) - \mathrm{IS}_{\overline{\Sigma}}(\mathbf{sym}(W_t^\ell), \mathbf{sym}(\overline{M}_{ee})) \\
&- \mathrm{IS}_{\overline{\Sigma}}(\mathbf{sym}(W_p^\ell), \mathbf{sym}(\overline{M}_{ee})) + m\,\mathrm{IS}_{\overline{\Sigma}}(\mathbf{sym}(\overline{M}_{ee}))\Big].
\end{aligned} \tag{54}$$

**Term $E_{4,tp}^\ell$ ($4y^2$-part).** Recall $y = s_0^\top \overline{M}_{se}\bar{\varepsilon}$ and define

$$H(s_0) := \overline{M}_{se}^\top s_0 s_0^\top \overline{M}_{se} \quad \Rightarrow \quad y^2 = \bar{\varepsilon}^\top H(s_0)\bar{\varepsilon}.$$

Conditioning on $s_0$ and using (50) yields

$$\begin{aligned}
E_{4,tp}^\ell = 4\,\mathbb{E}_{s_0}\Big[&\sum_{i=1}^m d_i^2\, \mathrm{IS}_{\overline{\Sigma}}(\mathbf{sym}(Q_{t,i}), \mathbf{sym}(Q_{p,i}), H(s_0)) - \mathrm{IS}_{\overline{\Sigma}}(\mathbf{sym}(W_t^\ell), H(s_0)) \\
&- \mathrm{IS}_{\overline{\Sigma}}(\mathbf{sym}(W_p^\ell), H(s_0)) + m\,\mathrm{IS}_{\overline{\Sigma}}(H(s_0))\Big].
\end{aligned} \tag{55}$$

Introduce $c(s_0) := \mathrm{tr}(\overline{\Sigma}\, H(s_0))$. Using cyclicity of trace,

$$c(s_0) = s_0^\top V_0 s_0, \qquad V_0 := \overline{M}_{se}\, \overline{\Sigma}\, \overline{M}_{se}^\top. \tag{56}$$

Define also

$$V_{t,i} := \overline{M}_{se}\, \overline{\Sigma}\, \mathbf{sym}(Q_{t,i})\, \overline{\Sigma}\, \overline{M}_{se}^\top, \qquad V_{tp,i} := \mathbf{sym}\Big(\overline{M}_{se}\, \overline{\Sigma}\, \mathbf{sym}(Q_{t,i})\, \overline{\Sigma}\, \mathbf{sym}(Q_{p,i})\, \overline{\Sigma}\, \overline{M}_{se}^\top\Big),$$

so that

$$\mathrm{tr}(\overline{\Sigma}\, \mathbf{sym}(Q_{t,i})\, \overline{\Sigma}\, H(s_0)) = s_0^\top V_{t,i} s_0, \qquad \mathrm{tr}(\overline{\Sigma}\, \mathbf{sym}(Q_{t,i})\, \overline{\Sigma}\, \mathbf{sym}(Q_{p,i})\, \overline{\Sigma}\, H(s_0)) = s_0^\top V_{tp,i} s_0. \tag{57}$$

*Step 1: expand the $k = 3$ term.* By Lemma 2 (the $k = 3$ case) with $\Omega = \overline{\Sigma}$,

$$\begin{aligned}
\mathrm{IS}_{\overline{\Sigma}}(\mathbf{sym}(Q_{t,i}), \mathbf{sym}(Q_{p,i}), H(s_0)) = {}& \mathrm{tr}(\overline{\Sigma}\, Q_{t,i})\, \mathrm{tr}(\overline{\Sigma}\, Q_{p,i})\, c(s_0) + 2\, \mathrm{tr}(\overline{\Sigma}\, Q_{t,i}\, \overline{\Sigma}\, Q_{p,i})\, c(s_0) \\
&+ 2\, \mathrm{tr}(\overline{\Sigma}\, Q_{t,i}\, \overline{\Sigma}\, H(s_0))\, \mathrm{tr}(\overline{\Sigma}\, Q_{p,i}) + 2\, \mathrm{tr}(\overline{\Sigma}\, Q_{p,i}\, \overline{\Sigma}\, H(s_0))\, \mathrm{tr}(\overline{\Sigma}\, Q_{t,i}) \\
&+ 8\, \mathrm{tr}(\overline{\Sigma}\, Q_{t,i}\, \overline{\Sigma}\, Q_{p,i}\, \overline{\Sigma}\, H(s_0)).
\end{aligned} \tag{58}$$

Under $\overline{\Sigma} = I_T \otimes \Sigma$,

$$\mathrm{tr}(\overline{\Sigma}\, Q_{t,i}) = \sigma_i^2, \qquad \mathrm{tr}(\overline{\Sigma}\, Q_{t,i}\, \overline{\Sigma}\, Q_{p,i}) = \mathbf{1}_{\{t=p\}}\sigma_i^4.$$

Multiplying by $d_i^2 = \sigma_i^{-4}$ and using (57) gives

$$d_i^2\, \mathrm{IS}_{\overline{\Sigma}}(Q_{t,i}, Q_{p,i}, H(s_0)) = c(s_0) + 2\, \mathbf{1}_{\{t=p\}}\, c(s_0) + 2d_i\, s_0^\top V_{t,i} s_0 + 2d_i\, s_0^\top V_{p,i} s_0 + 8d_i^2\, s_0^\top V_{tp,i} s_0. \tag{59}$$

Summing (59) over $i$ yields

$$\sum_{i=1}^m d_i^2\, \mathrm{IS}_{\overline{\Sigma}}(\mathbf{sym}(Q_{t,i}), \mathbf{sym}(Q_{p,i}), H(s_0)) = m\, c(s_0) + 2m\, \mathbf{1}_{\{t=p\}}\, c(s_0) + 2\, s_0^\top \Big(\sum_{i=1}^m d_i V_{t,i}\Big) s_0 + 2\, s_0^\top \Big(\sum_{i=1}^m d_i V_{p,i}\Big) s_0$$

$$+ 8\, s_0^\top \Big(\sum_{i=1}^m d_i^2 V_{tp,i}\Big) s_0. \tag{60}$$

*Step 2: expand the $k = 2$ terms.* By Lemma 2 (the $k = 2$ case),

$$\mathrm{IS}_{\overline{\Sigma}}(W_t^\ell, H(s_0)) = \mathrm{tr}(\overline{\Sigma}\, W_t^\ell)\, c(s_0) + 2\, \mathrm{tr}(\overline{\Sigma}\, W_t^\ell\, \overline{\Sigma}\, H(s_0)). \tag{61}$$

Under $\overline{\Sigma} = I_T \otimes \Sigma$ and $W_t^\ell = \Pi_t^\top \Sigma^{-1}\Pi_t$,

$$\mathrm{tr}(\overline{\Sigma}\, W_t^\ell) = \mathrm{tr}(\Sigma\, \Sigma^{-1}) = m,$$

and

$$\mathrm{tr}(\overline{\Sigma}\, W_t^\ell\, \overline{\Sigma}\, H(s_0)) = s_0^\top \Big(\overline{M}_{se}\, \overline{\Sigma}\, W_t^\ell\, \overline{\Sigma}\, \overline{M}_{se}^\top\Big) s_0. \tag{62}$$

Thus

$$\mathrm{IS}_{\overline{\Sigma}}(W_t^\ell, H(s_0)) = m\, c(s_0) + 2\, s_0^\top \Big(\overline{M}_{se}\, \overline{\Sigma}\, W_t^\ell\, \overline{\Sigma}\, \overline{M}_{se}^\top\Big) s_0, \tag{63}$$

and similarly for $p$.

*Step 3: cancellation via $\sum_i d_i Q_{t,i} = W_t^\ell$.* Note that

$$\sum_{i=1}^m d_i\, Q_{t,i} = \Pi_t^\top \Big(\sum_{i=1}^m d_i e_i e_i^\top\Big)\Pi_t = \Pi_t^\top \Sigma^{-1}\Pi_t = W_t^\ell.$$

Therefore

$$\sum_{i=1}^m d_i\, V_{t,i} = \overline{M}_{se}\, \overline{\Sigma}\, W_t^\ell\, \overline{\Sigma}\, \overline{M}_{se}^\top, \qquad \sum_{i=1}^m d_i\, V_{p,i} = \overline{M}_{se}\, \overline{\Sigma}\, W_p^\ell\, \overline{\Sigma}\, \overline{M}_{se}^\top. \tag{64}$$

*Step 4: assemble* (55). Substituting (60), (63), and $\mathrm{IS}_{\bar{\Sigma}}(H(s_0)) = c(s_0)$, and using the cancellation (64), the bracket in (55) reduces to

$$2m \, \mathbf{1}_{\{t=p\}} \, c(s_0) + 8 \, s_0^\top \Big( \sum_{i=1}^{m} d_i^2 V_{tp,i} \Big) s_0.$$

Hence

$$E_{4,tp}^\ell = 4 \, \mathbb{E}_{s_0} \left[ 2m \, \mathbf{1}_{\{t=p\}} \, c(s_0) + 8 \, s_0^\top \Big( \sum_{i=1}^{m} d_i^2 V_{tp,i} \Big) s_0 \right].$$

Using Lemma 2 (the $k = 1$ case) with $\Omega = \Sigma_0$ yields

$$E_{4,tp}^\ell = 8m \, \mathbf{1}_{\{t=p\}} \, \mathrm{IS}_{\Sigma_0}(V_0) \; + \; 32 \sum_{i=1}^{m} d_i^2 \, \mathrm{IS}_{\Sigma_0}(V_{tp,i}). \tag{65}$$

Moreover, under $\bar{\bar{\Sigma}} = I_T \otimes \Sigma$ the matrices $Q_{t,i}$ and $Q_{p,i}$ occupy disjoint time blocks when $t \neq p$, which implies $V_{tp,i} = 0$ for $t \neq p$.

**Conclusion.** The $K$-part decomposes as

$$\mathbb{E}[\|\widehat{G}_K\|_F^2] = \sum_{t,p=0}^{T-1} (E_{1,tp}^K + E_{2,tp}^K + E_{3,tp}^K + E_{4,tp}^K + E_{5,tp}^K + E_{6,tp}^K),$$

with $E_{i,tp}^K, i = 1, ..., 6$ given by (39), (40), (41), (45), (47), (49), The $\ell$-part decomposes as

$$\mathbb{E}[\|\widehat{G}_\ell\|_2^2] = \sum_{t,p=0}^{T-1} (E_{1,tp}^\ell + E_{2,tp}^\ell + E_{3,tp}^\ell + E_{4,tp}^\ell),$$

with $E_{1,tp}^\ell, E_{2,tp}^\ell, E_{3,tp}^\ell, E_{4,tp}^\ell$ given by (51), (52), (54), (65). This completes the proof. $\qquad\square$

# I. Proof of Corollary 7

*Proof.* From Theorem 6, the mean gradient with respect to $\ell$ is

$$\nabla_\ell J_T = -2 \, \mathrm{diag}\Big( \Sigma \sum_{t=0}^{T-1} \gamma^t (Q_a + \gamma B^\top \Lambda_{t+1} B) \Big).$$

Define

$$M := \sum_{t=0}^{T-1} \gamma^t (Q_a + \gamma B^\top \Lambda_{t+1} B).$$

Then $\nabla_\ell J_T = 0$ is equivalent to $\mathrm{diag}(\Sigma M) = 0$.

As $\Sigma = \mathrm{diag}(\sigma^2)$ is diagonal, for each $i \in [m]$,

$$0 = (\Sigma M)_{ii} = \Sigma_{ii} M_{ii}.$$

Since $Q_a \succ 0$ and $\Lambda_{t+1} \succeq 0$, we have $Q_a + \gamma B^\top \Lambda_{t+1} B \succ 0$ for every $t$, hence $M \succ 0$ and in particular $M_{ii} > 0$ for all $i$. Therefore $\Sigma_{ii} = 0$ for all $i$, i.e. $\Sigma = \mathbf{0}$.

Conversely, if $\Sigma = \mathbf{0}$, then $\nabla_\ell J_T = 0$ follows immediately from the formula above. $\qquad\square$

# J. Proof of Theorem 8

*Proof.* Recall the policy gradient estimators

$$\widehat{G}_K = \sum_{t=0}^{T-1} \Sigma^{-1} \varepsilon_t s_t^\top R(\tau), \qquad \widehat{G}_\ell = \sum_{t=0}^{T-1} \Big( \Sigma^{-1} (\varepsilon_t \odot \varepsilon_t) - \mathbf{1} \Big) R(\tau),$$

and the stacked noise $\bar{\varepsilon} := (\varepsilon_0^\top, \ldots, \varepsilon_{T-1}^\top)^\top \sim \mathcal{N}(0, \bar{\Sigma})$ with $\bar{\Sigma} := I_T \otimes \Sigma$.

Using Cauchy's inequality on each Frobenius entry, let $\eta_t = \Sigma^{-1}\varepsilon_t$.

$$\|\widehat{G}_K\|_{\mathrm{F}}^2 = \Big\| \sum_{t=0}^{T-1} \Sigma^{-1}\varepsilon_t s_t^\top \Big\|_{\mathrm{F}}^2 R(\tau)^2 = \sum_{i,j} \Big( \sum_{t=0}^{T-1} \eta_{t,i} s_{t,j} \Big)^2 R(\tau)^2$$

$$\leq \sum_{i,j} \Big( \sum_{t=0}^{T-1} \eta_{t,i}^2 \Big) \Big( \sum_{t=0}^{T-1} s_{t,j}^2 \Big) R(\tau)^2 = \Big( \sum_{t=0}^{T-1} \|\eta_t\|_2^2 \Big) \Big( \sum_{t=0}^{T-1} \|s_t\|_2^2 \Big) R(\tau)^2.$$

Using stacked $\bar{\varepsilon}$ and $\bar{s}$ we obtain

$$\mathrm{Var}_{\mathrm{Fro}}(\widehat{G}_K) \leq \mathbb{E}[\|\widehat{G}_K\|_{\mathrm{F}}^2] \leq \mathbb{E}[\|\bar{\Sigma}^{-1}\bar{\varepsilon}\|_2^2 \, \|\bar{s}\|_2^2 \, R(\tau)^2]. \tag{66}$$

By AM-GM inequality, we have the decomposition

$$\mathrm{Var}_{\mathrm{Fro}}(\widehat{G}_K) \leq 2 \underbrace{\mathbb{E}[\|\bar{\Sigma}^{-1}\bar{\varepsilon}\|_2^2 \|\mathcal{F}_S s_0\|_2^2 R(\tau)^2]}_{(\star)} + 2 \underbrace{\mathbb{E}[\|\bar{\Sigma}^{-1}\bar{\varepsilon}\|_2^2 \|\mathcal{F}_E \bar{\varepsilon}\|_2^2 R(\tau)^2]}_{(\dagger)}. \tag{67}$$

For the $\ell$ part, define the per-step score term

$$g_t := \Sigma^{-1}(\varepsilon_t \odot \varepsilon_t) - \mathbf{1} \in \mathbb{R}^m, \qquad \widehat{G}_\ell = \sum_{t=0}^{T-1} g_t R(\tau).$$

By Cauchy–Schwarz (or $\|\sum_{t=0}^{T-1} v_t\|_2^2 \leq T \sum_{t=0}^{T-1} \|v_t\|_2^2$),

$$\mathbb{E}[\|\widehat{G}_\ell\|_2^2] \leq T \, \mathbb{E}\Big[ \sum_{t=0}^{T-1} \|g_t\|_2^2 R(\tau)^2 \Big]. \tag{68}$$

We next bound $\sum_{t=0}^{T-1} \|g_t\|_2^2$ by a polynomial in $\bar{\varepsilon}$. Since $\Sigma$ is diagonal, write $\varepsilon_t = \Sigma^{1/2}\xi_t$ with $\xi_t \sim \mathcal{N}(0, I_m)$ i.i.d., so that

$$g_t = \Sigma^{-1}(\varepsilon_t \odot \varepsilon_t) - \mathbf{1} = (\xi_t \odot \xi_t) - \mathbf{1}, \qquad \|g_t\|_2^2 = \sum_{i=1}^m (\xi_{t,i}^2 - 1)^2.$$

Using $(a-1)^2 \leq 2a^2 + 2$ with $a = \xi_{t,i}^2 \geq 0$ gives

$$(\xi_{t,i}^2 - 1)^2 \leq 2\xi_{t,i}^4 + 2 \quad \Rightarrow \quad \|g_t\|_2^2 \leq 2\sum_{i=1}^m \xi_{t,i}^4 + 2m \leq 2\|\xi_t\|_2^4 + 2m.$$

Summing over $t$ yields

$$\sum_{t=0}^{T-1} \|g_t\|_2^2 \leq 2 \sum_{t=0}^{T-1} \|\xi_t\|_2^4 + 2mT.$$

Using $\|\xi_t\|_2^2 \geq 0$, we have $\sum_{t=0}^{T-1} \|\xi_t\|_2^4 \leq \Big( \sum_{t=0}^{T-1} \|\xi_t\|_2^2 \Big)^2 = \|\bar{\xi}\|_2^4$.

Finally, since $\bar{\varepsilon} = (I_T \otimes \Sigma^{1/2})\bar{\xi}$ and $\bar{\Sigma} = I_T \otimes \Sigma$, we have $\bar{\Sigma}^{-1/2}\bar{\varepsilon} = \bar{\xi}$, so $\|\bar{\xi}\|_2^4 = \|\bar{\Sigma}^{-1/2}\bar{\varepsilon}\|_2^4$. Therefore,

$$\sum_{t=0}^{T-1} \|g_t\|_2^2 \leq 2\|\bar{\Sigma}^{-1/2}\bar{\varepsilon}\|_2^4 + 2mT.$$

Plugging into (68) gives

$$\mathbb{E}[\|\widehat{G}_\ell\|_2^2] \leq T \, \mathbb{E}\Big[ \big(2\|\bar{\Sigma}^{-1/2}\bar{\varepsilon}\|_2^4 + 2mT\big) R(\tau)^2 \Big].$$

Assume the standard multi-step quadratic decomposition of the return

$$R(\tau) = -(x + 2y + z), \qquad x := s_0^\top \overline{M}_{ss} s_0, \quad y := s_0^\top \overline{M}_{se} \bar{\varepsilon}, \quad z := \bar{\varepsilon}^\top \overline{M}_{ee} \bar{\varepsilon},$$

**Proof of $K$-part expansion.** Define

$$S := \mathcal{F}_S^\top \mathcal{F}_S, \qquad E := \mathcal{F}_E^\top \mathcal{F}_E,$$

**Step 1: expand ($\star$).** Since $\|\mathcal{F}_S s_0\|_2^2 = s_0^\top S s_0$ and $R(\tau) = -(x + 2y + z)$,

$$(\star) = \mathbb{E}\left[ (\bar{\varepsilon}^\top \overline{\Sigma}^{-2} \bar{\varepsilon}) \, (s_0^\top S s_0) \, (x + 2y + z)^2 \right].$$

Expanding $(x + 2y + z)^2 = x^2 + z^2 + 2xz + 4y^2 + 4xy + 4yz$ and using independence $s_0 \perp\!\!\!\perp \bar{\varepsilon}$ and odd-moment cancellation in $\bar{\varepsilon}$ (which removes the $xy$ and $yz$ terms), we get

$$(\star) = E_{1,K}^{(\star)} + E_{2,K}^{(\star)} + E_{3,K}^{(\star)} + E_{4,K}^{(\star)},$$

with

$$
\begin{aligned}
E_{1,K}^{(\star)} &:= \mathbb{E}[\bar{\varepsilon}^\top \overline{\Sigma}^{-2} \bar{\varepsilon}] \, \mathbb{E}\left[ (s_0^\top S s_0) \, x^2 \right], \\
E_{2,K}^{(\star)} &:= \mathbb{E}[s_0^\top S s_0] \, \mathbb{E}\left[ (\bar{\varepsilon}^\top \overline{\Sigma}^{-2} \bar{\varepsilon}) \, z^2 \right], \\
E_{3,K}^{(\star)} &:= 2 \, \mathbb{E}\left[ (s_0^\top S s_0) \, x \right] \, \mathbb{E}\left[ (\bar{\varepsilon}^\top \overline{\Sigma}^{-2} \bar{\varepsilon}) \, z \right], \\
E_{4,K}^{(\star)} &:= 4 \, \mathbb{E}\left[ (s_0^\top S s_0) \, \mathbb{E}\left[ (\bar{\varepsilon}^\top \overline{\Sigma}^{-2} \bar{\varepsilon}) \, y^2 \mid s_0 \right] \right].
\end{aligned}
$$

Using Lemma 2,

$$\mathbb{E}[\bar{\varepsilon}^\top \overline{\Sigma}^{-2} \bar{\varepsilon}] = \mathrm{IS}_{\overline{\Sigma}}(\overline{\Sigma}^{-2}), \qquad \mathbb{E}[s_0^\top S s_0] = \mathrm{IS}_{\Sigma_0}(S),$$

$$\mathbb{E}\left[ (s_0^\top S s_0) x \right] = \mathrm{IS}_{\Sigma_0}(S, \overline{M}_{ss}), \qquad \mathbb{E}\left[ (s_0^\top S s_0) x^2 \right] = \mathrm{IS}_{\Sigma_0}(S, \overline{M}_{ss}, \overline{M}_{ss}),$$

and since $z = \bar{\varepsilon}^\top \overline{M}_{ee} \bar{\varepsilon}$,

$$\mathbb{E}\left[ (\bar{\varepsilon}^\top \overline{\Sigma}^{-2} \bar{\varepsilon}) z \right] = \mathrm{IS}_{\overline{\Sigma}}(\overline{\Sigma}^{-2}, \overline{M}_{ee}), \qquad \mathbb{E}\left[ (\bar{\varepsilon}^\top \overline{\Sigma}^{-2} \bar{\varepsilon}) z^2 \right] = \mathrm{IS}_{\overline{\Sigma}}(\overline{\Sigma}^{-2}, \overline{M}_{ee}, \overline{M}_{ee}).$$

For $E_{4,K}^{(\star)}$, note $y = s_0^\top \overline{M}_{se} \bar{\varepsilon}$ implies

$$y^2 = \bar{\varepsilon}^\top X(s_0) \bar{\varepsilon}, \qquad X(s_0) := \overline{M}_{se}^\top s_0 s_0^\top \overline{M}_{se}.$$

Conditioning on $s_0$ and applying Lemma 2 with $k = 2$,

$$\mathbb{E}_{\bar{\varepsilon}}\left[ (\bar{\varepsilon}^\top \overline{\Sigma}^{-2} \bar{\varepsilon}) y^2 \mid s_0 \right] = \mathrm{IS}_{\overline{\Sigma}}(\overline{\Sigma}^{-2}, X(s_0)).$$

Define

$$U := \overline{M}_{se} \overline{M}_{se}^\top, \qquad V := \overline{M}_{se} \overline{\Sigma} \overline{M}_{se}^\top.$$

Then $\mathrm{tr}(X(s_0)) = s_0^\top U s_0$ and $\mathrm{tr}(\overline{\Sigma} X(s_0)) = s_0^\top V s_0$, and since $\overline{\Sigma} \overline{\Sigma}^{-2} \overline{\Sigma} = I$, we obtain

$$\mathrm{IS}_{\overline{\Sigma}}(\overline{\Sigma}^{-2}, X(s_0)) = \mathrm{IS}_{\overline{\Sigma}}(\overline{\Sigma}^{-2}) \, (s_0^\top V s_0) + 2 \, (s_0^\top U s_0).$$

Therefore,

$$E_{4,K}^{(\star)} = 4 \left( \mathrm{IS}_{\overline{\Sigma}}(\overline{\Sigma}^{-2}) \, \mathrm{IS}_{\Sigma_0}(S, V) + 2 \, \mathrm{IS}_{\Sigma_0}(S, U) \right).$$

Altogether,

$$
\begin{aligned}
(\star) = {}& \mathrm{IS}_{\overline{\Sigma}}(\overline{\Sigma}^{-2}) \, \mathrm{IS}_{\Sigma_0}(S, \overline{M}_{ss}, \overline{M}_{ss}) + \mathrm{IS}_{\Sigma_0}(S) \, \mathrm{IS}_{\overline{\Sigma}}(\overline{\Sigma}^{-2}, \overline{M}_{ee}, \overline{M}_{ee}) \\
& + 2 \, \mathrm{IS}_{\Sigma_0}(S, \overline{M}_{ss}) \, \mathrm{IS}_{\overline{\Sigma}}(\overline{\Sigma}^{-2}, \overline{M}_{ee}) + 4 \left( \mathrm{IS}_{\overline{\Sigma}}(\overline{\Sigma}^{-2}) \, \mathrm{IS}_{\Sigma_0}(S, V) + 2 \, \mathrm{IS}_{\Sigma_0}(S, U) \right).
\end{aligned}
\tag{69}
$$

**Step 2: expand** ($\dagger$). Since $\|\mathcal{F}_E\bar\varepsilon\|_2^2 = \bar\varepsilon^\top E\bar\varepsilon$,

$$(\dagger) = \mathbb{E}\left[(\bar\varepsilon^\top \overline{\Sigma}^{-2}\bar\varepsilon)\,(\bar\varepsilon^\top E\bar\varepsilon)\,(x + 2y + z)^2\right].$$

As before, odd-moment cancellation removes $xy$ and $yz$, yielding

$$(\dagger) = E_{1,K}^{(\dagger)} + E_{2,K}^{(\dagger)} + E_{3,K}^{(\dagger)} + E_{4,K}^{(\dagger)},$$

with

$$\begin{aligned}
E_{1,K}^{(\dagger)} &:= \mathbb{E}\left[(\bar\varepsilon^\top \overline{\Sigma}^{-2}\bar\varepsilon)(\bar\varepsilon^\top E\bar\varepsilon)\right]\,\mathbb{E}[x^2],\\
E_{2,K}^{(\dagger)} &:= \mathbb{E}\left[(\bar\varepsilon^\top \overline{\Sigma}^{-2}\bar\varepsilon)(\bar\varepsilon^\top E\bar\varepsilon)\,z^2\right],\\
E_{3,K}^{(\dagger)} &:= 2\,\mathbb{E}[x]\,\mathbb{E}\left[(\bar\varepsilon^\top \overline{\Sigma}^{-2}\bar\varepsilon)(\bar\varepsilon^\top E\bar\varepsilon)\,z\right],\\
E_{4,K}^{(\dagger)} &:= 4\,\mathbb{E}\left[\mathbb{E}\left[(\bar\varepsilon^\top \overline{\Sigma}^{-2}\bar\varepsilon)(\bar\varepsilon^\top E\bar\varepsilon)\,y^2\,\big|\,s_0\right]\right].
\end{aligned}$$

Rewrite each term:

$$\mathbb{E}[x] = \mathrm{IS}_{\Sigma_0}(\overline{M}_{ss}), \qquad \mathbb{E}[x^2] = \mathrm{IS}_{\Sigma_0}(\overline{M}_{ss}, \overline{M}_{ss}),$$

$$\mathbb{E}\left[(\bar\varepsilon^\top \overline{\Sigma}^{-2}\bar\varepsilon)(\bar\varepsilon^\top E\bar\varepsilon)\right] = \mathrm{IS}_{\overline{\Sigma}}(\overline{\Sigma}^{-2}, E),$$

$$\mathbb{E}\left[(\bar\varepsilon^\top \overline{\Sigma}^{-2}\bar\varepsilon)(\bar\varepsilon^\top E\bar\varepsilon)\,z\right] = \mathrm{IS}_{\overline{\Sigma}}(\overline{\Sigma}^{-2}, E, \overline{M}_{ee}), \qquad \mathbb{E}\left[(\bar\varepsilon^\top \overline{\Sigma}^{-2}\bar\varepsilon)(\bar\varepsilon^\top E\bar\varepsilon)\,z^2\right] = \mathrm{IS}_{\overline{\Sigma}}(\overline{\Sigma}^{-2}, E, \overline{M}_{ee}, \overline{M}_{ee}).$$

For the mixed term, with $y^2 = \bar\varepsilon^\top X(s_0)\bar\varepsilon$,

$$\mathbb{E}_{\bar\varepsilon}\left[(\bar\varepsilon^\top \overline{\Sigma}^{-2}\bar\varepsilon)(\bar\varepsilon^\top E\bar\varepsilon)\,y^2 \mid s_0\right] = \mathrm{IS}_{\overline{\Sigma}}(\overline{\Sigma}^{-2}, E, X(s_0)),$$

For the mixed term, recall $X(s_0) = \overline{M}_{se}^\top s_0 s_0^\top \overline{M}_{se}$ and define

$$U := \overline{M}_{se}\overline{M}_{se}^\top, \qquad V := \overline{M}_{se}\overline{\Sigma}\,\overline{M}_{se}^\top, \qquad W := \overline{M}_{se}\,\overline{\Sigma}\,E\,\overline{\Sigma}\,\overline{M}_{se}^\top, \qquad Z := \mathbf{sym}(\overline{M}_{se}\,E\,\overline{\Sigma}\,\overline{M}_{se}^\top).$$

First, by Lemma 2 ($k = 3$) with $\Omega = \overline{\Sigma}$, $A = \overline{\Sigma}^{-2}$, $B = E$, $C = X(s_0)$,

$$\begin{aligned}
\mathrm{IS}_{\overline{\Sigma}}(\overline{\Sigma}^{-2}, E, X(s_0)) = {} & \mathrm{tr}(\overline{\Sigma}^{-1})\,\mathrm{tr}(\overline{\Sigma}E)\,\mathrm{tr}(\overline{\Sigma}X(s_0)) + 2\,\mathrm{tr}(E)\,\mathrm{tr}(\overline{\Sigma}X(s_0)) + 2\,\mathrm{tr}(\overline{\Sigma}E)\,\mathrm{tr}(X(s_0))\\
& + 2\,\mathrm{tr}(\overline{\Sigma}^{-1})\,\mathrm{tr}(\overline{\Sigma}E\overline{\Sigma}X(s_0)) + 8\,\mathrm{tr}(E\overline{\Sigma}X(s_0)),
\end{aligned}$$

and using $\overline{\Sigma}\,\overline{\Sigma}^{-2}\,\overline{\Sigma} = I$ together with $\mathrm{tr}(\overline{\Sigma}X(s_0)) = s_0^\top V s_0$, $\mathrm{tr}(X(s_0)) = s_0^\top U s_0$, $\mathrm{tr}(\overline{\Sigma}E\overline{\Sigma}X(s_0)) = s_0^\top W s_0$, and $\mathrm{tr}(E\overline{\Sigma}X(s_0)) = s_0^\top Z s_0$, we obtain

$$\mathrm{IS}_{\overline{\Sigma}}(\overline{\Sigma}^{-2}, E, X(s_0)) = \mathrm{IS}_{\overline{\Sigma}}(\overline{\Sigma}^{-2}, E)\,(s_0^\top V s_0) + 2\,\mathrm{IS}_{\overline{\Sigma}}(E)\,(s_0^\top U s_0) + 2\,\mathrm{IS}_{\overline{\Sigma}}(\overline{\Sigma}^{-2})\,(s_0^\top W s_0) + 8\,(s_0^\top Z s_0).$$

Taking expectation over $s_0 \sim \mathcal{N}(0, \Sigma_0)$ gives

$$\mathbb{E}_{s_0}\left[\mathrm{IS}_{\overline{\Sigma}}(\overline{\Sigma}^{-2}, E, X(s_0))\right] = \mathrm{IS}_{\overline{\Sigma}}(\overline{\Sigma}^{-2}, E)\,\mathrm{IS}_{\Sigma_0}(V) + 2\,\mathrm{IS}_{\overline{\Sigma}}(E)\,\mathrm{IS}_{\Sigma_0}(U) + 2\,\mathrm{IS}_{\overline{\Sigma}}(\overline{\Sigma}^{-2})\,\mathrm{IS}_{\Sigma_0}(W) + 8\,\mathrm{IS}_{\Sigma_0}(Z).$$

Consequently,

$$E_{4,K}^{(\dagger)} = 4\,\mathrm{IS}_{\overline{\Sigma}}(\overline{\Sigma}^{-2}, E)\,\mathrm{IS}_{\Sigma_0}(V) + 8\,\mathrm{IS}_{\overline{\Sigma}}(E)\,\mathrm{IS}_{\Sigma_0}(U) + 8\,\mathrm{IS}_{\overline{\Sigma}}(\overline{\Sigma}^{-2})\,\mathrm{IS}_{\Sigma_0}(W) + 32\,\mathrm{IS}_{\Sigma_0}(Z).$$

hence

$$\begin{aligned}
(\dagger) = {} & \mathrm{IS}_{\overline{\Sigma}}(\overline{\Sigma}^{-2}, E)\,\mathrm{IS}_{\Sigma_0}(\overline{M}_{ss}, \overline{M}_{ss}) + \mathrm{IS}_{\overline{\Sigma}}(\overline{\Sigma}^{-2}, E, \overline{M}_{ee}, \overline{M}_{ee})\\
& + 2\,\mathrm{IS}_{\Sigma_0}(\overline{M}_{ss})\,\mathrm{IS}_{\overline{\Sigma}}(\overline{\Sigma}^{-2}, E, \overline{M}_{ee}) + 4\,\mathbb{E}_{s_0}\left[\mathrm{IS}_{\overline{\Sigma}}(\overline{\Sigma}^{-2}, E, X(s_0))\right].
\end{aligned} \tag{70}$$

Combining (69), and (70) yields the desired multi-step expansion of $\mathbb{E}[\|\widehat{G}_K\|_{\mathrm{F}}^2]$.

**Proof of $\ell$-part expansion.** Then

$$\|\bar{\Sigma}^{-1/2}\bar{\varepsilon}\|_2^4 = (\bar{\varepsilon}^\top\bar{\Sigma}^{-1}\bar{\varepsilon})^2, \qquad R(\tau)^2 = (x + 2y + z)^2.$$

As above, odd-moment cancellation removes $xy$ and $yz$, so

$$R(\tau)^2 = x^2 + z^2 + 2xz + 4y^2.$$

Therefore

$$\mathbb{E}\Big[(2\|\bar{\Sigma}^{-1/2}\bar{\varepsilon}\|_2^4 + 2mT)\,R(\tau)^2\Big] = 2\sum_{i=1}^{4} A_i + 2mT\sum_{i=1}^{4} B_i,$$

where

$$
\begin{aligned}
A_1 &:= \mathbb{E}\big[(\bar{\varepsilon}^\top\bar{\Sigma}^{-1}\bar{\varepsilon})^2\,x^2\big] = \mathbb{E}[x^2]\,\mathbb{E}\big[(\bar{\varepsilon}^\top\bar{\Sigma}^{-1}\bar{\varepsilon})^2\big], \\
A_2 &:= \mathbb{E}\big[(\bar{\varepsilon}^\top\bar{\Sigma}^{-1}\bar{\varepsilon})^2\,z^2\big], \\
A_3 &:= 2\,\mathbb{E}\big[(\bar{\varepsilon}^\top\bar{\Sigma}^{-1}\bar{\varepsilon})^2\,xz\big] = 2\,\mathbb{E}[x]\,\mathbb{E}\big[(\bar{\varepsilon}^\top\bar{\Sigma}^{-1}\bar{\varepsilon})^2\,z\big], \\
A_4 &:= 4\,\mathbb{E}\big[(\bar{\varepsilon}^\top\bar{\Sigma}^{-1}\bar{\varepsilon})^2\,y^2\big] = 4\,\mathbb{E}_{s_0}\big[\mathrm{IS}_{\bar{\Sigma}}(\bar{\Sigma}^{-1},\bar{\Sigma}^{-1},X(s_0))\big],
\end{aligned}
$$

and

$$B_1 := \mathbb{E}[x^2], \qquad B_2 := \mathbb{E}[z^2], \qquad B_3 := 2\,\mathbb{E}[xz] = 2\,\mathbb{E}[x]\mathbb{E}[z], \qquad B_4 := 4\,\mathbb{E}[y^2].$$

Each term can be written using IS shorthand:

$$
\begin{aligned}
\mathbb{E}[x] &= \mathrm{IS}_{\Sigma_0}(\overline{M}_{ss}), \qquad \mathbb{E}[x^2] = \mathrm{IS}_{\Sigma_0}(\overline{M}_{ss}, \overline{M}_{ss}), \\
\mathbb{E}[z] &= \mathrm{IS}_{\bar{\Sigma}}(\overline{M}_{ee}), \qquad \mathbb{E}[z^2] = \mathrm{IS}_{\bar{\Sigma}}(\overline{M}_{ee}, \overline{M}_{ee}), \\
\mathbb{E}[y^2] &= \mathrm{IS}_{\Sigma_0}(V), \quad \text{where } V := \overline{M}_{se}\bar{\Sigma}\overline{M}_{se}^\top.
\end{aligned}
$$

Moreover,

$$\mathbb{E}\big[(\bar{\varepsilon}^\top\bar{\Sigma}^{-1}\bar{\varepsilon})^2\big] = \mathrm{IS}_{\bar{\Sigma}}(\bar{\Sigma}^{-1}, \bar{\Sigma}^{-1}),$$

$$\mathbb{E}\big[(\bar{\varepsilon}^\top\bar{\Sigma}^{-1}\bar{\varepsilon})^2\,z\big] = \mathrm{IS}_{\bar{\Sigma}}(\bar{\Sigma}^{-1}, \bar{\Sigma}^{-1}, \overline{M}_{ee}), \qquad \mathbb{E}\big[(\bar{\varepsilon}^\top\bar{\Sigma}^{-1}\bar{\varepsilon})^2\,z^2\big] = \mathrm{IS}_{\bar{\Sigma}}(\bar{\Sigma}^{-1}, \bar{\Sigma}^{-1}, \overline{M}_{ee}, \overline{M}_{ee}).$$

**Term $A_4$ ($y^2$-part).** Applying Lemma 2 ($k = 3$) with $\Omega = \bar{\Sigma}$, $A = B = \bar{\Sigma}^{-1}$, $C = X(s_0)$ and using $\bar{\Sigma}\,\bar{\Sigma}^{-1} = I$, we obtain

$$\mathrm{IS}_{\bar{\Sigma}}(\bar{\Sigma}^{-1}, \bar{\Sigma}^{-1}, X(s_0)) = (d^2 + 6d + 8)\,\mathrm{tr}(\bar{\Sigma}\,X(s_0)) = (d+2)(d+4)\,\mathrm{tr}(\bar{\Sigma}\,X(s_0)), \qquad d := mT.$$

Moreover,

$$\mathrm{tr}(\bar{\Sigma}\,X(s_0)) = \mathrm{tr}(s_0 s_0^\top \overline{M}_{se}\bar{\Sigma}\,\overline{M}_{se}^\top) = s_0^\top V s_0, \qquad V := \overline{M}_{se}\bar{\Sigma}\,\overline{M}_{se}^\top,$$

hence $\mathbb{E}_{s_0}[\mathrm{tr}(\bar{\Sigma}\,X(s_0))] = \mathrm{tr}(\Sigma_0 V) = \mathrm{IS}_{\Sigma_0}(V)$. Therefore,

$$A_4 = 4(d+2)(d+4)\,\mathrm{IS}_{\Sigma_0}(V), \qquad d = mT.$$

Hence,

$$
\begin{aligned}
\mathbb{E}\Big[\|\widehat{G}_\ell\|_2^2\Big] \leq T\bigg( & 2\Big[\mathrm{IS}_{\Sigma_0}(\overline{M}_{ss}, \overline{M}_{ss})\,\mathrm{IS}_{\bar{\Sigma}}(\bar{\Sigma}^{-1}, \bar{\Sigma}^{-1}) + \mathrm{IS}_{\bar{\Sigma}}(\bar{\Sigma}^{-1}, \bar{\Sigma}^{-1}, \overline{M}_{ee}, \overline{M}_{ee}) \\
& + 2\,\mathrm{IS}_{\Sigma_0}(\overline{M}_{ss})\,\mathrm{IS}_{\bar{\Sigma}}(\bar{\Sigma}^{-1}, \bar{\Sigma}^{-1}, \overline{M}_{ee}) + 4(d+2)(d+4)\,\mathrm{IS}_{\Sigma_0}(V)\Big] \\
& + 2mT\Big[\mathrm{IS}_{\Sigma_0}(\overline{M}_{ss}, \overline{M}_{ss}) + \mathrm{IS}_{\bar{\Sigma}}(\overline{M}_{ee}, \overline{M}_{ee}) + 2\,\mathrm{IS}_{\Sigma_0}(\overline{M}_{ss})\,\mathrm{IS}_{\bar{\Sigma}}(\overline{M}_{ee}) \\
& + 4\,\mathrm{IS}_{\Sigma_0}(V)\Big]\bigg),
\end{aligned}
\tag{71}
$$

where $X(s_0) := \overline{M}_{se}^\top s_0 s_0^\top \overline{M}_{se}$ and $V := \overline{M}_{se}\bar{\Sigma}\overline{M}_{se}^\top$.

$\square$

## K. Proof of Theorem 9

*Proof.* Recall that $\|\mathcal{F}_S\|_2^2 = \lambda_{\max}(\mathcal{F}_S^\top \mathcal{F}_S)$. Since

$$\mathcal{F}_S = \begin{bmatrix} I & F & \cdots & F^{T-1} \end{bmatrix}^\top,$$

we have

$$\mathcal{F}_S^\top \mathcal{F}_S = \sum_{t=0}^{T-1} (F^t)^\top F^t.$$

**Lower bound.** For any fixed $t \in \{0, \ldots, T-1\}$,

$$\mathcal{F}_S^\top \mathcal{F}_S = \sum_{k=0}^{T-1} (F^k)^\top F^k \succeq (F^t)^\top F^t.$$

Taking $\lambda_{\max}(\cdot)$ preserves the Loewner order, hence

$$\|\mathcal{F}_S\|_2^2 = \lambda_{\max}(\mathcal{F}_S^\top \mathcal{F}_S) \geq \lambda_{\max}((F^t)^\top F^t) = \|F^t\|_2^2.$$

Maximizing over $t$ yields $\max_{0 \leq t \leq T-1} \|F^t\|_2^2 \leq \|\mathcal{F}_S\|_2^2$.

**Upper bound.** For any unit vector $v$,

$$v^\top (\mathcal{F}_S^\top \mathcal{F}_S) v = \sum_{t=0}^{T-1} \|F^t v\|_2^2 \leq \sum_{t=0}^{T-1} \|F^t\|_2^2 \|v\|_2^2 = \sum_{t=0}^{T-1} \|F^t\|_2^2.$$

Taking the supremum over $\|v\|_2 = 1$ gives

$$\|\mathcal{F}_S\|_2^2 = \sup_{\|v\|_2=1} v^\top (\mathcal{F}_S^\top \mathcal{F}_S) v \leq \sum_{t=0}^{T-1} \|F^t\|_2^2.$$

$\square$

## L. Proof of Proposition 10

*Proof.* **(i) Polynomial estimator.** Since $P(\cdot, \cdot)$ and $\Phi(\cdot)$ are polynomial and $a_t = \Phi(s_t)^\top \theta + \varepsilon_t$, a simple induction shows that for each $t \leq T$, $s_t$ and $a_t$ are vector-valued polynomials in $\xi := (s_0, \varepsilon_0, \ldots, \varepsilon_{T-1})$. Because $r(\cdot, \cdot)$ is polynomial, the return $R(\tau) = \sum_{t=0}^{T-1} \gamma^t r(s_{t+1}, a_t)$ is also a polynomial in $\xi$. Moreover, for the Gaussian policy,

$$\nabla_\theta \log \pi(a_t \mid s_t) = \Phi(s_t)\Sigma^{-1}\varepsilon_t, \qquad \nabla_\ell \log \pi(a_t \mid s_t) = \Sigma^{-1}(\varepsilon_t \odot \varepsilon_t) - \mathbf{1},$$

so $\nabla_\theta \log \pi(a_t \mid s_t)$ and $\nabla_\ell \log \pi(a_t \mid s_t)$ are polynomials in $\xi$. Therefore $\widehat{G}_\theta = \sum_{t=0}^{T-1} \gamma^t R(\tau) \nabla_\theta \log \pi(a_t \mid s_t)$ and $\widehat{G}_\ell = \sum_{t=0}^{T-1} \gamma^t R(\tau) \nabla_\ell \log \pi(a_t \mid s_t)$ are polynomials in $\xi$.

**Indexing convention.** We use $j$ to index vector coordinates: for $\widehat{G}_\theta \in \mathbb{R}^d$ and $\widehat{G}_\ell \in \mathbb{R}^m$, $\widehat{G}_{\theta,j}$ denotes the $j$-th component of $\widehat{G}_\theta$ (and similarly $\widehat{G}_{\ell,j}$). For monomials, $\alpha \in \mathbb{N}^{n+mT}$ is a multi-index and $\xi^\alpha := \prod_{i=1}^{n+mT} \xi_i^{\alpha_i}$.

**(ii) Exact moments via Isserlis.** By construction, $\xi$ is a centered jointly Gaussian vector with covariance $\mathrm{Cov}(\xi) = \mathrm{diag}(\Sigma_0, \Sigma, \ldots, \Sigma)$. Since $\widehat{G}_\theta$ (and $\widehat{G}_\ell$) are polynomials in $\xi$, each admits a finite monomial expansion $\widehat{G}_{\theta,j}(\xi) = \sum_\alpha c_{j,\alpha} \xi^\alpha$ (and similarly for $\widehat{G}_{\ell,j}$). Therefore $\mathbb{E}[\widehat{G}_\theta]$ and $\mathbb{E}[\|\widehat{G}_\theta\|_\mathrm{F}^2]$ (and likewise for $\widehat{G}_\ell$) reduce to finitely many Gaussian monomial moments $\mathbb{E}[\xi^\alpha]$. These moments are given exactly by Isserlis' (Wick) theorem: all odd-order moments vanish, and each even-order moment is a sum over pairings of covariance entries determined by $\mathrm{Cov}(\xi)$. Thus $\mathbb{E}[\widehat{G}_\theta]$, $\mathbb{E}[\widehat{G}_\ell]$, $\mathbb{E}[\|\widehat{G}_\theta\|_\mathrm{F}^2]$, and $\mathbb{E}[\|\widehat{G}_\ell\|_\mathrm{F}^2]$ are exactly computable from $\mathrm{Cov}(\xi)$. $\square$

## M. Proof of Theorem 11

*Proof.* We prove (24) and (25) separately.

**Part (i): proof of** (24). Let

$$u(\tau) := \sum_{t=0}^{T-1} J_\theta(s_t)^\top \Sigma^{-1} \varepsilon_t, \qquad \text{so that} \qquad \widehat{G}_\theta(\tau) = R(\tau)\, u(\tau).$$

*Step 1 (sum-of-norms).* Using $\|\sum_{t=0}^{T-1} x_t\|_2^2 \le T \sum_{t=0}^{T-1} \|x_t\|_2^2$, with $x_t = J_\theta(s_t)^\top \Sigma^{-1} \varepsilon_t$, we have

$$\|u(\tau)\|_2^2 \le T \sum_{t=0}^{T-1} \|J_\theta(s_t)^\top \Sigma^{-1} \varepsilon_t\|_2^2.$$

Hence

$$\mathbb{E}\Big[\|\widehat{G}_\theta(\tau)\|_2^2 \,\Big|\, s_0\Big] = \mathbb{E}\big[R(\tau)^2 \|u(\tau)\|_2^2 \,\big|\, s_0\big]$$
$$\le T \sum_{t=0}^{T-1} \mathbb{E}\big[R(\tau)^2 \|J_\theta(s_t)^\top \Sigma^{-1} \varepsilon_t\|_2^2 \,\big|\, s_0\big]. \tag{72}$$

*Step 2 (operator/Frobenius bound).* For each $t$,

$$\|J_\theta(s_t)^\top \Sigma^{-1} \varepsilon_t\|_2 \le \|J_\theta(s_t)\|_{\mathrm{F}} \, \|\Sigma^{-1}\|_2 \, \|\varepsilon_t\|_2,$$

Plugging into (72) gives

$$\mathbb{E}\Big[\|\widehat{G}_\theta(\tau)\|_2^2 \,\Big|\, s_0\Big] \le T\|\Sigma^{-1}\|_2^2 \sum_{t=0}^{T-1} \mathbb{E}\big[R(\tau)^2 \|J_\theta(s_t)\|_{\mathrm{F}}^2 \|\varepsilon_t\|_2^2 \,\big|\, s_0\big]. \tag{73}$$

*Step 3 (Cauchy–Schwarz).* For each $t$, apply Cauchy–Schwarz with $X = R(\tau)^2$ and $Y = \|J_\theta(s_t)\|_{\mathrm{F}}^2 \|\varepsilon_t\|_2^2$:

$$\mathbb{E}[XY \mid s_0] \le \sqrt{\mathbb{E}[X^2 \mid s_0]\, \mathbb{E}[Y^2 \mid s_0]} = \sqrt{\mathbb{E}[R(\tau)^4 \mid s_0]} \, \sqrt{\mathbb{E}[\|J_\theta(s_t)\|_{\mathrm{F}}^4 \|\varepsilon_t\|_2^4 \mid s_0]}.$$

Thus (73) implies

$$\mathbb{E}\Big[\|\widehat{G}_\theta(\tau)\|_2^2 \,\Big|\, s_0\Big] \le T\|\Sigma^{-1}\|_2^2 \sqrt{\mathbb{E}[R(\tau)^4 \mid s_0]} \sum_{t=0}^{T-1} \sqrt{\mathbb{E}[\|J_\theta(s_t)\|_{\mathrm{F}}^4 \|\varepsilon_t\|_2^4 \mid s_0]}. \tag{74}$$

*Step 4 (independence of $\varepsilon_t$ and $s_t$).* Since $s_t$ is a measurable function of $(s_0, \varepsilon_0, \ldots, \varepsilon_{t-1})$ and $\varepsilon_t$ is independent of $(\varepsilon_0, \ldots, \varepsilon_{t-1})$, we have $\varepsilon_t \perp s_t$ conditional on $s_0$. Therefore

$$\mathbb{E}[\|J_\theta(s_t)\|_{\mathrm{F}}^4 \|\varepsilon_t\|_2^4 \mid s_0] = \mathbb{E}[\|J_\theta(s_t)\|_{\mathrm{F}}^4 \mid s_0]\, \mathbb{E}[\|\varepsilon\|_2^4].$$

*Step 5 (Gaussian fourth moment).* For $\varepsilon \sim \mathcal{N}(0, \Sigma)$,

$$\mathbb{E}\|\varepsilon\|_2^4 = (\mathrm{tr}\,\Sigma)^2 + 2\,\mathrm{tr}(\Sigma^2).$$

Substituting the last two displays into (74) yields (24).

**Part (ii): proof of** (25). Define

$$q(\varepsilon_t) := \nabla_\ell \log \pi(a_t \mid s_t) = \Sigma^{-1}(\varepsilon_t \odot \varepsilon_t) - \mathbf{1} = ((\varepsilon_t \odot \varepsilon_t) \odot e^{-2\ell}) - \mathbf{1} \in \mathbb{R}^m,$$

so that

$$\widehat{G}_\ell(\tau) = R(\tau) \sum_{t=0}^{T-1} q(\varepsilon_t).$$

*Step 1 (sum-of-norms).* Using $\| \sum_{t=0}^{T-1} x_t \|_2^2 \le T \sum_{t=0}^{T-1} \|x_t\|_2^2$ with $x_t = q(\varepsilon_t)$:

$$\|\widehat{G}_\ell(\tau)\|_2^2 = R(\tau)^2 \Big\| \sum_{t=0}^{T-1} q(\varepsilon_t) \Big\|_2^2 \le R(\tau)^2 \, T \sum_{t=0}^{T-1} \|q(\varepsilon_t)\|_2^2.$$

Taking $\mathbb{E}[\cdot \mid s_0]$ and applying Cauchy–Schwarz termwise gives

$$\mathbb{E}[\|\widehat{G}_\ell(\tau)\|_2^2 \mid s_0] \le T \sum_{t=0}^{T-1} \mathbb{E}\big[R(\tau)^2 \|q(\varepsilon_t)\|_2^2 \mid s_0\big]$$

$$\le T \sum_{t=0}^{T-1} \sqrt{\mathbb{E}[R(\tau)^4 \mid s_0]} \, \sqrt{\mathbb{E}[\|q(\varepsilon_t)\|_2^4]}$$

$$= T^2 \sqrt{\mathbb{E}[R(\tau)^4 \mid s_0]} \, \sqrt{\mathbb{E}[\|q(\varepsilon)\|_2^4]}, \tag{75}$$

where the last line uses that $\varepsilon_t$ are i.i.d.

*Step 2 (exact fourth moment of the log-std score).* Assume $\ell = \log \sigma$ and $\Sigma = \mathrm{diag}\left(e^{2\ell}\right)$ is diagonal. Then $\varepsilon = \mathrm{diag}\left(e^\ell\right) z$ with $z \sim \mathcal{N}(0, I_m)$, and elementwise

$$q_i(\varepsilon) = \frac{\varepsilon_i^2}{e^{2\ell_i}} - 1 = z_i^2 - 1,$$

so the distribution of $q(\varepsilon)$ does not depend on $\ell$. Let $w_i := (z_i^2 - 1)^2$. Then

$$\|q(\varepsilon)\|_2^2 = \sum_{i=1}^m w_i, \qquad \|q(\varepsilon)\|_2^4 = \left( \sum_{i=1}^m w_i \right)^2 = \sum_{i=1}^m w_i^2 + 2 \sum_{1 \le i < j \le m} w_i w_j.$$

Using independence of $(z_i)$, we have $\mathbb{E}[w_i w_j] = \mathbb{E}[w_i]\mathbb{E}[w_j]$ for $i \ne j$. Moreover, for $z \sim \mathcal{N}(0,1)$,

$$\mathbb{E}[(z^2 - 1)^2] = 2, \qquad \mathbb{E}[(z^2 - 1)^4] = 60.$$

Therefore

$$\mathbb{E}\|q(\varepsilon)\|_2^4 = m \cdot 60 + 2 \binom{m}{2} \cdot (2 \cdot 2) = 4m(m + 14).$$

Plugging this into (75) yields

$$\mathbb{E}[\|\widehat{G}_\ell(\tau)\|_2^2 \mid s_0] \le 2T^2 \sqrt{m(m + 14)} \, \sqrt{\mathbb{E}[R(\tau)^4 \mid s_0]}.$$

$\square$

