# OpenReview forum: "Non-Uniform Noise-to-Signal Ratio in the REINFORCE Policy-Gradient Estimator"
_ICML.cc/2026/Conference — ICML 2026 regular_

### Official Review · Reviewer_KqCc · 2026-03-03

**Soundness:** 2
**Presentation:** 2
**Significance:** 2
**Originality:** 2
**Overall Recommendation:** 3
**Confidence:** 3

**Summary:**

The authors study the noise-to-signal ratio (NSR) of the REINFORCE estimator, a prototypical estimator for on-policy actor-only policy gradient (PG) methods.
In particular, the paper aims at advancing the understanding of the causes responsible for the instabilities or the slowing down of the learning process when using REINFORCE.
The authors approach this by studying the NSR, i.e., the variance of the estimator of the gradient normalized by the squared norm of the true gradient itself, in a setting where continuous state and action spaces are considered, and where it is considered to learn the policy variance besides the mean parameterization.
The authors show that the NSR of REINFORCE can be exactly characterized in LQG and polynomial systems with finite horizon and linear gaussian policies. For non-linear systems with gaussian noise policies the authors characterize the variance of the via an upper bound.
Finally, the authors empirically show that the NSR's landscape is non-uniform, in the sense that it explodes as much as the policy reaches determinism in the learning process.

**Compliance With Llm Reviewing Policy:**

Affirmed.

**Final Justification:**

After having read the other reviews and the rebuttals, I decided to increase my score to 3.
The authors clarified their contribution in my rebuttal and, in other rebuttals, they investigated various points (as the addition of baselines, of entropy regularization, the role of the batch size). My final concern was in the fact that the authors propose the NSR as a tool for studying training instabilities from an empirical standpoint. In the rebuttal, in response to my and the others' concerns, the authors included more experimental validation that is worth to include in the main paper. Finally, in response to the acknowledgements, the authors run more worth-to-include experiments on high-dimensional control tasks of the proposed phenomenon and two new theorems that seem to build a first link between the NSR/Variance and learning instabilities, which help in motivating the work.
However, I feel that at this stage the effort to complete the paper is more than a minor revision, but my judgement is borderline and I am open to discuss with other reviewers to reconsider my scoring.

**Key Questions For Authors:**

Please, see weaknesses.

**Limitations:**

Yes.

**Strengths And Weaknesses:**

**Strengths**
1. The goal of understanding the causes of the instabilities of the learning process related to the usage of REINFORCE is relevant, given the recent renewed interest for such an estimator.
2. The authors consider a practical-like scenario, where the state and action spaces are continuous and where the policy gradient is not only computed w.r.t. the mean parameters, but also w.r.t. the policy variance, differently from what done by many theoretical papers (up to my knowledge).
3. The paper is well organized as the settings for characterizing the NSR are presented incrementally w.r.t. their complexity.
4. The study on the NSR could potentially lead to extensions considering PG convergence studies, maybe enabling novel analysis techniques based on the NSR itself instead of the variance of the estimator.

**Weaknesses**
1. The literature review on PGs could be expanded. I suggest to cite [1], which is considered to be the seminal paper introducing REINFORCE. Moreover, I suggest to add a little dissertation on why the G(PO)MDP [2] estimator (i.e., REINFORCE exploiting Markovianity) is not considered here, since it is known to suffer from less variance [3] (I guess that the analysis presented in this paper can easily be translated to incorporate G(PO)MDP). Finally, there is some work that is already characterizing the variance of REINFORCE [3,4,5] in terms of the policy variance (as a special case) as $Var[\hat{G}(\theta)] \le \mathcal{O}(\sigma^{-2})$.
2. I feel that the choice of the study of the NSR could be motivated better. Indeed, the author just say that it is a natural way to characterize how informative is a stochastic gradient relative to its mean. But actually, it appears unclear the practical utility of studying the NSR in place of the variance itself.
3. One of the main takeaways of the paper appears to be that the NSR, besides depending on the interaction length, scales inversely to the policy variance $\sigma^{2}$, either when the NSR can be exactly characterized or just when just an upper bound is available. However, bounds on this kind are already present for the variance of both REINFORCE and G(PO)MDP [3,4,5] which are derived in more general settings and that are already saying that the variance landscape (as the one of the NSR) is blowing up as much as the policy becomes more deterministic. As already said in the previous point, I feel that a discussion on the benefits of studying NSR over the variance lacks.
4. The experimental campaign lacks of more commentary on the hyperparameters used (which I suggest to include in the appendix). Indeed, the training collapse may depend on several factors (up to my knowledge). Moreover, it would have been helpful to see the curve of the variance of the policy to see how sharp the NSR's curve is changing w.r.t. the policy variance (plotted against iterations in addition to the top part of the figures). Furthermore, I would have appreciated the direct comparison against the empirical variance of the gradient estimator, maybe empirically validating the advantages of studying the NSR. Moreover, a study over the optimization keeping the policy variance fixed (too see whether collapses are related to the policy variance learning) would help the experimental campaign. Some simulations on MuJoCo tasks would help as well. Finally, I suggest to include some information regarding the seeding and the uncertainty of the shown results.

**Minors**
1. (right col., line 10) typo: the policy gradient "lemma" is mostly known as the policy gradient "theorem".
2. (right col., line 53) $P$ is not defined. Moreover, here the expected return is defined as $J(\theta)$, while in the rest of the paper it is used as $J(K,\Sigma$). I suggest to unify the notation.
3. Ignoring $s_{0}$ for the rewards seems a little bit strange, but may be just notation. Is there a particular motivation? Since I think that the reward for taking action $a_{0}$ is to be considered in $s_{0}$ instead of in $s_{1}$. But actually, nothing changes in the analysis.
4. (eq. 16) typo: missed "\left(" and "\right)".
5. For notational consistency, I suggest to change $Var_{Fro}$ with $Var_{F}$.
6. Figure 5, the red box is misplaced.

**References**
1. Williams, R. J. (1992). Simple statistical gradient-following algorithms for connectionist reinforcement learning. Machine learning, 8(3), 229-256.
2. Baxter, J., & Bartlett, P. L. (2001). Infinite-horizon policy-gradient estimation. journal of artificial intelligence research, 15, 319-350.
3. Papini, M., Pirotta, M., & Restelli, M. (2022). Smoothing policies and safe policy gradients. Machine Learning, 111(11), 4081-4137.
4. Yuan, R., Gower, R. M., & Lazaric, A. (2022, May). A general sample complexity analysis of vanilla policy gradient. In International Conference on Artificial Intelligence and Statistics (pp. 3332-3380). PMLR.
5. Montenegro, A., Mussi, M., Metelli, A. M., & Papini, M. (2024). Learning optimal deterministic policies with stochastic policy gradients. Proceedings of the 41st International Conference on Machine Learning (ICML). 2024.

---

> ### Author Rebuttal · Authors · 2026-03-31
>
> We thank the reviewer for the thoughtful suggestions. We will revise the paper accordingly, add the requested references, and address all minor comments in the final version. **Figures containing new experimental results that support our responses can be found in the folder:** https://drive.google.com/drive/folders/1AtLZolmcN4TPIXgl9J0tdNFYplXLJmHZ?usp=share_link.
>
> ## **Weakness 1**
>
> We will cite references in the paper. For the GPOMDP estimator in [3,4] and the case $\beta=1$ in [2], it is essentially vanilla policy gradient with half of the terms removed. For $\beta<1$ in [2], it becomes a discounted sum. **Our theorem can be directly applied to these cases by changing the return $R(\tau)$ to running return $G_t$ which is still a quadratic function of the underlying Gaussian variables**. In **Figure R2**, we show the NSR of the REINFORCE and GPOMDP estimators, together with their ratio. The ratio remains nearly constant, around 2-3, and the GPOMDP estimator follows the same qualitative blow-up pattern as the REINFORCE estimator. And please see weakness 3 for discussions on related works.
>
> ## **Weakness 2**
> **First, NSR is not intended to replace variance, but to complement it:** NSR should be viewed as a more informative metric built on top of variance and analyzing NSR necessarily requires computing variance first.
>
> **Second, NSR can be more informative than variance alone for two reasons:**
> - **Theoretically, convergence depends on noise *relative* to signal, not variance alone**. In particular, bounded NSR (strong-growth conditions) can guarantee convergence to an exact stationary point, whereas bounded variance alone typically guarantees convergence only to a neighborhood. Bounded NSR can also imply faster convergence rates; see Corollary 3.5 in [4].
> - **Empirically, variance alone can be misleading:** a point may have large variance yet still admit stable learning if the gradient norm is also large. Near an optimum, even smaller absolute variance can dominate once the true gradient becomes small. **Figure R1** illustrates this: early in training, variance is large but learning is smooth because the signal is strong; later, variance decreases but oscillations grow because the signal weakens and NSR rises. Thus NSR explains convergence behavior more faithfully than variance alone.
>
> ## **Weakness 3**
> We agree that inverse-covariance scaling alone is not the main novelty, and we will clarify this in the revision. Our contribution is not simply to show that the NSR/variance grows as the policy becomes deterministic, but to **(a) provide an *exact* method for computing these quantities**, rather than relying on upper bounds or Monte Carlo estimation, and **(b) to use these exact formulas to reveal how the NSR/variance depends on both policy parameters and system properties**, such as the feedback gain $K$, horizon $T$, transition matrices $(A,B)$, and the initial-state covariance $\Sigma_0$.
>
> Most prior work on REINFORCE/GPOMDP focuses on deriving *upper bounds* on the variance. While such bounds are valuable, they do not reveal the exact dependence of the variance or the tightness of the bound in structured systems. In particular, upper bounds are mainly used to justify the *stability* of policy gradient methods, whereas **understanding the causes of *instability* requires a more precise characterization of the variance/NSR**. To the best of our knowledge, our paper is the first to provide *exact* characterizations of both the variance and the NSR for linear and polynomial systems. These exact formulas make it possible to (a) study optimization dynamics together with the quality of the statistical estimator, as we do in this paper, and (b) they also provide a foundation for future work on precisely characterizing stable and unstable regions for policy gradient methods.
>
> ## **Weakness 4**
> We will add implementation details, hyper-parameters, and seed information in the appendix. For exact NSR landscapes there is no Monte Carlo uncertainty, and for optimization trajectories they consistently show the same trend. Next, we provide a few new experimental results addressing the reviewer’s questions.
> - **The statistical properties plays a major role in policy collapse**. We include additional experiments suggesting that the statistical properties of the gradient estimator play at least a major role in the observed policy-collapse phenomenon (see **Question 2 in response to reviewer 1D7g**), and that the same phenomenon also appears in the MuJoCo Ant environment (see **Weakness 1 in response to reviewer 4WF9**).
> - **Exact NSR/Variance is more accurate than Monte Carlo estimates.** **Figure R3** shows that empirical variance estimated via Monte Carlo can be inaccurate in tracking the true variance / NSR trend, especially in regions with large variance and small gradient.
>
> ## Reference
>
> 6. Zhao, T., Hachiya, H., Niu, G., & Sugiyama, M. (2011). Analysis and improvement of policy gradient estimation. NeurIPS.

---

> > ### Author Rebuttal · Reviewer_KqCc · 2026-04-03
> >
> > I thank the authors for the time spent in addressing my concerns. In my opinion, the discussion on the GPOMDP estimator (both in theory and in practice) would enrich the paper and I suggest to include it in the final version of the paper. Furthermore, I thank the authors for the clarification on the role and the relation between the study of the NSR and the sole variance. I have also appreciated the additional empirical results, that I suggest to include in the final version of the work.
> > After reading the other reviews and rebuttals, I decided to increase my score to 3 at this stage. This is mainly tied to the experimental campaign. While the authors clarified that they quantify exactly the NSR and variance for linear and polynomial systems, they also acknowledged that the connections with training instabilities are supported only empirically (rebuttal to reviewer wY2R). So, I believe a deeper investigation would significantly strengthen the paper's practical claims. I highlight that many experiments run for this rebuttal are in that direction, but in my opinion adding more experimental investigations in high-dimensional control tasks under deep policies, as started in the rebuttal with the Ant experiment, would enrich the paper. However, I am open to discuss with other reviewers.

---

> > > ### Author Response · Authors · 2026-04-07
> > >
> > > We thank the reviewer for the careful reconsideration and for acknowledging that our previous rebuttal addressed the corresponding concerns. We will add the discussion about GPOMDP and more empirical results in the final version.
> > >
> > > To provide additional formal intuition, we include two stylized results showing how variance/NSR blow-up can induce instability under SGD.
> > >
> > > - **Theorem R1** focuses on a generic scalar function: if the variance blows up like $1/\sigma^p$, then once $\sigma$ is sufficiently small, a single SGD step has positive probability of leaving any given radius.
> > >
> > > - **Theorem R2** is a sharper finite-horizon LQG result: the gradient noise itself has a $1/\sigma$ tail with uniformly positive probability, which in turn implies escape of the SGD iterate.
> > >
> > > ### **Theorem R1 [Escape from a fixed ball]**
> > > Consider a function $f$ of a scalar variable $\sigma$. Let $\sigma_k$ satisfy the SGD update
> > > $$\sigma_{k+1}=\sigma_k-\eta(g(\sigma_k)+\varepsilon_{k+1}),
> > > ~\eta>0$$
> > > where $g$ is the gradient and $\varepsilon_{k+1}$ is centered noise.
> > > Assume that there exists a constant $c>0$ such that for all $k$ and all $\sigma_k\neq 0$,
> > > $$
> > > \mathbb E[\varepsilon_{k+1}^2]\ge \frac{c}{|\sigma_k|^p}, ~ p>0.
> > > $$
> > > Assume also that $g$ is continuous at $0$ and that $g(0)=0$.
> > >
> > > Then for every $R>0$, there exists $\delta>0$ such that
> > > $0<|\sigma_k|\le \delta$ gives $\mathbb P(|\sigma_{k+1}|>R)>0$
> > >
> > > ### **Proof sketch**
> > > Define
> > > $$
> > > m(\sigma):=\sigma-\eta g(\sigma).
> > > $$
> > > Since $g$ is continuous at $0$ and $g(0)=0$, we have $m(\sigma)\to 0$ as $\sigma\to 0$. Hence
> > > $$
> > > B_\delta:=\sup_{|\sigma|\le \delta}|m(\sigma)|<\infty,
> > > ~
> > > B_\delta\to 0
> > > ~ \text{as } \delta\downarrow 0.
> > > $$
> > >
> > > Choose $\delta>0$ such that
> > > $$
> > > \frac{c}{\delta^p}>
> > > (\frac{R+B_\delta}{\eta})^2.
> > > $$
> > > Then by contradiction
> > > $$
> > > \mathbb P(|\varepsilon_{k+1}|>\frac{R+B_\delta}{\eta})>0.
> > > $$
> > > Using the update formula together with the bound $|m(\sigma_k)|\le B_\delta$, this implies the conclusion.
> > >
> > > ### **Explanation**
> > >
> > > This shows that if the gradient noise has magnitude $\Theta(1/\sigma^p)$, then the closer the policy gets to the optimum, the more vulnerable it becomes to destabilization by noise. This is consistent with the collapse behavior observed in our experiments.
> > >
> > > ### **Theorem R2 [Escape in LQG]**
> > > Consider the finite-horizon LQG setting of Theorems 4 and 5, with
> > > $$
> > > \Sigma=\sigma^2 I_m, \Sigma_0=\sigma^2 I_n,
> > > $$
> > > where $\sigma_0>0,T<\infty$ and $Q_s\succ 0, Q_a\succ 0, A\neq 0$
> > >
> > > Let $\widehat G_K$ be the REINFORCE estimator of the gradient with respect to $K$, and define
> > > $$
> > > Z_\sigma:=\widehat G_K-\mathbb E[\widehat G_K].
> > > $$
> > > Then there exist constants $c>0$, $c_0>0$, and $\sigma_\ast>0$, independent of $\sigma$, such that for all $0<\sigma\le \sigma_\ast$,
> > > $$
> > > \mathbb P(\|Z_\sigma\|_F\ge \frac{c}{\sigma})\ge c_0.
> > > $$
> > > Moreover, for the SGD update
> > > $$
> > > K^+=K-\eta \widehat G_K,
> > > $$
> > > for every $R>0$ there exists $\sigma_R>0$ such that
> > > $$
> > > 0<\sigma\le \sigma_R
> > > \Longrightarrow
> > > \mathbb P(\|K^+\|_F>R)\ge c_0.
> > > $$
> > >
> > > ### **Proof sketch**
> > > In the exact second-moment decomposition from Theorem 5, the leading term is nonzero and scales like $\Theta(\sigma^{-2})$ so
> > > $$
> > > \mathbb E\|Z_\sigma\|_F^2 \ge \frac{a}{\sigma^2}
> > > $$
> > > for small enough $\sigma$.
> > >
> > > Next, writing $\bar\varepsilon=\sigma\bar\xi$ with $\bar\xi\sim\mathcal N(0,I)$, the lifted state, return, and REINFORCE estimator are polynomials in $(s_0,\bar\xi)$, with leading factor $1/\sigma$. Hence
> > > $$
> > > \widehat G_K = \sigma^{-1}H_1 + H_2 + \sigma H_3 + \sigma^2 H_4,
> > > $$
> > > where the $H_i$ are Gaussian-polynomial random matrices independent of $\sigma$ in distribution. Therefore
> > > $$
> > > \mathbb E\|Z_\sigma\|_F^4 \le \frac{b}{\sigma^4}.
> > > $$
> > >
> > > Applying Paley-Zygmund to $Y_\sigma:=\|Z_\sigma\|^2$ gives
> > > $$
> > > \mathbb P(\|Z_\sigma\|_F\ge \frac{c}{\sigma})\ge c_0.
> > > $$
> > >
> > > Finally, write the SGD step as
> > > $$
> > > K^+
> > > = K-\eta\widehat G_K =
> > > M_\sigma-\eta Z_\sigma,
> > > M_\sigma:=K-\eta\mathbb E[\widehat G_K].
> > > $$
> > > The family $\{M_\sigma\}$ is uniformly bounded for small $\sigma$. Thus, choosing $\sigma$ small enough we get
> > > $$
> > > \mathbb P(\|K^+\|_F>R)\ge c_0.
> > > $$
> > >
> > > ### **Explanation**
> > > This theorem further gives a uniformly positive probability compared to Theorem R1. If the SGD updates have $n$ consective update steps and $K,\sigma$ remains stable, then the probility is $(1-c_0)^n$ which will decay to 0, make the collapse uavoildable.
> > >
> > > A complete rigorous characterization of how NSR/variance causes optimization instability in modern RL is beyond the scope of this paper, since it would also require modeling noise covariance structure and optimizer interaction, especially for adaptive methods. Our focus here is on the exact variance/NSR of REINFORCE, its scaling laws, and its connection to the numerical phenomena observed in our experiments.
> > >
> > > ### **New Experiments**
> > >
> > > We further add two more experiments on Mujoco "Hopper-v5" and "Halfcheetah-v5" (see reply to reviewer 4WF9).

---

### Official Review · Reviewer_1D7g · 2026-03-12

**Soundness:** 3
**Presentation:** 3
**Significance:** 2
**Originality:** 4
**Overall Recommendation:** 4
**Confidence:** 4

**Summary:**

This paper investigates the fundamental causes of training instability and policy collapse in policy-gradient methods through the lens of the **Noise-to-Signal Ratio (NSR)**. While it is well-known that policy-gradient estimators like REINFORCE suffer from high variance, this work moves beyond treating variance as a uniform constant. Instead, it characterizes the non-uniform landscape of NSR, defined as the estimator’s variance normalized by the squared norm of the true gradient, and shows how this ratio evolves as a policy approaches optimality.

The paper’s core contributions are divided into theoretical characterizations across different system classes:

* **Linear-Quadratic Systems:** For finite-horizon linear systems with Gaussian policies, the authors provide exact, closed-form expressions (for one-step) and numerical evaluation procedures (for multi-step) for the NSR. They prove that as a policy becomes more deterministic ($\Sigma \to 0$) or as the initial state distribution becomes broader, the NSR can blow up, scaling inversely with policy covariance.
* **Polynomial Systems:** The authors extend their moment-evaluation framework to systems with polynomial dynamics and feedback, showing that the NSR can still be computed exactly via Gaussian moment evaluation, though complexity increases with the system degree.
* **General Nonlinear Systems:** For more complex cases (like neural policies), the authors derive a general upper bound on the variance, identifying the policy’s Jacobian and the return’s fourth moment as key drivers of noise.
* **Empirical Discovery of the NSR Blow-up:** Through experiments on systems like the double integrator and inverted pendulum, the paper demonstrates a consistent phenomenon: NSR typically increases and can diverge as the policy approaches an optimum. This creates the oscillations and training collapses frequently observed in practice where the signal vanishes while the relative noise explodes.

Ultimately, the work provides a rigorous mathematical foundation for the exploration-exploitation tension in policy gradients: the very process of converging toward a deterministic optimal policy inherently destabilizes the gradient estimator used to get there.

**Compliance With Llm Reviewing Policy:**

Affirmed.

**Final Justification:**

Through the "true gradient" experiments which decouple the statistical noise (NSR) from the optimization landscape, confirming that NSR is a primary driver of policy collapse. The generalization of their theorem to include baselines and the derivation of an entropy-regularized optimal covariance $\Sigma^\star$ provide valuable theoretical insights.

However, the NSR explosion remains a persistent pathology under approximate baselines, and the exact moment-evaluation algorithm's scalability is still restricted to very short horizons ($T=10$). While the paper offers a profound scope into the mechanics of REINFORCE-style updates, its predictive power for complex, long-horizon deep RL tasks remains more qualitative than quantitative.

Nevertheless, the clarity it brings to training instability justifies its publication. I keep Weak Accept rather than a higher score for the remaining gap between theory and broad practice.

**Key Questions For Authors:**

**1. Impact of Baselines on NSR Divergence:**
The current analysis focuses on the vanilla REINFORCE estimator. In practice, baselines (e.g., state-value functions) are crucial for variance reduction. Could you provide a theoretical discussion or a toy-example simulation showing whether the "NSR explosion" near the optimum persists when an optimal or near-optimal baseline is employed? *Reason: If baselines eliminate the divergence of NSR, the paper’s findings might be restricted to a specific sub-class of RL that is rarely used in modern applications.*

**2. Decoupling Causality in Training Instability:**
In Figure 6, fluctuations in the objective and NSR appear synchronized. How do you distinguish between "high NSR causing poor updates" and "the policy entering a high-variance/low-gradient region of the environment, which naturally inflates the NSR"? Could you perform an experiment where you use a large batch size (to artificially suppress noise) at the point of instability to see if the policy still collapses? *Reason: This would clarify whether the instability is a statistical estimation problem (NSR) or an inherent landscape property of the MDP.*

**3. Quantitative Sensitivity to Entropy Regularization:**
Your theory suggests that NSR scales with $O(\text{tr}(\Sigma^{-1}))$. In most PPO/SAC implementations, an entropy coefficient or a minimum $\sigma$ is enforced. Based on your bounds, can you derive a "safe" lower bound for $\sigma$ that keeps the NSR below a critical threshold to guarantee stable convergence? As this would transform the paper’s theoretical contribution into a practical diagnostic tool for RL practitioners.

**4. Complexity of Polynomial Systems:** While Proposition 9 claims exactness, the authors admit computational intractability for high-degree/long-horizon systems. Can you quantify the scalability wall of the moment-evaluation algorithm?

**Limitations:**

Yes

**Strengths And Weaknesses:**

### 1. Soundness
* Strengths: The paper demonstrates rigorous mathematical derivation. The utilization of the Wick/Isserlis theorem to decompose the higher-order moments of the REINFORCE gradient into computable quadratic forms is an ingenious and rigorous approach. The exact computation for LQG and polynomial systems offerred greater determinism than traditional Monte Carlo estimates.
* Weaknesses:
Over-idealized Assumptions: All exact characterizations are based on "Vanilla REINFORCE" (without baselines). In practical RL, state-value function baselines rewrite the variance structure of the gradient. The paper lacks a comparative discussion on how much of the observed NSR explosion stems from intrinsic algorithmic defects versus the lack of standard baseline processing. Since most modern RL algorithms rely on advantage functions, it remains unclear if the identified NSR explosion is a universal problem or one specific to vanilla REINFORCE.

Convergence Logic Chain: The paper observes that increasing NSR leads to SGD oscillations but does not theoretically prove whether the growth rate of NSR exceeds the compensation capacity of learning rate decay or batch size increases.

### 2. Presentation
* Strengths: The paper features an extremely clear structure with a logical progression: starting from the simplest single-step LQG, introducing technical tools (Lemma 1 & 2), extending to multi-step systems (Theorem 5), and finally addressing polynomial and non-linear systems. This narrative approach, moving from the specific to the general, is very easy for expert readers to follow.
* Reproducibility: The paper provides clear matrix recurrence formulas (e.g., forward-backward recursion for $P_t$ and $\Lambda_t$), making the experimental results theoretically straightforward to reproduce.
* Suggestions for Improvement: The heatmaps of NSR in Figure 2 and Figure 5 are very intuitive, but it is recommended to add a quantitative discussion on the "tightness of the upper bound" in the non-linear systems section (Section 4.0).

### 3. Significance

* Strengths: The research addresses a core pain point in RL training—instability in the late stages of training. It provides an intuitive mathematical explanation for "why models tend to crash just as they are about to converge." Linking control stability (spectral radius $\rho(F)$) with gradient estimation noise holds significant interdisciplinary value.
* Limitations:
* Narrow Scope: The conclusions are primarily limited to Gaussian policies and specific dynamical systems. The applicability of this NSR theory remains unverified in discrete action spaces, non-Gaussian noise, or tasks with complex constraints.
* Limited Practical Guidance: The paper identifies the problem but proposes no supporting defense mechanisms (e.g., an algorithm for dynamically adjusting the learning rate based on NSR), which limits its direct reference value for practicing engineers.

### 4. Originality
* Strengths:
Novel Perspective: Past research has focused predominantly on "how to reduce variance," whereas this paper systematically explores "how variance is distributed in the parameter space (Non-uniform landscape)." This quantification of the non-uniformity of the Gradient Landscape is rare in the RL field.
* Technical Integration: Introducing moment evaluation algorithms for polynomial systems into policy gradient analysis represents a creative application of existing mathematical tools.
* Revealing Phenomena: The paper explicitly articulates the paradox that the optimal solution lies in a high-NSR region and experimentally reveals the causal link between policy collapse and a single erroneous gradient step, deepening the understanding of policy gradient fragility.

---

> ### Author Rebuttal · Authors · 2026-03-31
>
> We thank the reviewer for the thoughtful and constructive feedback. Below we respond to the main concerns. **Figures containing new experimental results that support our responses can be found in the folder:** https://drive.google.com/drive/folders/1AtLZolmcN4TPIXgl9J0tdNFYplXLJmHZ?usp=share_link.
>
> ## **Soundness weakness (role of baselines) & Question 1**
>
> We agree that baselines are essential in modern policy-gradient methods. Our paper focuses on the vanilla REINFORCE estimator because it already exhibits a sharp and highly non-uniform NSR landscape in a setting where exact characterization is possible.
>
> To address concerns, we provide **a generalized version of our main theorem that incorporates a baseline $b_t(s_t)$** (see **response to reviewer wY2R, weakness 3**). To summarize, baselines and actor-critic estimators can substantially reduce NSR. However, **avoiding the blow-up generally requires exact value-function estimation; with approximate baselines or critics, the same qualitative pathology can still persist**.
>
> ## **Significance weakness**
>
> We focus on Gaussian policies because they are the standard choice in continuous-control RL and are widely used in practical implementations such as Stable-Baseline3. We agree that discrete-action problems, non-Gaussian exploration, and constrained settings are important, but they are beyond the scope of the paper. For discrete actions, we believe the same blow-up can still occur in bandit settings with or without baselines (see Proposition 3 in [R1]).
>
> ## **Question 2**
>
> We conducted additional analyses and provide more detailed plots.
>
> (1) We show a **detailed plot for policy collapse on quadratic system**. In **Figure R4**, we show the objective, log-standard deviation of policy, Adam momentum and NSR over training. To better visualize the dynamics, **Figure R5** zooms into iterations $45{,}600$-$46{,}400$. The collapse occurs around the 550th iteration in the interval, while the policy covariance is still decreasing. As the NSR grows large, the stochastic gradient estimator becomes more prone to producing inaccurate update directions, and Adam’s momentum correspondingly increases (red line). When the policy covariance reaches its minimum, the NSR is near its maximum (purple line) and the accumulated momentum is also large; at this point, the policy collapses. A similar pattern appears in the green line, where the decrease in policy covariance is accompanied by a rapid growth in optimizer momentum.
>
> (2) We performed several additional experiments suggesting that policy collapse is not explained solely by the optimization landscape, and that **the statistical properties of the gradient estimator play at least a major role**:
> - Increasing the batch size at the point where collapse begins delays the collapse, but does not eliminate it (**Figure R8**).
> - Replacing the stochastic gradient estimator with the true gradient prevents collapse (**Figure R9**).
> - Constraining the policy covariance reduces the oscillation amplitude; once the covariance is prevented from becoming too small, collapse no longer occurs (**Figure R10**).
>
> ## **Question 3**
>
> We appreciate this question. Entropy regularization can indeed be incorporated into our framework, and for Gaussian policies it remains analytically tractable because the added entropy term is explicit in $\Sigma$.
>
> Consider the entropy-regularized finite-horizon objective
> $$J_\tau(K,\Sigma):=J(K,\Sigma)+\tau\mathbb{E}\left[\sum_{t=0}^{T-1}\gamma^t \mathcal{H}(\pi(\cdot | s_t))\right]$$
>
> **On the one hand, the optimal covariance is no longer zero**. In the diagonal setting, the optimal covariance satisfies
> $$\Sigma_{ii}^\star=\frac{\tau G_T}{2M_{ii}(K^\star)},$$
> so entropy regularization yields a strictly positive optimal covariance, in contrast to the unregularized setting where the optimum is deterministic. Plugging such a positive covariance into the variance/NSR expressions can solve for the entropy parameter corresponding to a given upper bound on the resulting NSR.
>
> **On the other hand, this does not prevent the policy from entering the small-covariance regime**. The scaling law remains the same as in the unregularized case, so if SGD takes an incorrect update direction, the policy may still collapse.
>
> ## **Question 4**
>
> We thank the reviewer for pointing this out. The computational complexity grows combinatorially with the polynomial degree and exponentially with the horizon $T$. In our current implementation, for the quadratic system we can compute the exact variance up to $T=10$ in roughly one hour. However, parallelization is possible through memoization of repeated moments/subexpressions, exploitation of sparsity and symmetry in the polynomial expansion. We will add this discussion in the revision.
>
> ## Reference
>
> R1. Mei, J., Chung, W., Thomas, V., Dai, B., Szepesvari, C., & Schuurmans, D. (2022). The role of baselines in policy gradient optimization. NeurIPS, 35, 17818-17830.

---

> > ### Author Rebuttal · Reviewer_1D7g · 2026-04-02
> >
> > The authors have effectively addressed all major concerns, and here are some further discussions.
> >
> > **1. Quantitative Mitigation by Approximate Baselines:**
> > While you successfully generalized the theorem to include baselines, the core concern is the **degree of mitigation**. In your revised analysis, could you provide a sensitivity plot showing the NSR growth curve under varying levels of baseline approximation error (e.g., using a value function $V_{\phi}$ with added noise)? This would clarify if the 'NSR explosion' remains the dominant failure mode in practical Actor-Critic setups.
> >
> > **2. The 'Race' between Batch Size and NSR:**
> > Your observation that increasing batch size only delays collapse is intriguing. Theoretically, does this imply that for the systems studied, the NSR $\eta(\theta)$ grows faster than any polynomial of the distance to the optimum $1/\|\theta - \theta^\star\|$? A brief discussion on the 'scaling race' between batch size and NSR divergence would add significant theoretical depth.
> >
> > **3. From Theory to Diagnostic Tool:**
> > You derived the optimal covariance $\Sigma^\star$ under entropy regularization. To make this paper more impactful for practitioners, could you propose a concrete **'Stability Criterion'** or a heuristic (e.g., a ratio of current $\Sigma$ to $\Sigma^\star$) that could serve as a monitor during training to preemptively detect impending policy collapse?

---

> > > ### Author Response · Authors · 2026-04-07
> > >
> > > We thank the reviewer for the thoughtful follow-up questions and for acknowledging that the main concerns have been addressed. The link to new figures can still be found in https://drive.google.com/drive/folders/1AtLZolmcN4TPIXgl9J0tdNFYplXLJmHZ?usp=share_link. Below we clarify these points and will incorporate the corresponding discussion into the revision.
> > >
> > > ### **Question 1**
> > >
> > > We add a sensitivity plot that varies the baseline approximation error and reports the resulting NSR as the policy covariance changes. In our analytic decomposition, an exact baseline removes the state-only term $s_0^\top M_{ss}s_0$, while an approximate baseline leaves a residual term $s_0^\top D s_0$, where $D$ is a small symmetric error matrix. As clarified in our previous response, even when $D$ is small, the NSR can still blow up in the near-deterministic regime; we verify this behavior in **Figure R11**. Quantitatively, the $E_{1,K}$ contribution to the second moment scales as $\Theta(\|D\|^2)$. Hence, reducing the baseline error magnitude from $0.1$ to $0.01$ reduces this term by approximately $10^{-2}$.
> > >
> > > ### **Question 2**
> > >
> > > We clarify that, in the experiment shown in Figure R8, we do not vary the batch size continuously. Instead, we increase the batch size only after the first collapse occurs.
> > >
> > > Regarding the relation between NSR and the distance to the optimum, note that the distance between the current policy standard deviation $\sigma$ and the optimal standard deviation $0$ is simply $|\sigma|$. Therefore, the NSR scales as $\Theta(1/\sigma^{2})$, that is, quadratically in $1/|\sigma|$.
> > >
> > > To understand the scaling relationship between batch size and NSR, recall that the variance of a stochastic gradient estimator with batch size $N$ scales as $\Theta(1/N)$. Therefore, to keep the estimator variance bounded as $\sigma \to 0$, the batch size must scale as $N = O(1/\sigma^{2})$. We further add an experiment with adaptive batch sizing in **Figure R12** to illustrate this trade-off. While this improves stability, it also significantly increases the runtime; in the limit, the required batch size diverges as the policy approaches the optimum. This supports the same conclusion as Figure R9: sufficiently accurate gradients can prevent collapse.
> > >
> > > ### **Question 3**
> > >
> > > We fully agree with the reviewer that a preemptive monitor would be useful in practice. In the linear setting, the optimal covariance can be computed analytically from the gain matrix $K$ and the system dynamics. This naturally suggests monitoring the ratio
> > > $$
> > > \rho_t = \frac{\sigma_t}{\sigma_t^\star}
> > > $$
> > > in the scalar setting, or, more generally,
> > > $$
> > > \rho_t = \frac{\lambda_{\min}(\Sigma_t)}{\lambda_{\min}(\Sigma_t^\star)}.
> > > $$
> > > This ratio can be incorporated into training either through a hard lower bound or through a soft penalty/barrier, with the goal of preventing the policy covariance from shrinking too far below its entropy-regularized target.
> > >
> > > At the same time, our experiment in Figure R10 shows that enforcing such a lower bound does not fully eliminate policy collapse; rather, it mitigates the oscillation amplitude. Therefore, this criterion should be viewed as a practical stability heuristic rather than a complete cure. More broadly, it reflects a trade-off between stability and accuracy: increasing the entropy coefficient $\tau$ tends to improve stability, but it may also bias the final solution away from the original optimum.

---

### Official Review · Reviewer_4WF9 · 2026-03-13

**Soundness:** 2
**Presentation:** 2
**Significance:** 2
**Originality:** 2
**Overall Recommendation:** 4
**Confidence:** 3

**Summary:**

This paper studies the noise-to-signal ratio (NSR) of the REINFORCE policy-gradient estimator in reinforcement learning. The NSR is defined as the variance of the gradient estimator divided by the squared norm of the true gradient, capturing how noisy gradient estimates are relative to the true optimization signal. The authors argue that increasing NSR during training may explain instability or slowdown often observed in policy-gradient methods. The results reveal that NSR is non-uniform across parameters and time steps, and can grow significantly with the planning horizon, potentially explaining difficulties in training policy-gradient methods.

**Compliance With Llm Reviewing Policy:**

Affirmed.

**Final Justification:**

I think most of my concerns have been resolved by the rebuttal and hence I will lean to acceptance.

**Key Questions For Authors:**

1. Have the authors measured NSR during training of neural policies (e.g., PPO or REINFORCE with neural networks)? Does it increase in the way predicted by your analysis?

2. Can the NSR characterization be used to design improved algorithms (e.g., adaptive baselines, gradient normalization, or horizon-aware updates)?

3. To what extent do the insights obtained from linear/polynomial systems transfer to nonlinear environments and neural network policies?

4. How would the NSR behave for advantage-based estimators or actor–critic gradients? Would similar horizon scaling appear?

**Limitations:**

The main limitations of the work are theoretical scope and practical applicability. First, the exact results rely on simplified system assumptions, which may not reflect modern reinforcement learning settings. Second, the work lacks empirical validation on realistic environments, making it difficult to judge its practical significance. Third, the analysis focuses on diagnosing the problem rather than solving it, leaving open the question of how these insights can improve RL algorithms.

**Strengths And Weaknesses:**

### Weaknesses

1. Strong assumptions limit applicability. Most exact results rely on very restrictive system assumptions, including linear dynamics, Gaussian policies, linear or polynomial feedback, finite horizon settings, etc. These assumptions allow tractable moment computations but significantly restrict the applicability to real RL systems. Modern reinforcement learning typically uses nonlinear environments and neural-network policies, where the presented closed-form NSR formulas cannot be applied. Although the paper later provides a variance upper bound for general nonlinear systems (Theorem 10), this result appears much weaker and does not provide the same level of insight as the exact characterizations. Consequently, it is unclear how much of the paper’s main conclusions extend beyond the specific linear/polynomial settings studied.

2. The paper appears heavily theoretical but provides little empirical evidence demonstrating that NSR behaves as predicted in practical RL systems. In particular, the paper does not convincingly show whether NSR correlates with training instability and whether the theoretical predictions match observed behavior in common RL benchmarks (e.g., MuJoCo). Without empirical validation, it is difficult to assess whether the theoretical results explain real-world training failures or are mostly artifacts of the simplified models.

3. While the analysis identifies situations where NSR becomes large, the paper does not clearly translate these findings into actionable algorithmic improvements.  However, the paper largely stops at analysis without proposing concrete mitigation strategies, which limits the impact of the work.

4. The paper positions itself as analyzing NSR instead of variance, but the conceptual novelty relative to prior variance analyses is somewhat limited. Previous work has extensively studied high variance in REINFORCE and introduced many variance-reduction techniques (baselines, GAE, actor–critic methods). The paper could more clearly articulate how NSR analysis provides fundamentally new insight beyond variance analysis, and whether existing variance-reduction methods implicitly control NSR.

5. The analysis focuses exclusively on REINFORCE, but modern RL methods frequently use actor–critic estimators. It would be informative to compare NSR behavior across these estimators. Without such comparisons, it is unclear whether the results provide insight into modern RL algorithms.

---

> ### Author Rebuttal · Authors · 2026-03-31
>
> We thank the reviewer for the thoughtful and constructive feedback. We believe the central issue raised here is whether the phenomenon we identify is specific to highly structured systems. Our main point is that **exact characterization is currently possible only in structured systems, while experiments in more complex environments provide preliminary empirical evidence of a similar qualitative pattern.** We will revise the paper to make this point clearer. **Figures containing new experimental results that support our responses can be found in the folder:** https://drive.google.com/drive/folders/1AtLZolmcN4TPIXgl9J0tdNFYplXLJmHZ?usp=share_link.
>
> ## **Weakness 1 & 2, Question 1 & 3**
>
> We respond in three parts.
>
> **First, we add new experiments in a MuJoCo environment that exhibit a similar qualitative pattern beyond linear and polynomial systems**. In the new **Figure R6**, we report the objective, policy log-standard deviation, and NSR throughout training on *Ant-v3* as in [R1]. **Figure R7** zooms into the critical region and reveals two policy-collapse events. In both cases, we observe the same recurring pattern:
> $$
> \text{high return} \rightarrow \text{high NSR} \rightarrow \text{policy collapse} \rightarrow \text{return and NSR both decrease}.
> $$
> This behavior is qualitatively consistent with the mechanism hypothesized from our original figures. Together with the neural-policy LQR and pendulum experiments already included in the paper, these new results support the hypothesis that the phenomenon is not confined to the simplest models.
>
> **Second, we begin with linear and polynomial systems not because complex systems are unimportant, but because these are the settings in which the NSR can currently be characterized exactly**. In complex environments such as MuJoCo, NSR and variance must be estimated by Monte Carlo, and **these estimates can be fragile**. Even in the linear setting, where the exact NSR is available, we show the **Monte Carlo estimates can greatly deviate from the ground truth (see **Figure R3**)**. In more complex settings, the estimation noise can become so large that it obscures the very effect one aims to study. Therefore, if the goal is to understand the mechanism precisely rather than heuristically, structured systems are not merely a matter of convenience; **they are the setting in which the question becomes mathematically identifiable**.
>
> **Third, exact characterization in simple systems lets us disentangle two issues that are confounded in more complex environments**: the optimization dynamics themselves and the quality of the stochastic estimator used to drive them. These settings also provide a foundation for future work aimed at precisely characterizing stable and unstable regions for policy gradient methods.
>
> ## **Weakness 3 & Question 2**
>
> Although algorithm design is not the primary goal of this paper, an example algorithmic improvement is **adaptive batch sizing**: since the NSR is highly non-uniform across training, one can increase the batch size when the optimizer enters high-NSR regions. More broadly, our results motivate stabilizing exploration mechanisms, such as maintaining a covariance floor or using entropy control when the learned policy becomes nearly deterministic. We will add this discussion to better highlight the practical implications of the analysis.
>
> ## **Weakness 4 & 5, Question 4**
>
> We agree that baselines and actor-critic estimators are important. **To address this concern, we will add a generalized statement with a baseline/critic term** (see also our **response to reviewer wY2R, Weakness 3**). To summarize, baselines and actor-critic estimators can substantially reduce NSR. However, our analysis suggests that avoiding the blow-up generally requires exact value-function estimation; with approximate learned baselines or critics, residual terms still remain, so the same qualitative pathology can still persist.
>
> Consider the one-step actor-critic estimator
> $$
> \hat G_K^{AC} = \sum_{t=0}^{T-1}
> \Sigma^{-1}\epsilon_t s_t^\top \delta_t^{\hat V},
> \qquad
> \delta_t^{\hat V} = r_t+\gamma \hat V_{t+1}(s_{t+1})-\hat V_t(s_t).
> $$
>
> For **a general critic $\hat V$**, exact Gaussian moment calculations are no longer available, so one typically obtains upper bounds rather than exact formulas. For **structured cases**, however, a well-chosen quadratic critic makes $\delta_t^{\hat V}$ a quadratic function of $(s_t,\epsilon_t)$ and can cancel leading-order terms, potentially reducing the residual to $O(\sigma)$. **But this cancellation is delicate: with an approximate baseline or an imperfect critic, residual terms remain, and the same blow-up mechanism can still persist as the policy covariance shrinks**.
>
> ## Reference
>
> R1. Dohare, S., Hernandez-Garcia, J. F., Lan, Q., Rahman, P., Mahmood, A. R., \& Sutton, R. S. (2024). *Loss of plasticity in deep continual learning*. Nature, 632(8026), 768--774.

---

> > ### Author Rebuttal · Reviewer_4WF9 · 2026-04-03
> >
> > Thank the authors for addressing my concerns. I will increase the score accordingly. More strong experimental results are preferred to strengthen the current version.

---

> > > ### Author Response · Authors · 2026-04-07
> > >
> > > Thank you very much for your positive feedback and for acknowledging our rebuttal. We are grateful that our clarifications addressed your concerns, and we sincerely appreciate your willingness to reconsider the score. We also appreciate your suggestion regarding stronger experimental results; we agree that additional experiments would further strengthen the paper, and **we added two more experiments on Mujoco "Hopper-v5" and "Halfcheetah-v5" environment in Figure R13 and Figure R14**. Additional figures can still be found in https://drive.google.com/drive/folders/1AtLZolmcN4TPIXgl9J0tdNFYplXLJmHZ?usp=share_link
> > >
> > > In these figures, we can see the objective and NSR / Variance consistently show coupled fluctuation during training. This supports our explanation in the rebuttal that when objective increases, NSR / variance also increases, which increase the possibility to collapse and when collapse happens they decrease together.
> > >
> > > Further **we show theoretical evidence that the large variance / NSR can lead to unstable update in SGD (see reply to reviewer wY2R and KqCc)**.

---

### Official Review · Reviewer_wY2R · 2026-03-14

**Soundness:** 3
**Presentation:** 2
**Significance:** 2
**Originality:** 3
**Overall Recommendation:** 4
**Confidence:** 4

**Summary:**

This paper analyzes the noise-to-signal ratio (NSR) of the REINFORCE policy gradient estimator by characterizing the first and second-order moments of the estimator in several settings. For one-step linear systems, the paper derives closed-form expressions for these moments; for multi-step linear systems and polynomial dynamical systems, it provides exact procedures for computing them; and for general nonlinear systems it derives upper bounds on the variance. Using these results, the authors analyze how the NSR depends on system parameters, showing in the linear case that it scales directly with the variance of the initial state and inversely with the exploration noise variance. The paper also presents experiments that visualize the NSR landscape and optimization trajectories, suggesting that large NSR values may correspond to unstable or oscillatory behavior of REINFORCE near optimal policies.

**Compliance With Llm Reviewing Policy:**

Affirmed.

**Final Justification:**

The authors manage to show in their rebuttal that studying the NSR is important for proving convergence of policy gradient, at least in some limited settings.

**Key Questions For Authors:**

[Q.1] The paper argues that large noise-to-signal ratio (NSR) explains instability in REINFORCE training. Could the authors clarify whether their results establish a formal relationship between NSR and instability of the optimization dynamics, or whether NSR should instead be interpreted as a diagnostic quantity that correlates with unstable training behavior?

[Q.2] Since the definition of NSR involves dividing by $||\nabla J(\theta)||^2$, which tends to zero near stationary points, NSR may become large even if the variance of the gradient estimator does not increase. How should one interpret large NSR values near optimal policies in this context?

**Limitations:**

Yes

**Strengths And Weaknesses:**

Strengths

[S.1] The paper provides analytical characterizations of the first and second moments of the REINFORCE gradient estimator across several dynamical-system settings. In particular, it derives closed-form expressions for one-step linear systems and exact procedures for multi-step linear and polynomial systems. The derivations appear technically correct and may be of interest to researchers studying the variance properties of policy gradient estimators.

[S.2] The paper attempts to connect theoretical quantities, specifically the noise-to-signal ratio of the REINFORCE estimator, with observed optimization behavior in reinforcement learning, which is an important question for understanding instability in policy-gradient methods.

Weaknesses

[W.1] The main narrative of the paper appears somewhat stronger than what is formally established. The paper suggests that large noise-to-signal ratio (NSR) explains instability of REINFORCE training, but the theoretical results primarily characterize the moments and variance of the gradient estimator. The paper does not establish a formal link between large NSR and instability of the optimization dynamics, so the causal interpretation should be stated more carefully.

[W.2] The definition of the noise-to-signal ratio
$\mathrm{NSR}(\theta)=\frac{\mathrm{Var}(\widehat{G}_\theta)}{||\nabla J(\theta)||^2}$
naturally becomes large near stationary points where $||\nabla J(\theta)|| \to 0$. As a result, large NSR near optimal policies may arise due to vanishing signal rather than exploding variance, making the interpretation of NSR “blow-up” as a cause of training instability somewhat ambiguous.

[W.3] The analysis focuses on the vanilla REINFORCE estimator without baselines or critic-based variance reduction. While this is a reasonable scope for a theoretical study, most modern policy-gradient algorithms employ baselines, advantage estimators, or actor–critic methods to reduce gradient variance. As a result, it remains unclear to what extent the NSR phenomena identified in the paper extend to these widely used methods.

---

> ### Author Rebuttal · Authors · 2026-03-31
>
> We thank the reviewer for highlighting our current wording may sound stronger than what is formally established. We will revise the paper to state more carefully. Below we respond to the main concerns. **Figures containing new experimental results that support our responses can be found in the folder:** https://drive.google.com/drive/folders/1AtLZolmcN4TPIXgl9J0tdNFYplXLJmHZ?usp=share_link.
>
> ## **Weakness 1 \& Question 1**
>
> We clarify that, at present, the relationship between non-uniform NSR/variance and optimization instability is supported only by empirical observations. **These findings should therefore be interpreted as numerical evidence (and possibly conjectures), rather than as rigorous theoretical statements.** That said, we hope future work may establish formal theoretical connections.
>
> Additionally, we highlight several further experiments that reinforce the observed relationship between NSR/variance and training instabilities:
> - In the detailed ablation study conducted in **response to Reviewer 1D7g (Question 2)**, we examined **training under larger batch sizes, constrained policy covariance, and exact gradients**. These experiments suggest that **statistical properties of the gradient estimator play at least a major role in the observed instability and policy collapse**.
> - Following the discussion in the "loss of plasticity" / "policy collapse" literature [R2], we also measure the NSR and objective during training in the **MuJoCo Ant environment** and observe the same qualitative phenomenon (see **response to Reviewer 4WF9, Weakness 1**).
>
> ## **Weakness 2 \& Question 2**
>
> We agree that NSR can blow up near a stationary point if the variance remains lower bounded. We would like to clarify this from two angles:
> - In Theorems 3 and 5, we also show that the variance itself can be characterized exactly. In our structured settings, the **variance itself can blow up near a stationary point**, so the large NSR is not only due to a vanishing denominator, but can also reflect a genuinely diverging numerator.
> - More generally, if NSR blows up, this means that the variance does not shrink to zero, or shrinks more slowly than the gradient norm. In this case, one typically only obtains an $O(\frac{1}{\epsilon^4})$ iteration complexity rather than $O(\frac{1}{\epsilon^2})$ (see [R1]). Therefore, an increasing NSR near optimality can make SGD slower and less stable near convergence.
>
> ## **Weakness 3**
>
> We provide below a **generalized version of our main theorem that incorporates a baseline $b_t(s_t)$**. To summarize, **baselines and actor-critic estimators can substantially reduce NSR. However, avoiding the blow-up requires exact value-function estimation**.
>
> - **General baseline supports a better upper bound.** Let $\delta_t := G_t - b_t(s_t)$ be the residual on the running return. If $b_t(s_t)$ is an arbitrary function, then $\delta_t$ is generally no longer a polynomial in the Gaussian variables, so the exact Isserlis moment calculations are no longer directly available. In that case, it is more natural to work with upper bounds. Following the same proof pattern as Theorem 10, one is led to the schematic estimate by changing $\sqrt{E[R(\tau)^4\mid s_0]}$ to $\sqrt{E[\delta_t^4\mid s_0]}$. Therefore, when the baseline is accurate and $\delta_t$ is much smaller than $G_t$, the variance upper bound can be substantially improved.
> - **Only exact quadratic baseline in linear system fix blow-up.** For the linear systems in Section 2, if one uses the exact baseline, then the return term can reduce from $O(1)$ to $O(\sigma)$ near the deterministic limit. Since the mean-parameter score scales as
> $$
> \Sigma^{-1}\epsilon = O(\sigma^{-1}),
> $$
>
> the resulting REINFORCE estimator becomes $O(1)$ rather than $O(\sigma^{-1})$, so the corresponding second moment no longer exhibits the vanilla $O(\sigma^{-2})$ blow-up.
>
> However, this cancellation is fragile. If instead we use an approximate baseline
> $$
> \tilde b_q(s_t)=b_q(s_t)+e(s_t),
> $$
>
> where $e(s_t)$ denotes the baseline approximation error, then the residual generally remains $O(1)$, and the estimator again scales as $O(\sigma^{-1})$. In practice, baselines are typically approximated by neural networks and are not exact, so the small-covariance blow-up can still persist even when a baseline is used. We will clarify this discussion in the revision.
>
> And for discussion for actor-critic methods please refer to the **response to reviewer 4WF9 (Weakness 5)**.
>
> ## Reference
>
> R1. Yuan, R., Gower, R. M., & Lazaric, A. (2022, May). *A general sample complexity analysis of vanilla policy gradient*. In International Conference on Artificial Intelligence and Statistics (pp. 3332-3380). PMLR.
>
> R2. Dohare, S., Hernandez-Garcia, J. F., Lan, Q., Rahman, P., Mahmood, A. R., \& Sutton, R. S. (2024). *Loss of plasticity in deep continual learning*. Nature, 632(8026), 768--774.

---

> > ### Author Rebuttal · Reviewer_wY2R · 2026-04-02
> >
> > The rebuttal clarifies that the connection between NSR and optimization instability is currently supported by empirical evidence rather than formal theory, which addresses my concern about overstated claims. The additional discussion on variance scaling and the role of baselines is also helpful. However, the lack of a formal link between NSR and instability remains, and the empirical evidence, while strengthened, is still somewhat limited. Overall, my assessment remains unchanged.

---

> > > ### Author Response · Authors · 2026-04-07
> > >
> > > We thank the reviewer for the careful reconsideration and for acknowledging that our previous rebuttal addressed the corresponding concerns.
> > >
> > > To address the reviewer's remaining concerns, we provide two theorems that formally establish the connection between optimization instability and NSR/variance blow-up.
> > >
> > > - **Theorem R1** focuses on a generic scalar function: if the variance blows up like $1/\sigma^p$, then once $\sigma$ is sufficiently small, **a single SGD step has a positive probability of leaving a ball of any given radius**.
> > >
> > > - **Theorem R2** provides a sharper finite-horizon LQG result: the gradient noise itself has a $1/\sigma$ tail with uniformly positive probability, which in turn implies instability of the SGD iterate.
> > >
> > > ### **Theorem R1 [Escape from a fixed ball]**
> > > Consider a function $f$ of a scalar variable $\sigma$. Let $\sigma_k$ satisfy the SGD update
> > > $$\sigma_{k+1}=\sigma_k-\eta(g(\sigma_k)+\varepsilon_{k+1}),
> > > ~\eta>0$$
> > > where $g$ is the gradient and $\varepsilon_{k+1}$ is centered noise.
> > > Assume that there exists a constant $c>0$ such that for all $k$ and all $\sigma_k\neq 0$,
> > > $$
> > > \mathbb E[\varepsilon_{k+1}^2]\ge \frac{c}{|\sigma_k|^p}, ~ p>0.
> > > $$
> > > Assume also that $g$ is continuous at $0$ and that $g(0)=0$.
> > >
> > > Then for every $R>0$, there exists $\delta>0$ such that
> > > $0<|\sigma_k|\le \delta$ gives $\mathbb P(|\sigma_{k+1}|>R)>0$
> > >
> > > ### **Proof sketch**
> > > Define
> > > $$
> > > m(\sigma):=\sigma-\eta g(\sigma).
> > > $$
> > > Since $g$ is continuous at $0$ and $g(0)=0$, we have $m(\sigma)\to 0$ as $\sigma\to 0$. Hence
> > > $$
> > > B_\delta:=\sup_{|\sigma|\le \delta}|m(\sigma)|<\infty,
> > > ~
> > > B_\delta\to 0
> > > ~ \text{as } \delta\downarrow 0.
> > > $$
> > >
> > > Choose $\delta>0$ such that
> > > $$
> > > \frac{c}{\delta^p}>
> > > (\frac{R+B_\delta}{\eta})^2.
> > > $$
> > > Then by contradiction
> > > $$
> > > \mathbb P(|\varepsilon_{k+1}|>\frac{R+B_\delta}{\eta})>0.
> > > $$
> > > Using the update formula together with the bound $|m(\sigma_k)|\le B_\delta$, this implies the conclusion.
> > >
> > > ### **Explanation**
> > >
> > > This shows that if the gradient noise has magnitude $\Theta(1/\sigma^p)$, then the closer the policy gets to the optimum, the more vulnerable it becomes to destabilization by noise. This is consistent with the collapse behavior observed in our experiments.
> > >
> > > ### **Theorem R2 [Escape in LQG]**
> > > Consider the finite-horizon LQG setting of Theorems 4 and 5, with
> > > $$
> > > \Sigma=\sigma^2 I_m, \Sigma_0=\sigma^2 I_n,
> > > $$
> > > where $\sigma_0>0,T<\infty$ and $Q_s\succ 0, Q_a\succ 0, A\neq 0$
> > >
> > > Let $\widehat G_K$ be the REINFORCE estimator of the gradient with respect to $K$, and define
> > > $$
> > > Z_\sigma:=\widehat G_K-\mathbb E[\widehat G_K].
> > > $$
> > > Then there exist constants $c>0$, $c_0>0$, and $\sigma_\ast>0$, independent of $\sigma$, such that for all $0<\sigma\le \sigma_\ast$,
> > > $$
> > > \mathbb P(\|Z_\sigma\|_F\ge \frac{c}{\sigma})\ge c_0.
> > > $$
> > > Moreover, for the SGD update
> > > $$
> > > K^+=K-\eta \widehat G_K,
> > > $$
> > > for every $R>0$ there exists $\sigma_R>0$ such that
> > > $$
> > > 0<\sigma\le \sigma_R
> > > \Longrightarrow
> > > \mathbb P(\|K^+\|_F>R)\ge c_0.
> > > $$
> > >
> > > ### **Proof sketch**
> > > In the exact second-moment decomposition from Theorem 5, the leading term is nonzero and scales like $\Theta(\sigma^{-2})$ so
> > > $$
> > > \mathbb E\|Z_\sigma\|_F^2 \ge \frac{a}{\sigma^2}
> > > $$
> > > for small enough $\sigma$.
> > >
> > > Next, writing $\bar\varepsilon=\sigma\bar\xi$ with $\bar\xi\sim\mathcal N(0,I)$, the lifted state, return, and REINFORCE estimator are polynomials in $(s_0,\bar\xi)$, with leading factor $1/\sigma$. Hence
> > > $$
> > > \widehat G_K = \sigma^{-1}H_1 + H_2 + \sigma H_3 + \sigma^2 H_4,
> > > $$
> > > where the $H_i$ are Gaussian-polynomial random matrices independent of $\sigma$ in distribution. Therefore
> > > $$
> > > \mathbb E\|Z_\sigma\|_F^4 \le \frac{b}{\sigma^4}.
> > > $$
> > >
> > > Applying Paley-Zygmund to $Y_\sigma:=\|Z_\sigma\|^2$ gives
> > > $$
> > > \mathbb P(\|Z_\sigma\|_F\ge \frac{c}{\sigma})\ge c_0.
> > > $$
> > >
> > > Finally, write the SGD step as
> > > $$
> > > K^+
> > > = K-\eta\widehat G_K =
> > > M_\sigma-\eta Z_\sigma,
> > > M_\sigma:=K-\eta\mathbb E[\widehat G_K].
> > > $$
> > > The family $\{M_\sigma\}$ is uniformly bounded for small $\sigma$. Thus, choosing $\sigma$ small enough we get
> > > $$
> > > \mathbb P(\|K^+\|_F>R)\ge c_0.
> > > $$
> > >
> > > ### **Explanation**
> > > This theorem further gives a uniformly positive probability compared to Theorem R1. If the SGD updates have $n$ consective update steps and $K,\sigma$ remains stable, then the probility is $(1-c_0)^n$ which will decay to 0, make the collapse uavoildable.
> > >
> > > A complete rigorous characterization of how NSR/variance causes optimization instability in modern RL is beyond the scope of this paper, since it would also require modeling noise covariance structure and optimizer interaction, especially for adaptive methods. Our focus here is on the exact variance/NSR of REINFORCE, its scaling laws, and its connection to the numerical phenomena observed in our experiments.
> > >
> > > ### **New Experiments**
> > >
> > > We further add two more experiments on Mujoco "Hopper-v5" and "Halfcheetah-v5" (see reply to reviewer 4WF9).

---

### Decision · Program_Chairs · 2026-04-30

**Decision:**

Accept (regular)

**Comment:**

This paper focuses on the instability or slow-learning-progress issue in deep RL by investigating noise-to-signal ratio of a policy gradient estimator. For certain interesting cases, the authors are able to characterize the ratio exactly. For general nonlinear dynamics and expressive policies, a general upper bound on the variance is derived. Using these tools, it was observed that the ratio increases as the policy converges, sometimes causing blowing up and instability.

The reviewers are overall appreciative of the work and many of their major concerns are resolved during rebuttal. The authors even added a new experiment regarding policy collapse of PPO related to loss of plasticity, providing suggestive empirical evidence that the phenomenon extends to PPO and connects to loss-of-plasticity dynamics.